# Multi-Subspace Multi-Modal Modeling for Diffusion Models: Estimation, Convergence and Mixture of Experts

**Ruofeng Yang**[1,3†], **Yongcan Li**[1†], **Bo Jiang**[1], **Cheng Chen**[2], **Shuai Li**[1,4*]
[1]Shanghai Jiao Tong University, {wanshuiyin, joseph_y, bjiang, shuaili8}@sjtu.edu.cn
[2]East China Normal University, chchen@sei.ecnu.edu.cn
[3]Shanghai Key Laboratory of Intelligent Information Processing
[4]Shanghai Innovation Institute
[†]Equal Contribution [*]Corresponding Author

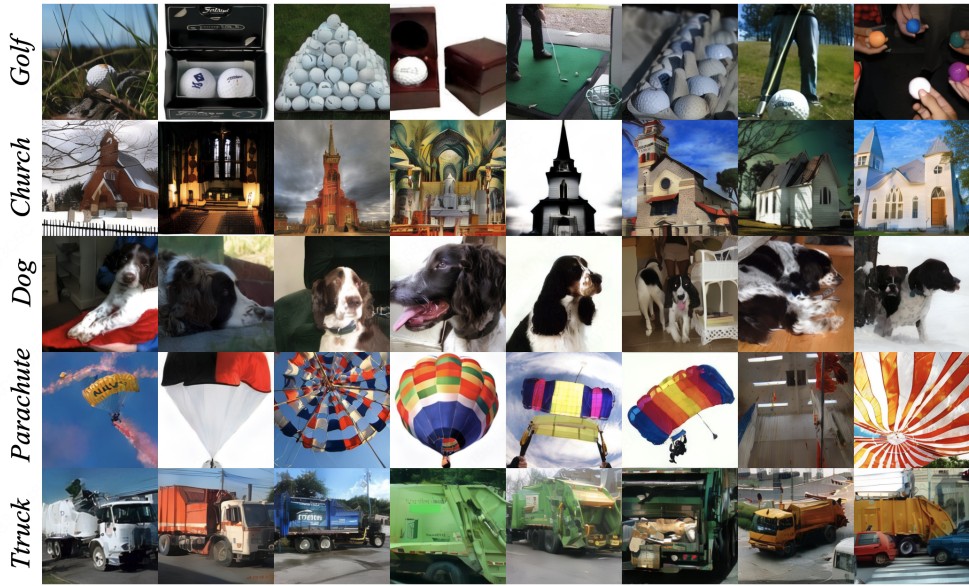

Figure 1: ImageNet results with expert specific VAE and small latent *2-layer Softmax-type network*.

## ABSTRACT

Recent diffusion models demonstrate remarkable sample efficiency and fast optimization, contradicting standard estimation bounds that suffer from the curse of dimensionality $n^{-1/D}$ with the data dimension $D$. Since images are usually a union of low-dimensional manifolds, current works model the data as a union of linear subspaces with Gaussian latent and achieve a $1/\sqrt{n}$ bound. Though this modeling reflects the multi-manifold property, the Gaussian latent can not capture the multi-modal property of the latent manifold. To bridge this gap, we propose the mixture subspace of low-rank mixture of Gaussian (MoLR-MoG) modeling, which models the target data as a union of $K$ linear subspaces, and each subspace admits a mixture of Gaussian latent ($n_k$ modals with dimension $d_k$). With this modeling, the corresponding score function naturally has a mixture of expert (MoE) structure, captures the multi-modal information, and contains nonlinear property. Empirically, our MoE-latent MoG network significantly outperforms MoLRG Gaussian baselines and matches MoE-latent U-Net performance with $10\times$ fewer parameters, validating its practical suitability. Theoretically, we provide provable convergence guarantees for the optimization process and establish an estimation error bound of $R^4 \sqrt{\sum_{k=1}^{K} n_k} \sqrt{\sum_{k=1}^{K} n_k d_k} / \sqrt{n}$, successfully escaping the dimensionality curse. Collectively, with MoLR-MoG modeling, this work explains why diffusion

models only require a small training sample and enjoy a fast optimization process. Furthermore, we also show the potential of MoE structure for diffusion models from the manifold perspective.

# 1 INTRODUCTION

Recently, diffusion models have achieved impressive performance in many areas, such as 2D, 3D, and video generation (Rombach et al., 2022; Ho et al., 2022; Chen et al., 2023a; Ma et al., 2024b; Liu et al., 2024; Tan et al., 2024; 2025; Ma et al., 2024a; Feng et al., 2025; Yan et al., 2025). Due to the score matching technique, diffusion models enjoy a more stable training process and can achieve great performance with a small training dataset.

Despite empirical successes, theoretical guarantees for score estimation in diffusion models often suffer from the curse of dimensionality. Standard bounds for architectures like deep ReLU networks and diffusion transformers yield a minimax rate of $n^{-s'/D}$ ($s'$ is the smoothness parameter of the score function ) (Oko et al., 2023; Hu et al., 2024b;a; Fu et al., 2024), which fails to explain their sample efficiency on high-dimensional data. To bridge this gap, recent works exploit specific structural assumptions about target distributions. One line of research adopts multi-modal modeling, such as Mixture of Gaussians (MoG), to better capture real-world data and improve estimation bounds (Shah et al., 2023; Cui et al., 2023; Chen et al., 2024b; Zhang et al., 2025). Another approach assumes data resides on a low-dimensional linear subspace ($x = Az$ with bounded $z \in \mathbb{R}^d$), successfully reducing the estimation error to $n^{-2/d}$ and eliminating the dependence on the ambient dimension $D$ (Chen et al., 2023b; Yuan et al., 2023; Guo et al., 2024). However, as shown in Brown et al. (2023) and Kamkari et al. (2024), though the image dataset admits low dimension, it is a union of manifolds instead of one manifold. Inspired by this observation, Wang et al. (2024) model the image data as a union of linear subspaces, assume each subspace admits a low-dimensional Gaussian (mixture of low-rank Gaussians (MoLRG)), and achieve a $1/\sqrt{n}$ estimation error. Though the union of the linear subspace is closer to the real-world image dataset, the latent Gaussian assumption is far away from the low-dimensional multi-modal manifold (Brown et al., 2023). Hence, the following two natural questions remain open:

*Can we propose a modeling that reflects the multi-manifold multi-modal property of real-world data?*

*Can we escape the curse of dimensionality and enjoy a fast convergence rate based on this modeling?*

In this work, for the first time, we propose and analyze the mixture of low-rank mixture of Gaussian (MoLR-MoG) distribution, which is more realistic than MoLRG since it captures the multi-modal property of real-world distribution and has a nonlinear score function. Based on this modeling, we first induce a MoE-latent nonlinear score function and conduct experiments to show that MoLR-MoG modeling is closer to the real-world data. After that, we simultaneously analyze the estimation and optimization error of diffusion models and explain why diffusion models achieve great performance.

## 1.1 OUR CONTRIBUTION

**MoLR-MoG Modeling and MoE Structure Nonlinear Score.** We propose the MoLR-MoG modeling for the target data, which captures the multi low-dimensional manifold and multi-modal property of real-world data and naturally introduces the MoE-latent MoG score. Through the real-world experiments, we show that with this score, diffusion models can generate images that is comparable with the deep neural network MoE-latent Unet and only has $10\times$ smaller parameters. On the contrary, the MoE-latent Gaussian score induced by previous MoLRG modeling can only generate blurry images, which indicates MoLR-MoG is a suitable modeling for the real-world data.

**Take Advantage of MoLR-MoG to Escape the Curse of Dimensionality.** For the estimation error, we show that by taking advantage of the union of a low-dimensional linear subspace and the latent MoG property, diffusion models escape the curse of dimensionality. More specifically, we achieve the $R^4\sqrt{\Sigma_{k=1}^{K}n_k}\sqrt{\Sigma_{k=1}^{K}n_k d_k}/\sqrt{n}$ estimation error, where $R$ is the diameter of the target data, $d_k$ is the latent dimension and $n_k$ is the number of the modal in the $k$-the subspace. This result clearly shows the dependence on the number of linear subspaces, modal, and the latent dimensions $R, d_k$.

**Strongly Convex Property and Convergence Guarantee.** After directly analyzing the estimation error, we study how to optimize the highly non-convex score-matching objective function. Facing nonlinear latent MoG scores, we use the gradient descent (GD) algorithm to optimize the objective function. To obtain the convergence guarantee, we take advantage of the closed form of nonlinear MoG score and show that the landscape around the ground truth parameter is strongly convex. Then, with a great initialization area, we prove the convergence guarantee when considering MoLR-MoG.

## 2 RELATED WORK

**Estimation Error Analysis for Diffusion Models.** As shown in Section 1, a series of works Oko et al. (2023) study the general target data with a deep NN and achieve the minimax $n^{-s'/D}$ result. Then, some works analyze the general target data with a 2-layer wide NN and achieve $n^{-2/5}$ estimation error with $\exp(n)$ NN size (Li et al., 2023; Han et al., 2024). For the multi-modal modeling, some works study MoG data and improve the estimation error (Shah et al., 2023; Cui et al., 2023; Chen et al., 2024b). Except for the MoG modeling, Cole and Lu (2024) assume data is close to Gaussian and then prove the model escapes the curse of dimensionality. Mei and Wu (2023) analyze Ising models and prove that the term corresponds to $n$ is $1/\sqrt{n}$. For the low-dimensional modeling, some works assume the target data admits a linear subspace (Chen et al., 2023b; Yuan et al., 2023). Chen et al. (2023b) assume data admit a linear subspace $x = Az$ with $z \in \mathbb{R}^d$ and achieve a $n^{-2/d}$. As the image is a union of low-dimensional manifolds, Wang et al. (2024) models the target data as a union of linear subspaces with Gaussian latent and achieve $1/\sqrt{n}$ estimation error for each subspace.

**Optimization Analysis for Diffusion Models.** Since the score is highly nonlinear (except for Gaussian), only a few works analyze the optimization process, and most of them focus on the external dimensional space (Bruno et al., 2023; Cui and Zdeborová, 2023; Shah et al., 2023; Chen et al., 2024b; Li et al., 2023; Han et al., 2024). Since the score function of MoG has a nonlinear closed-form, a series of works design algorithms for diffusion models to learn the MoG (Bruno et al., 2023; Cui and Zdeborová, 2023; Shah et al., 2023; Chen et al., 2024b). For the general target data, Li et al. (2023) and Han et al. (2024) adopt a wide 2-layer ReLU NN to simplify the problem to a convex optimization. However, as discussed above, their NN has $\exp(n)$ size. For the latent space, Yang et al. (2024a) assume target data adopts a linear subspace with Gaussian latent and provide the closed-form minimizer. Wang et al. (2024) analyze the optimization process of each linear subspace separately, which is also reduced to the optimization for the Gaussian.

## 3 PRELIMINARIES

First, we introduce the basic knowledge and notation of diffusion models. Let $p_0$ be the data distribution. Given $x_0 \sim p_0 \in \mathbb{R}^D$, the forward process is defined by:

$$\mathrm{d}x_t = f(t)x_t \, \mathrm{d}t + g(t) \, \mathrm{d}B_t,$$

where $\{B_t\}_{t \in [0,T]}$ is a $D$-dimensional Brownian motion, $f(t)$ is the coefficient of the drift term and $g(t)$ is the coefficient of the diffusion term. Let $p_t$ be the density function of the forward process. After determining the forward process, the conditional distribution $p_t(x_t|x_0)$ has a closed-form

$$p_t(x_t|x_0) = \mathcal{N}\left(x_t; s_t x_0, s_t^2 \sigma_t^2 I_D\right),$$

where $s_t = \exp\left(\int_0^t f(\xi)\mathrm{d}\xi\right), \sigma_t = \sqrt{\int_0^t g^2(\xi)/s^2(\xi)\mathrm{d}\xi}$. To generate samples from $p_0$, diffusion models reverse the given forward process and obtain the following reverse process (Song et al., 2020):

$$\mathrm{d}y_t = \left[f(t)y_t - g(t)^2 \nabla \log p_t(y_t)\right] \mathrm{d}t + g(t)\mathrm{d}\bar{B}_t, \quad y_0 \sim p_0$$

where $\bar{B}_t$ is a reverse-time Brownian motion. A conceptual way to approximate the score function is to minimize the score matching (SM) objective function:

$$\min_{s_\theta \in \text{NN}} \mathcal{L}_{\text{SM}} = \int_\delta^T \mathbb{E}_{x_t \sim q_t} \|\nabla \log p_t(x_t) - s_\theta(x_t, t)\|_2^2 \, \mathrm{d}t, \tag{1}$$

where NN is a given function class and $\delta > 0$ is the early stopping parameter to avoid a blow-up score. Since the ground truth score $\nabla \log p_t$ is unknown, this objective function can not be calculated. To avoid this problem, Vincent (2011) propose the denoised score matching (DSM) objective function:

$$\min_{s_\theta \in \text{NN}} \mathcal{L}_{\text{DSM}} = \int_\delta^T \mathbb{E}_{x_0 \sim q_0} \mathbb{E}_{x_t|x_0} \|\nabla \log p_t(x_t|x_0) - s_\theta(x_t, t)\|_2^2 \, \mathrm{d}t.$$

As shown in Vincent (2011), the DSM and SM objective functions differ up to a constant independent of optimized parameters, which indicates these objective functions have the same landscape.

## 3.1 Mixture of low-rank mixture of Gaussian (MoLR-MoG) Modeling

This part shows our MoLR-MoG modeling, which reflects the low-dimensional (Gong et al., 2019) and multi-modal property (Brown et al., 2023; Kamkari et al., 2024) of real-world data. More specifically, we assume the data distribution lives near a union of $K$ linear subspaces rather than arbitrary manifolds. Concretely, for the $k$-th subspace (represented by a matrix $A_k^* \in \mathbb{R}^{D \times d_k}$ with orthonormal columns or the $k$-th manifold), we place a $n_k$-modal MoG within that subspace:

$$w_k(x) = \sum_{l=1}^{n_k} \pi_{k,l} \mathcal{N}\big(x; A_k^* \mu_{k,l}^*, A_k^* \Sigma_{k,l}^* A_k^{*\top}\big),$$

where covariance $\Sigma_{k,l}^* = U_{k,l}^* U_{k,l}^{*\top}, l = 1, \ldots, n_k$ with $U_{k,l}^* \in \mathbb{R}^{d_k \times d_{k,l}}$ $(d_{k,l} \leq d_k)$ and $\mu_{k,l}^*$ is the mean of the $l$-th modal of the $k$-th subspace. As shown in (Brown et al., 2023), the different manifold has different $d_k$ and we do not require that $d_k$ is exactly the same for each manifold. Then, the target distribution has the following form

$$p_0 = \sum_{k=1}^{K} \frac{1}{K} \sum_{l=1}^{n_k} \pi_{k,l} \mathcal{N}\big(x; A_k^* \mu_{k,l}^*, A_k^* \Sigma_{k,l}^* A_k^{*\top}\big). \tag{2}$$

From the universal approximation perspective, by placing enough components and choosing parameters $\{\pi_{k,l}, \mu_{k,l}^*, \Sigma_{k,l}^*\}$, a MoG can approximate any smooth density arbitrarily well, which is more general than the Gaussian latent of Yang et al. (2024a) and Wang et al. (2024).

**Mixture of Experts (MoE)-Nonlinear MoG score.** Let $\gamma_t = s_t \sigma_t$, $\Sigma_{k,l,t,A} = s_t^2 A_k^* U_{k,l}^* U_{k,l}^{*\top} A_k^{*\top} + \gamma_t^2 I$ and $\delta_{k,l,t,A}(x) = x - s_t \mu_{k,l}^* - \frac{s_t^2}{s_t^2 + \gamma_t^2} A_k^* U_{k,l}^* U_{k,l}^{*\top} A_k^{*\top}(x - s_t \mu_{k,l}^* A_k^*)$. Under the MoLR-MoG modeling, the score function has the following form:

$$\nabla \log p_t(x) = -\frac{1}{\gamma_t^2} \frac{\sum_{k=1}^{K} \frac{1}{K} \sum_{l=1}^{n_k} \pi_{k,l} \mathcal{N}(x; s_t \mu_{k,l}^* A_k^*, A_k^* \Sigma_{k,l,t,A}^* A_k^{*\top}) \delta_{k,l,t,A}(x)}{\sum_{k=1}^{K} \frac{1}{K} \sum_{l=1}^{n_k} \pi_{k,l} \mathcal{N}(x; s_t \mu_{k,l}^* A_k^*, A_k^* \Sigma_{k,l,t,A} A_k^{*\top})},$$

This score function has a MoE structure, where each expert is the latent nonlinear MoG score. The linear encoder $A_k$ first encodes images to the $k$-th manifold, and diffusion models run the denoising process. After that, the linear decoder $A_k^\top$ decodes the denoised latent to the full-dimensional images. Since the estimation error introduced by the linear encoder and decoder has the order $D d_k^3 / \sqrt{n}$ (Yang et al., 2024a) and is not the dominant term, we assume the linear encoder and decoder are perfectly learned and focus on the more difficult latent MoG diffusion part in this work. From the empirical part, this operation is similar to using the pretrained stable diffusion VAE and only training the diffusion models in the latent space. For the $k$-th low-dimensional manifold, the score function is

$$\nabla \log p_{t,k}(x^{\mathrm{LD}}) = -\frac{1}{\gamma_t^2} \frac{\sum_{l=1}^{n_k} \pi_{k,l} \mathcal{N}(x^{\mathrm{LD}}; s_t \mu_{k,l}^*, \Sigma_{k,l,t}^*) \delta_{k,l,t}(x^{\mathrm{LD}})}{\sum_{l=1}^{n_k} \pi_{k,l} \mathcal{N}(x; s_t \mu_{k,l}^*, \Sigma_{k,l,t}^*)}, \tag{3}$$

where $x^{\mathrm{LD}} \in \mathbb{R}^{d_k}$ is a variable in the $k$-th low-dimensional subspace, $\Sigma_{k,l,t} = s_t^2 U_{k,l}^* U_{k,l}^{*\top} + \gamma_t^2 I$ and $\delta_{k,l,t}(x^{\mathrm{LD}}) = x^{\mathrm{LD}} - s_t \mu_{k,l}^* - \frac{s_t^2}{s_t^2 + \gamma_t^2} U_{k,l}^* U_{k,l}^{*\top}(x^{\mathrm{LD}} - s_t \mu_{k,l}^*)$. Let

$$s_k^*(x^{\mathrm{LD}}, t) = \nabla \log p_{t,k}(x^{\mathrm{LD}}), \quad s^*(x^{\mathrm{LD}}, t) = (s_1^*(x^{\mathrm{LD}}, t), s_2^*(x^{\mathrm{LD}}, t), \ldots, s_K^*(x^{\mathrm{LD}}, t)),$$

where the parameters are $\theta^* = \{\mu_{k,l}^*, U_{k,l}^*\}_{k=1,\ldots,K}$. In this work, we want to learn the parameters of the ground truth score function. Hence, we construct a NN function class $s_\theta = (s_1(\cdot, \cdot), s_2(\cdot, \cdot), \ldots, s_K(\cdot, \cdot))$ according to the above closed-from of MoE-latent MoG score. Let $\theta$ is the union of $\mu_{k,l}$ and $U_{k,l}$. Since we mainly focus on the estimation and optimization in the latent subspace, we omit the superscript LD of the latent subspace when there is no ambiguity.

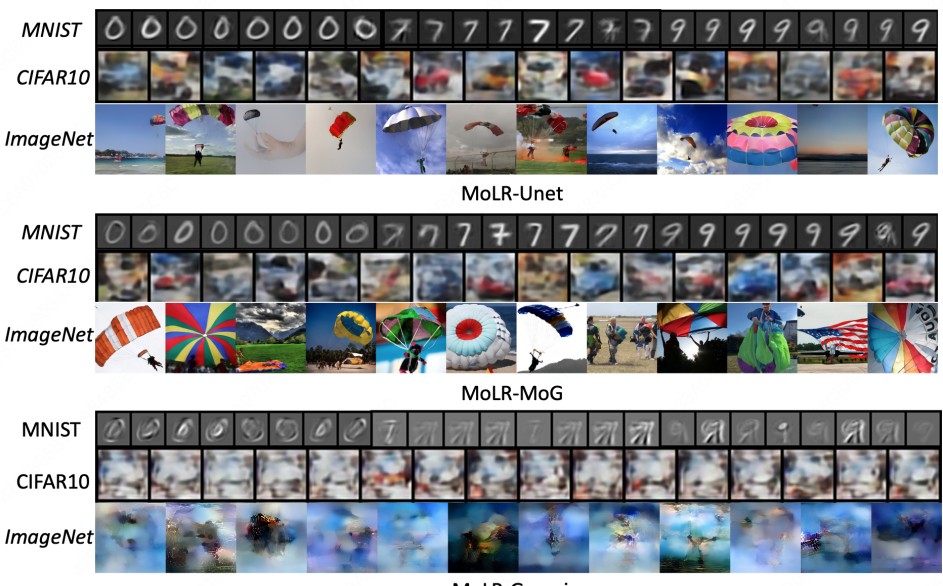

Figure 3: Results of Different Modeling on Real-world Data.

We note that this modeling can capture the information of each low-dimensional manifold and the multi-modal property of each latent distribution. In the next section, through the real-world experiments, we show that the MoE-latent MoG score has a better performance compared with the MoE-latent Gaussian

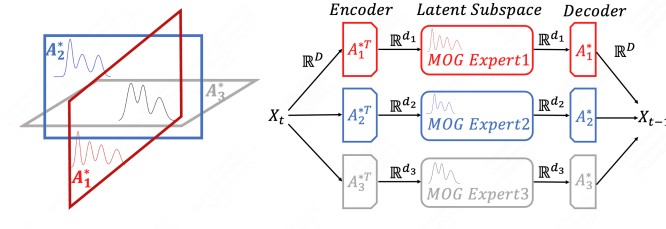

**(a) MoLR-MoG Modeling** **(b) MoE-nonlinear MoG Score**

Figure 2: MoLR-MoG Modeling and Corresponding Nonlinear Score

score induced by MoLRG modeling and compatible with the results of the MoE-latent Unet. In Section 5 and 6, we prove that by using the property of MoLR-MoG modeling, diffusion models can escape the curse of dimensionality and enjoy a fast convergence rate.

*Remark* 3.1 (Comparison with MoLRG modeling). Wang et al. (2024) provide the first multi-subspace modeling, which is an important and meaningful step and Li et al. (2025) further design noisy MoLRG to study the representation of diffusion models. However, they assume a Gaussian latent with $0$ mean, which can not capture the multi-modal property of real-world data. We also note that the MoLR-MoG modeling can not be viewed as MoLRG with $\sum_{k=1}^{K} n_k$ subspace since this modeling assumes there are $\sum_{k=1}^{K} n_k$ VAE, which is not reasonable in the real-world setting.

## 4 EXPERIMENTS FOR MOE-LATENT MOG SCORE

In this section, we empirically validate that our MoLR-MoG framework effectively models real-world data distributions. Specifically, we demonstrate that the MoE-latent MoG score not only generates semantically clear images that significantly outperform the MoLRG baseline, but also achieves performance comparable to a general MoLR-U-Net while requiring $10\times$ fewer parameters across MNIST, CIFAR-10, and ImageNet 256 (Figure 3).

Following Brown et al. (2023), we train 10 VAEs for each number in the MNIST, which represents our $K$ low-dimensional manifold. In this part, we adopt nonlinear VAEs to achieve a good performance in real-world datasets. However, we still note that a series of theoretical works adopt linear subspaces, and our MoLR-MoG modeling with linear VAEs makes a step toward explaining the good performance of diffusion models. Subsequently, we train diffusion models comparing three parameterizations: a latent U-Net, a latent MoG NN (parameterized via Eq. 3 with $n_k \in \{4, 8, 40\}$

for MNIST, CIFAR-10, and ImageNet 256, respectively, for $k \in [K]$), and a latent Gaussian NN utilizing a closed-form linear score (Wang et al., 2024).

**Discussion.** From a qualitative perspective, as shown in Figure 3, the generation results with MoLRG modeling are difficult to distinguish specific numbers. On the contrary, the MoE-latent MoG can generate clean images comparable with the images generated by MoLR-Unet, which means this modeling captures the multi-modal property of each low-dimensional manifold. The training loss curve (Figure 4) shows that the loss of MoE-MoG NN is significantly smaller than the MoE-Gaussian and close to MoE-Uet, which indicates MoE-MoG NN efficiently approximates the ground-truth score and supports our theoretical results. From a quantitative perspective, we calculate the CLIP score for the

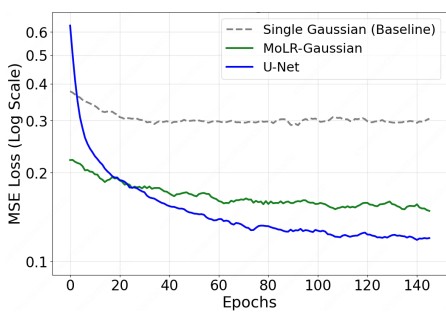

Figure 4: Loss Curve for CIFAR-10

parachute class of ImageNet with text prompts "a photo of parachute". The Clip score for MoLR with Unet, MoG, and Gaussian NN is $0.304$, $0.293$, and $0.254$, which indicates MoLR-MoG achieves almost comparable text-to-image alignment with MoE-Unet. Furthermore, the MoLR-MoG NN contains many fewer parameters compared to Unet since it uses the prior of latent MoG.

**Discussion on Expert-Specific VAE.** As shown in the score of MoLR-MoG, different from latent diffusion models with a single VAE, there are $K$ VAEs to encode the input to the corresponding manifold. We note that this operation is important for MoLR-MoG with small MoG experts. As shown in Figure 5, with a unified VAE, the unified latent is complex, and a MoG expert can not learn a meaningful image with the target class. Hence, with a unified VAE, latent diffusion models require a large latent Unet. However, with an expert-specific VAE (for example, we fine-tune the pretrained VAE with the parachute class dataset), the latent manifold becomes simple, and latent MoG experts are enough to generate clear models, which also supports our theoretical modeling.

We note that these experiments aim to show that the MoLR-MoG modeling is reasonable instead of achieving the SOTA performance. It is possible to achieve great performance with a small-sized NN using MoLR-MoG modeling in the application. For large-scale datasets without labels, we can use a clustering algorithm to divide the data into different clusters. Then, we can train a VAE encoder, decoder, and latent MoG score for each cluster. For the VAE training, we do not require training the VAE from a sketch. We can LoRA fine-tune a VAE pretrained on large-scale datasets (for example, DC-AE (Chen et al., 2024a) for our ImageNet experiments) for each expert, which shares a pretrained VAE backbone and has a smaller model size. When generating images, we activate different VAE LoRA according to the clustering weight, which matches the spirit of MoE. We leave it as an interesting future work.

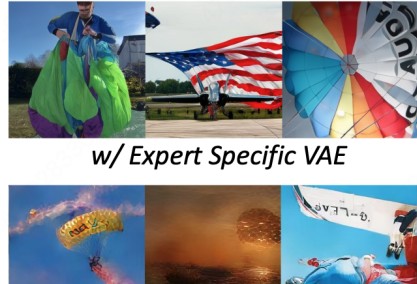

*w/ Expert Specific VAE*

*w/ Unified VAE*

Figure 5: MoLR-MoG with Different VAE

## 5 ESCAPE THE CURSE OF DIMENSIONALITY WITH MoLR-MoG MODELING

This section shows that diffusion models can escape the curse of dimensionality by using MoLR-MoG properties. Before introducing our results, we first introduce the assumption on the target data.

**Assumption 5.1.** For $x \sim p_0$, we have that $\|x\|_2 \leq R$.

The bounded-support assumption is widely used in theoretical works (Chen et al., 2022; Yang et al., 2024b; Bortoli, 2022; Yang et al., 2025a;b) and is naturally satisfied by image datasets. For a latent MoG, each component concentrates almost all mass within a few standard deviations of its mean, so by taking the most component means and variances, one can choose $R$ large enough that $\|x\|_2 \leq R$ holds with high probability.

Since Moe-latent MoG score has a closed-form, we only need to learn the parameters $\mu_{k,l}$ and $U_{k,l}$ at a fixed time $t$. As a result, we consider the estimation error at a fixed time $t$. Let $\ell(\theta; x, t) = \left\| s_\theta(x, t) - s^*(x, t) \right\|_2^2$ be the per-sample squared error at time $t$. In this part, we study the estimation error with a limited training dataset $\{x_i\}_{i=1}^n$:

$$\left| \mathcal{L}(\theta) - \widehat{\mathcal{L}}_n(\theta) \right|, \text{ with } \widehat{\mathcal{L}}_n(\theta) = \frac{1}{n} \Sigma_{i=1}^n \ell(\theta; x_i, t).$$

To obtain the estimation error, we first provide the Lipschitz constant for $s_\theta$ and the loss function by fully using the property of MoLR-MoG modeling and MoE-latent MoG score.

**Lemma 5.2.** *[Lipschitz Continuity] Let $L_{\mu_l}$ and $L_{U_k}$ be the Lipschitz constant w.r.t. $s_\theta$. With MoLR-MoG modeling and Theorem 5.1, there is a constant*

$$L \leq \sqrt{\Sigma_{i=1}^K n_k (L_{\mu_l}^2 + L_{U_k}^2)} = O\left( (\Sigma_{k=1}^K n_k)^{\frac{1}{2}} C_w \right)$$

*such that for any $\theta, \theta'$, $\left\| s_\theta(x, t) - s_{\theta'}(x, t) \right\|_2 \leq L \left\| \theta - \theta' \right\|_2$, where $C_w = \frac{(R + s_t B_\mu)^3 s_t^2}{\gamma_t^4}, B_\mu = \max_{k,l} \|\mu_{k,l}\|_2$. For $s_\theta$ and $s^*$, we have that $2\|s_\theta(x, t) - s^*(x, t)\|_2 \leq 2(R + s_t B_\mu)/\gamma_t^2 := L_l$.*

Then, we obtain the Lipschitz constant $L' = L_l L$ for the whole loss function. With this Lipschitz property, the next step is to argue that fitting the network on $n$ samples generalizes to the true population loss. We do so by controlling the Rademacher complexity of the loss class and then using a Bernstein concentration argument to obtain the following theorem.

**Theorem 5.3.** *Denote by $\widehat{\mathcal{L}}_n(\theta)$ the empirical loss on $n$ i.i.d. samples and by $\mathcal{L}(\theta)$ its population counterpart. Then there exist constants $C_1, C_2$ such that with probability at least $1 - \delta$, for all $\theta \in \Theta$,*

$$\left| \mathcal{L}(\theta) - \widehat{\mathcal{L}}_n(\theta) \right| \leq O\left( C_1 \frac{(R + s_t B_\mu)^4 s_t^2 \sqrt{\Sigma_{k=1}^K n_k}}{\gamma_t^6} \sqrt{\frac{\Sigma_{k=1}^K n_k d_k}{n}} + C_2 \sqrt{\frac{\log(1/\delta)}{n}} \right).$$

*where $C_1 = \max_{\theta \in \Theta} \|\theta_i - \theta_j\|_2$, $C_2 = \sigma \log 2$, $\sigma^2 = \sup_{\theta \in \Theta} Var[\ell(\theta; X, t)]$.*

This result removes the exponential dependence on $D$ with the number of latent subspace $K$, the latent dimension $d_k$, and the number of modalities $n_k$ at each linear subspace, which reflects the key feature of the real-world data and escape the curse of dimensionality. The remaining question is why diffusion models enjoy a fast and stable optimization process. In the next part, we show that with MoLR-MoG modeling, the objective function is locally strongly convex and answer this question.

## 6    STRONGLY CONVEX PROPERTY AND CONVERGENCE GUARANTEE

In this part, by using the property of MoLR-MoG modeling, we derive explicit expressions for the *Jacobian* and *Hessian* of the objective function for 2-modal MoG latent and general MoG latent. Then, we establish conditions under which the resulting score-matching loss is locally strongly convex for each setting. Finally, we provide the convergence guarantee for the optimization.

### 6.1    2-MODAL LATENT MoG HESSIAN ANALYSIS AND OPTIMIZATION

In this section, we show that, under sufficient cluster separation, the Hessian matrix near $\theta^*$ simplifies to a block-diagonal form, yielding local strong convexity, which derives a linear convergence rate. As discussed in Section 3.1, following the real-world setting, we consider the optimization dynamic in the $k$-th latent subspace. While our modeling contains $K$ encoders and decoders, facing an input image $x$, we can first determine which cluster image $x$ belongs to, and then use the corresponding $A_k$ to encode it into the corresponding latent space. Then, we only use data belonging to $k$ clustering to train the $k$-th latent MoG score. This operation matches our experimental settings, and Wang et al. (2024) also adopts this operation. When considering the optimization problem, to simplify the calculation of the Hessian matrix, we set $d_{k,l} = 1$.

Similar to Shah et al. (2023), we start from a latent 2-modal MoG with the same covariance matrix $\Sigma_k^*$ and $\mu_{k,1}^* = \mu_k^*, \mu_{k,2}^* = -\mu_k^*$, which leads to the following score:

$$\nabla \log p_{t,k}(x) = -\frac{1}{\gamma_t^2} \frac{\frac{1}{2}\mathcal{N}(x; s_t \mu_k^*, \Sigma_k^*)\, \delta_k'(x) + \frac{1}{2}\mathcal{N}(x; -s_t \mu_k^*, \Sigma_k^*)\, \epsilon_k(x)}{\frac{1}{2}\mathcal{N}(x; s_t \mu_k^*, \Sigma_k^*) + \frac{1}{2}(x; -s_t \mu_k^*, \Sigma_k^*)}, \tag{4}$$

where $\epsilon_k(x) = x - s_t\mu_k^* - \frac{s_t^2}{s_t^2+\gamma_t^2}U_k^*U_k^{*\top}(x - s_t\mu_k^*)$, and $\delta_k'(x) = x + s_t\mu_k^* - \frac{s_t^2}{s_t^2+\gamma_t^2}U_k^*U_k^{*\top}(x + s_t\mu_k^*)$. Before providing the convergence guarantee, we make an assumption on the 2-MoG latent distribution.

**Assumption 6.1.** [Separation within a cluster] Within each cluster $k$, the two symmetric peaks are well separated in the sense that $\|s_t\mu_k^* - (-s_t\mu_k^*)\| \geq \Delta_{\text{intra}}$, for some $\Delta_{\text{intra}} \gg \gamma_t$. Consequently, if a sample $x$ is drawn from the "+" peak then its responsibility under the "−" peak satisfies

$$r_k^-(x) = \frac{\frac{1}{2}\mathcal{N}(x; -s_t\mu_k^*, \Sigma_k^*)}{\frac{1}{2}\mathcal{N}(x; s_t\mu_k^*, \Sigma_k^*) + \frac{1}{2}\mathcal{N}(x; -s_t\mu_k^*, \Sigma_k^*)} = O\big(e^{-\Delta_{\text{intra}}^2/(2\gamma_t^2)}\big) \ll 1,$$

and symmetrically $r_k^+(x) \ll 1$ when $x$ is drawn from the "−" peak.

The above assumption means that the separation of the two modals is sufficient. For each symmetric sub-peak, if the distance between them is relatively small, we can view them as having a mean of 0. Since they are the same distribution ($\mu = 0$ and $\Sigma = U_kU_k^\top + \gamma_t^2 I$), they are the same regardless of how they mix, which indicates that we can assume $r_k^+ \approx 1$ or $r_k^- \approx 1$. Moreover, in practice, if raw data do not exhibit such clear gaps, one can always apply a simple linear embedding to magnify inter-mean distances relative to noise, thereby enforcing the same hard-assignment regime.

Since the ground truth score function has a closed-form under the MoLR-MoG modeling, we focus on the score matching objective function $\mathcal{L}_{\text{SM}}(\theta)$ instead of $\mathcal{L}_{\text{DSM}}(\theta)$ and abbreviate $\mathcal{L}_{\text{SM}}(\theta)$ as $\mathcal{L}(\theta)$. We note that $\mathcal{L}_{\text{SM}}(\theta)$ and $\mathcal{L}_{\text{DSM}}(\theta)$ are equivalent up to a constant independent of $\theta$, which indicates the optimization landscape is the same. Furthermore, when considering the convergence guarantee under a 2-layer wide ReLU NN, Li et al. (2023) also adopt score matching objective $\mathcal{L}_{\text{SM}}$ instead of $\mathcal{L}_{\text{DSM}}$. Though calculating the bound of Jacobian $J_k^\mu(x) = \partial_{\mu_k}s_\theta$, $J_k^U(x)$ and the Hessian matrix w.r.t. $\mathcal{L}$, we provide the local strongly convexity parameters for the objective function.

**Lemma 6.2.** [Local Strong Convexity] Combining Lemma B.4 with continuity of $\nabla^2\mathcal{L}$, there exist $\alpha > 0$ and neighborhood $U$ of $\theta^*$ such that $\nabla^2\mathcal{L}(\theta) \succeq \alpha I, \forall\theta \in \Theta$. If $\forall x \in \mathbb{R}^{d_k}, r_k^+(x) = 1$ or $r_k^-(x) = 1$ are strictly satisfied,

$$\alpha = \min\left\{\frac{s_t^2}{(s_t^2+\gamma_t^2)^2}, \frac{4(U_k^\top\mu_k)^2 + \|U_k\|_2^2\|\mu_k\|_2^2 - \|U_k\|_2\|\mu_k\|_2\sqrt{8(U_k^\top\mu_k)^2 + \|U_k\|_2^2\|\mu_k\|_2^2}}{2}\right\}.$$

**Theorem 6.3.** [Local Linear Convergence] Under Assumptions 5.1 and 6.1, if we take $\eta_m = \eta = 2/(\eta + L')$, and $\kappa = L'/\alpha$, then there exists a neighborhood $U$ of $\theta^*$ such that

$$\|\theta^{(m)} - \theta^\star\|_2 \leq \left(\frac{\kappa - 1}{\kappa + 1}\right)^m \|\theta^{(0)} - \theta^\star\|_2,$$

where $m$ is the number of gradient descent iterations.

This result gives a lower bound on the convergence rate near $\theta^\star$. Due to its strongly convex property, the convergence rate is fast, which explains the fast and stable optimization process.

*Proof Overview.* Assumption 6.1 justifies the Jacobian simplification (Lemma B.2), which in turn yields the Hessian block structure (Lemma B.4). By Schur complement, this result gives local strong convexity (Lemma 6.2). Combining with the Lipschitz constant, we finish the proof.

## 6.2 GENERAL MoG LATENT HESSIAN ANALYSIS AND OPTIMIZATION

We now extend our analysis to the case where each subspace $k$ carries an *asymmetric* Gaussian mixture (Equation 3). As before, we first state the key separation assumption and show that on each subspace, the individual Gaussian distributions in the mixture of Gaussian are highly separated from each other. Then, we simplify the Hessian and prove local convexity. Finally, we conclude a linear convergence rate based on the strongly convex and smooth property.

**Assumption 6.4.** [Highly Separated Gaussian] Consider the Gaussian mixture

$$p_k(x) = \sum_{l=1}^{n_k} \pi_{k,l}\mathcal{N}(x; \mu_{k,l}, \Sigma_{k,l}), \qquad r_{k,l}(x) := \frac{\pi_{k,l}\mathcal{N}(x; \mu_{k,l}, \Sigma_{k,l})}{\sum_{i=1}^{n_k}\pi_{k,i}\mathcal{N}(x; \mu_{k,i}, \Sigma_{k,i})}.$$

There exist constants $\varepsilon \ll 1$ and $\delta \ll 1$ such that when $x \sim p_k$ we have

$$\Pr_{x \sim p_k} \left( \exists l \in \{1, \dots, n_k\} \text{ with } r_{k,l}(x) \geq 1 - \varepsilon \right) \geq 1 - \delta.$$

*Justification.* With MoLR-MoG modeling, after adding diffusion noise of scale $\gamma_t$, each point $x$ remains within $O(\gamma_t)$ of the subspace's moment-matched center $\bar{\mu}_k$. Concretely, the subspace structure (or a preliminary projection onto principal components) ensures $\|x - \bar{\mu}_k\|_2 \leq \Delta = C\gamma_t$ with high probability, for some moderate constant $C$. Hence, any third-order Taylor term $\propto \|x - \bar{\mu}_k\|^3$ is $O(\gamma_t^3)$, which vanishes compared to the leading Hessian scale $O(\gamma_t^2)$. In the following corollary, we further show the approximation effect of equivalent Gaussians.

**Corollary 6.5.** *Assume that* $\|\mu_{k,i}^* - \mu_{k,j}^*\|_2 \leq \delta$, $\|U_{k,i}^* - U_{k,j}^*\|_2 \leq \epsilon$ *and* $\|x - \bar{\mu}_k^*\|_2 \leq \Delta$. *We have*

$$\| \log p(x) - \log \bar{p}(x) \|_2 = O(\epsilon + \delta\Delta + \Delta^3)$$

*Remark* 6.6 (Separated Gaussian Simplification). For simplicity of description, we assume the individual Gaussian distributions in the mixture of Gaussians are highly separated. Actually, if there are $n_k'$ Gaussians that are not separated from each other, we can employ clustering techniques to transform them into $n_k$ mutually independent Gaussian distributions. The error caused by such an operation can be calculated using corollary 6.5. The core intuition is that the modals should not have much influence on each other. Hence, we can also use the idea of recursion to first cluster the general MoG into a 2-modal MoG latent. Then, we can use the analysis of Section 6.1 with Theorem 6.1.

Then, similar to the above section, we also calculate the Hessian matrix and show the local strong convex parameters. Finally, we provide the convergence guarantee for general MoLR-MoG modeling.

**Lemma 6.7.** *[Eigenvalues of the Hessian] Assume Theorem 6.4, the Hessian at the $k$-th subspace is convex on a neighborhood of $\theta^*$. If $\forall x \in \mathbb{R}^{d_k}$, $r_k^+(x) = 1$ or $-1$ are strictly satisfied, we have*

$$\lambda_{\min}(H_{\mu_{k,l}\mu_{k,l}}) = \frac{\pi_{k,l} s_t^2}{(s_t^2 + \gamma_t^2)^2},$$

*and* $\lambda_{\min}(H_{U_{k,l}U_{k,l}})$ *has the following form:*

$$\left( \pi_{k,l} 4(U_{k,l}^\top \mu_{k,l}))^2 + \|U_{k,l}\|_2^2 \|\mu_{k,l}\|_2^2 - \|U_{k,l}\|_2 \|\mu_{k,l}\|_2 \sqrt{8(U_{k,l}^\top \mu_{k,l}))^2 + \|U_{k,l}\|_2^2 \|\mu_{k,l}\|_2^2} \right) / 2.$$

**Lemma 6.8.** *[Local Strong Convexity] Assume Theorem 6.4, in a neighborhood of $\theta^*$, $\nabla^2 \mathcal{L}(\theta) \succeq \alpha' I, \alpha' > 0, \forall \theta \in \Theta$. If $\forall x \in \mathbb{R}^{d_k}$, $\exists l \in [n_k], r_{k,l}(x) = 1$ are strictly satisfied, $\alpha' = \min\{\lambda_1, \lambda_2\}$, where $\lambda_1 = \min_{l=1 \dots, n_k} \frac{c_{k,l} \gamma_t^4}{(s_t^2 + \gamma_t^2)^2}$, $\lambda_2 = \min_{l=1,2,\dots,n_k} = \lambda_{\min}(H_{U_{k,l}U_{k,l}})$.*

Thus, even without symmetry, equivalent Gaussians and sufficient subspace separation recover the same local convexity and linear convergence guarantees as in the asymmetric case. Similar to Theorem 6.3, under Theorem 6.4, we can obtain a convergence guarantee.

*Remark* 6.9 (Previous MoG Learning through Score Matching). Shah et al. (2023) and Chen et al. (2024b) consider MoG data and analyze the optimization process of diffusion models at the full space. However, these works aim to design a specific algorithm to learn the MoG distribution instead of using a standard optimization algorithm. On the contrary, by using the MoLR-MoG property to calculate the Hessian matrix, we adopt the GD algorithm and obtain the convergence guarantee.

*Remark* 6.10 (Initialization). Since the multi-modal GMM latent leads to a highly non-convex landscape, Theorem 6.3 and the corresponding asymmetric variant require the initialization to be around $\theta^*$ to guarantee local strong convexity and obtain a local convergence guarantee. As the MoLR-MoG is the first step to model the multi low-dimensional and multi-modal property, we leave the analysis of the global convergence guarantee as an interesting future work.

## 6.3 ANALYSIS WITHOUT HIGHLY SEPARATED CONDITION

In this part, we extend our analysis to latent MoG with overlap, which is closer to the real-world data. We define the pairwise overlap factor $\xi_{i,j}(x)$ between components $i$ and $j$ at the $k$-th manifold

$$\xi_{i,j}(x) \triangleq r_{k,i}(x) r_{k,j}(x).$$

and the maximum expected overlap for the manifold as: $\epsilon_{\text{overlap}} = \max_i \sum_{j \neq i} \mathbb{E}\_x \sim p_t[\xi_{i,j}(x)]$.

Without the high-separation assumption, our analysis proceeds in two steps. With the overlap factor $\epsilon_{\text{overlap}}$, we first examine the block-diagonal Hessian, deriving a refined lower bound $\alpha$. Second, we analyze the full Hessian by treating off-diagonal interference as a perturbation bounded by the overlap factor. Applying Weyl's Inequality, we prove that the global matrix remains positive definite provided the perturbation (introduced by the overlap) is smaller than the effective diagonal curvature $\alpha$, thus guaranteeing linear convergence.

**Lemma 6.11** (Minimum Curvature for 2-Mode Mixture). *Consider a mixture of two Gaussian components. Let $\epsilon_{\text{overlap}} = \sup_x r_k^+(x) r_k^-(x)$ denote the maximum pointwise overlap factor. The minimum eigenvalue of the ideal Hessian matrix, denoted as $\alpha_{2\text{-mode}}$, is bounded below by:*

$$\alpha_{2\text{-mode}} \triangleq (1 - 4\epsilon_{\text{overlap}}) \min\left(\lambda_{\min}(H_{\mu_k \mu_k}), \lambda_{\min}(H_{U_k U_k})\right),$$

*and*

$$\lambda_{\min}(H) \geq \alpha_{2\text{-mode}} - C' \epsilon_{\text{overlap}} > 0,$$

where $C'$ is defined in D.1.3.

**Lemma 6.12** (Minimum Curvature for Multi-Modal). *Let $\epsilon_{k,l}^{total} = \sum_{j \neq l} \mathbb{E}[\xi_{j,l}(x)]$ represent the total probability mass leaking from the $l$-th component due to overlap. The minimum eigenvalue of the block-diagonal Hessian, denoted as $\alpha_{Multi\text{-}Modal}$, is determined by the component with the minimum effective mass:*

$$\alpha_{Multi\text{-}Modal} \triangleq \min_{l \in \{1, \dots, n_k\}} \left[ (\pi_{k,l} - \epsilon_{k,l}^{total}) \min\left(\lambda_{\min}(H_{\mu_{k,l} \mu_{k,l}}), \lambda_{\min}(H_{U_{k,l} U_{k,l}})\right) \right],$$

*and*

$$\lambda_{\min}(H) \geq \alpha_{Multi\text{-}Modal} - \tilde{C} \cdot \epsilon_{overlap},$$

*where $\tilde{C}$ is defined in D.2.4. For the Hessian to remain positive definite, the intrinsic weight of every cluster must exceed its total confusion with other clusters (i.e., $\pi_{k,l} > \epsilon_{k,l}^{total}$ for all $l$).*

## 7 CONCLUSION

In this work, we provide a mixture of low-rank mixture of Gaussian (MoLR-MoG) modeling for target data, which reflects the low-dimensional and multi-modal property of real-world data. Through the real-world experiments, we first show that the MoLR-MoG is a suitable modeling for the real-world data. Then, we analyze the estimation error and optimization process under the MoLR-MoG modeling and explain why diffusion models can achieve great performance with a small training dataset and a fast optimization process.

For the estimation error, we show that with the MoLR-MoG modeling, the estimation error is $R^4 \sqrt{\Sigma_{k=1}^K n_k} \sqrt{\Sigma_{k=1}^K n_k d_k} / \sqrt{n}$, which means diffusion models can take fully use of the multi subspace, low-dimensional and multi-modal information to escape the curse of dimensionality. For the optimization process, we conducted a detailed analysis of the score-matching loss landscape. By formulating the exact score in both symmetric and asymmetric mixture settings, we derived explicit expressions for the parameter Jacobians and identified the dominant components under standard separation assumptions. Then, we prove that the population loss becomes strongly convex in a neighborhood of the ground truth score function, by estimating the Hessian and presenting lower bounds on both its minimal eigenvalue and the convergence rate. Then, we provide the local convergence guarantee for the score matching objective function, which explains the fast and stable training process of diffusion models.

**Future work and limitation.** Though we have extended the situation to multi-manifold MoG, how to extend the analysis to more general non-Gaussian sub-manifolds (e.g. heavy-tailed or multi-modal beyond second moments) by higher-order moment matching is still unknown. Meanwhile, we wish to design optimization algorithms or network architectures that explicitly leverage the block-diagonal Hessian structure for faster training. For example, we can perform a natural-gradient step separately in each block with a block-diagonal Hessian with decomposed data, which will accelerate the optimization process.

**Ethics statement.** Our work aims to deepen the understanding of the modeling of diffusion models and explain the success of diffusion models from a theoretical perspective. The MoLR-MoG modeling has the potential to achieve a great performance with fewer parameters. Hence, this work can be viewed as an important step in understanding diffusion models, and the societal impact is similar to general generative models (Mirsky and Lee, 2021).

**Reproducibility statement.** The detail and description of the real-world experiments are provided in Section E. We detail the model, hyperparameters and data.

**Acknowledgement.** The corresponding author Shuai Li is supported by National Natural Science Foundation of China (92570111) and Shanghai Key Laboratory of Intelligent Information Processing, Fudan University (Grant No. IIPL-2025-RD1-04).

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

APPENDIX

# A SCORE FUNCTION ERROR ESTIMATION

In this part, we analyze the estimation error of diffusion models under the MoLR-MoG modeling and show why models can escape the curse of dimensionality under this modeling. As a start, we first calculate the closed-form of score function under MoLR-MoG modeling.

## A.1 CALCULATE $\nabla \log p_t(x)$ AND DECOMPOSITION

Consider the $k$-th subspace

$$p_{t,k}(x) = \sum_{l=1}^{n_k} \pi_{k,l} \mathcal{N}\left(\mu_{k,l}, \Sigma_{k,l}\right)$$

where $\Sigma_{k,l} = s_t^2 U_{k,l} U_{k,l}^\top + \gamma_t^2 I$. We know that

$$\Sigma_{k,l}^{-1} = \frac{1}{\gamma_t^2}\left(I - \frac{s_t^2}{s_t^2+\gamma_t^2} U_{k,l} U_{k,l}^\top\right),$$

$$\nabla p_{t,k}(x) = \frac{1}{\gamma_t^2} \sum_{l=1}^{n_k} \pi_{k,l}\mathcal{N}(\mu_{k,l},\Sigma_{k,l})\left(I - \frac{s_t^2}{s_t^2+\gamma_t^2} U_{k,l} U_{k,l}^\top\right)(x-\mu_{k,l}),$$

which indicates

$$\nabla \log p_{t,k}(x) = \frac{\nabla p_{t,k}(x)}{p_{t,k}(x)} = \frac{1}{\gamma_t^2}\frac{\sum_{l=1}^{n_k} \pi_{k,l}\mathcal{N}(\mu_{k,l},\Sigma_{k,l})\left(I - \frac{s_t^2}{s_t^2+\gamma_t^2} U_{k,l} U_{k,l}^\top\right)(x-\mu_{k,l})}{\sum_{l=1}^{n_k} \pi_{k,l}\mathcal{N}(\mu_{k,l},\Sigma_{k,l})}.$$

Let

$$s^*(x,t) = \left(s_1^*(x,t), s_2^*(x,t), \ldots, s_K^*(x,t)\right).$$

We want to learn the parameters of the score function for each subspace:

$$s_k^*(x,t) = \nabla \log p_{t,k}(x),$$

where the parameters are $\{\mu_{k,l}^*, U_{k,l}^*\}, k = 1, ..., K$.

Define

$$\mathcal{L}(s_k) = \mathbb{E}\left[\|s_k(x,t) - s_k^*(x,t)\|^2\right], \quad \hat{\mathcal{L}}_n(s_k) = \frac{1}{n}\sum_{i=1}^{n}\|s_k(x_i,t_i) - s_k^*(x_i,t_i)\|^2.$$

We have the following decomposition:

$$\mathcal{L}(\hat{s}_{k,\hat{\theta}_n}) - \hat{\mathcal{L}}_n(s_{k,\hat{\theta}_n}) = \underbrace{\mathcal{L}(\hat{s}_{k,\hat{\theta}_n}) - \hat{\mathcal{L}}(s_k^*)}_{\text{Estimation}} + \underbrace{\hat{\mathcal{L}}(s_k^*) - \hat{\mathcal{L}}(s_{k,\theta^*})}_{\text{Approximation}} + \underbrace{\hat{\mathcal{L}}_n(s_{k,\theta^*}) - \hat{\mathcal{L}}_n(\hat{s}_{k,\hat{\theta}_n})}_{\text{optimization}}.$$

We can also obtain that

$$\mathcal{L}(s) = \sum_{k=1}^{K} \mathcal{L}(s_k).$$

Since *Estimation* and *Approximation* reflect the fitting ability of the network, we analyze the first term first. Then, in the next section, we analyze the optimization dynamic.

## A.2 ESTIMATION ERROR ANALYSIS

First, we show that $f$ and loss function are Lipschitz. We will first prove that $s_k$ is Lipschitz for $\forall k$, then we can know that $s$ is Lipschitz.

**Lemma A.1.** *[Lipschitz Continuity] Let $L_{\mu_l}$ and $L_{U_k}$ be the Lipschitz constant w.r.t. $s_\theta$. With MoLR-MoG modeling and Theorem 5.1, there is a constant*

$$L \leq \sqrt{\Sigma_{i=1}^K n_k (L_{\mu_l}^2 + L_{U_k}^2)} = O\left((\Sigma_{k=1}^K n_k)^{\frac{1}{2}} C_w\right)$$

*such that for any $\theta, \theta'$, $\left\| s_\theta(x,t) - s_{\theta'}(x,t) \right\|_2 \leq L \left\| \theta - \theta' \right\|_2$, where $C_w = \frac{(R + s_t B_\mu)^3 s_t^2}{\gamma_t^4}, B_\mu = \max_{k,l} \|\mu_{k,l}\|_2$. For $s_\theta$ and $s^*$, we have that $2\|s_\theta(x,t) - s^*(x,t)\|_2 \leq 2(R + s_t B_\mu)/\gamma_t^2 := L_l$.*

**Proof.** Since we analyze the estimation error at a fixed time $t$, we ignore subscript $t$ for $\Sigma_{k,l,t}$, $w_{k,t}$, $w_{l,k,t}$ and $\delta_{k,l,t}$ and define by

$$\Sigma_{k,l} = s_t^2 U_{k,l} U_{k,l}^\top + \gamma_t^2 I \,,$$
$$w_k(x) = \Sigma_{l=1}^{n_k} \pi_{k,l} \mathcal{N}(x; s_t \mu_{k,l}, \Sigma_{k,l}) \,,$$
$$w_{k,l} = \frac{1}{K} \pi_{k,l} \mathcal{N}(x; s_t \mu_{k,l}, \Sigma_{k,l}) \,,$$
$$\delta_{k,l}(x) = x + s_t \mu_{k,l} - \frac{s_t^2}{s_t^2 + \gamma_t^2} U_{k,l} U_{k,l}^\top (x + s_t \mu_{k,l}) \,.$$

Assume that $\|U_{k,l}\|_2 \leq B_U, \|\mu_{k,l}\|_2 \leq B_\mu, \max\{B_U, B_\mu\} = C$, and $\|x\|_2 \leq R$ for $\forall x \in X$. For $\Sigma_{k,l}$, we know that

$$\Sigma_{k,l} = U_{k,l} U_{k,l}^\top + \gamma_t^2 I \succ \gamma_t^2 I \Rightarrow \lambda_{\min}(\Sigma_{k,l}) \geq \gamma_t^2 \Rightarrow \|\Sigma_{k,l}^{-1}\|_2 \leq \frac{1}{\gamma_t^2} \,.$$

To obtain the first $L$ in this lemma, we need to bound $\left\| \frac{\partial s_{k,\theta}(x,t)}{\partial \mu_{k,l}} \right\|_2$ and $\left\| \frac{\partial s_{k,\theta}(x,t)}{\partial U_{k,l}} \right\|_2$.

**The bound of $\left\| \frac{\partial s_{k,\theta}(x,t)}{\partial \mu_{k,l}} \right\|_2$.** For the latent score of the $k$-th subspace, we have that

$$s_{k,\theta}(x,t) = -\frac{1}{\gamma_t^2} \frac{\Sigma_{l=1}^{n_k} w_{k,l}(x) \delta_{k,l}(x)}{w_k(x)} \,,$$

$$\frac{\partial s_{k,\theta}(x,t)}{\partial \mu_{k,l}} = -\frac{1}{\gamma_t^2} \frac{\Sigma_{l=1}^{n_k} (\frac{\partial w_{k,l}(x)}{\partial \mu_{k,l}} \delta_{k,l}(x) + \frac{\partial \delta_{k,l}(x)}{\partial \mu_{k,l}} w_{k,l}(x)) w_k(x) - \frac{\partial w_k(x)}{\partial \mu_{k,l}} (\Sigma_{l=1}^{n_k} w_{k,l}(x) \delta_{k,l}(x))}{w_k^2(x)} \,,$$

$$\left\| \frac{\partial s_{k,\theta}(x,t)}{\partial \mu_{k,l}} \right\|_2 \leq \frac{1}{\gamma_t^2} \left( \left\| \frac{\Sigma_{l=1}^{n_k} (\frac{\partial w_{k,l}(x)}{\partial \mu_{k,l}} \delta_{k,l}(x) + \frac{\partial \delta_{k,l}(x)}{\partial \mu_{k,l}} w_{k,l}(x))}{w_k(x)} \right\|_2 + \left\| \frac{\frac{\partial w_k(x)}{\partial \mu_{k,l}} (\Sigma_{l=1}^{n_k} w_{k,l}(x) \delta_{k,l}(x))}{w_k^2(x)} \right\|_2 \right) \,.$$

To bound this term, we separately show that

$(1) w_k(x)$ has a lower bound.

$(2) w_{k,l}(x), \delta_{k,l}(x), \dfrac{\partial w_{k,l}(x)}{\partial \mu_k}, \dfrac{\partial \delta_{k,l}(x)}{\partial \mu_k}$ have upper bounds.

$(3) \left\| \dfrac{\frac{\partial w_k(x)}{\partial \mu_{k,l}} \delta_{k,l}(x)}{w_k} \right\|_2, \left\| \dfrac{\Sigma_{l=1}^{n_k} \frac{\partial \delta_{k,l}(x)}{\partial \mu_{k,l}} w_{k,l}(x)}{w_k(x)} \right\|_2, \left\| \dfrac{\frac{\partial w_k(x)}{\partial \mu_{k,l}} \Sigma_{l=1}^{n_k} w_{k,l}(x) \delta_{k,l}(x)}{w_k^2(x)} \right\|_2$ have upper bounds.

$(1)$ $w_k(x)$ has a lower bound.

$$w_k(x) = \Sigma_{l=1}^{n_k} \pi_{k,l} \mathcal{N}(x; s_t \mu_{k,l}, \Sigma_{k,l}), \text{which is continuous.}$$

Since continuous function has maximum and minimum in a closed internal and $\|x\|_2 \leq R$, we can assume that $w_k(x) \geq m_w$. And for any $x$, $w_k(x) > 0$, so $m_w > 0$ holds.

$(2)$ $w_{k,l}(x), \delta_{k,l}(x), \frac{\partial \delta_{k,l}(x)}{\partial \mu_k}, \frac{\partial w_{k,l}(x)}{\partial \mu_k}$ have upper bounds.

We already know that continuous function has maximum and minimum in a closed internal and $\|x\|_2 \le R$. Thus, we can assume that $w_k(x) \le M_{w_k}$. We also have that

$$w_k(x) \le M_{w_k} \le \Sigma_{l=1}^{n_k} \pi_{k,l} (2\pi)^{-\frac{n}{2}} |\Sigma_{k,l}|^{-\frac{1}{2}}.$$

For the second term, we have that

$$\delta_{k,l}(x) = x - s_t \mu_{k,l} - \frac{s_t^2}{s_t^2 + \gamma_t^2} U_{k,l} U_{k,l}^\top (x - s_t \mu_{k,l}) = \left( I - \frac{s_t^2}{s_t^2 + \gamma_t^2} U_{k,l} U_{k,l}^\top \right)(x - s_t \mu_{k,l}),$$

whose $L_2$ norm is bounded by

$$\left\| \left( I - \frac{s_t^2}{s_t^2 + \gamma_t^2} U_{k,l} U_{k,l}^\top \right)(x - s_t \mu_{k,l}) \right\|_2 \le \|x - s_t \mu_{k,l}\|_2 \le \|x\|_2 + \|s_t \mu_{k,l}\|_2 \le R + s_t B_\mu.$$

Then, for the third term, we know that

$$\frac{\partial \delta_{k,l}(x)}{\partial \mu_{k,l}} = -s_t + \frac{s_t^3}{s_t^2 + \gamma_t^2} U_{k,l} U_{k,l}^\top = -s_t \left( I - \frac{s_t^2}{s_t^2 + \gamma_t^2} U_{k,l} U_{k,l}^\top \right).$$

For the last term, we have we have the following expression

$$\frac{\partial w_{k,l}(x)}{\partial \mu_{k,l}} = -\frac{s_t}{2} \mathcal{N}(x; s_t \mu_{k,l}, \Sigma_{k,l}) \Sigma_{k,l}^{-1} (x - s_t \mu_{k,l}).$$

For term $\|\Sigma_{k,l}^{-1}(x - s_t \mu_{k,l})\|_2$, we have that

$$\|\Sigma_{k,l}^{-1}(x - s_t \mu_{k,l})\|_2 \le \|\Sigma_{k,l}^{-1}\|_2 \|x - s_t \mu_{k,l}\|_2 = \frac{1}{\gamma_t^2} \|x - s_t \mu_{k,l}\|_2 \le \frac{1}{\gamma_t^2}(R + \|s_t \mu_{k,l}\|_2),$$

which indicates

$$\left\| \frac{\partial w_{k,l}(x)}{\partial \mu_{k,l}} \right\|_2 \le s_t \mathcal{N}(x; s_t \mu_{k,l}, \Sigma_{k,l}) \frac{1}{\gamma_t^2}(R + \|s_t \mu_{k,l}\|_2) \le s_t \mathcal{N}(x; s_t \mu_{k,l}, \Sigma_{k,l}) \frac{1}{\gamma_t^2}(R + s_t B_\mu)$$

$$\left\| \frac{\partial w_k(x)}{\partial \mu_{k,l}} \right\|_2 \le \Sigma_{l=1}^{n_k} s_t \mathcal{N}(x; s_t \mu_{k,l}, \Sigma_{k,l}) \frac{1}{\gamma_t^2}(R + s_t B_\mu).$$

(3) $\left\| \frac{\frac{\partial w_k(x)}{\partial \mu_{k,l}} \delta_{k,l}(x)}{w_k} \right\|_2$, $\left\| \frac{\Sigma_{l=1}^{n_k} \frac{\partial \delta_{k,l}(x)}{\partial \mu_{k,l}} w_{k,l}(x)}{w_k(x)} \right\|_2$, $\left\| \frac{\frac{\partial w_k(x)}{\partial \mu_{k,l}} \Sigma_{l=1}^{n_k} w_{k,l}(x) \delta_{k,l}(x)}{w_k^2(x)} \right\|_2$ have upper bounds.

For the first two term,

$$\left\| \frac{\frac{\partial w_k(x)}{\partial \mu_{k,l}} \delta_{k,l}(x)}{w_k} \right\|_2 \le \frac{s_t}{\gamma_t^2}(R + s_t B_\mu)^2,$$

and

$$\left\| \frac{\partial \delta_{k,l}(x)}{\partial \mu_{k,l}} \right\|_2 = \text{Constant} \le s_t, \quad \left\| \frac{\Sigma_{l=1}^{n_k} \frac{\partial \delta_{k,l}(x)}{\partial \mu_{k,l}} w_{k,l}(x)}{w_k(x)} \right\|_2 \le s_t.$$

For the third term, we know that

$$\left\| \frac{\frac{\partial w_k(x)}{\partial \mu_{k,l}} \Sigma_{l=1}^{n_k} w_{k,l}(x) \delta_{k,l}(x)}{w_k^2(x)} \right\|_2 \le \left\| \frac{s_t w_k^2(x) \frac{s_t}{\gamma_t^2}(R + s_t B_\mu)}{w_k^2(x)} \right\|_2 = \frac{s_t^2}{\gamma_t^2}(R + s_t B_\mu).$$

Combined with the above three, we obtain the bound for $\left\| \frac{\partial s_{k,\theta}(x,t)}{\partial \mu_{k,l}} \right\|_2$:

$$\left\| \frac{\partial s_{k,\theta}(x,t)}{\partial \mu_{k,l}} \right\|_2 \le \frac{1}{\gamma_t^2} \left( \left\| \frac{\Sigma_{l=1}^{n_k} (\frac{\partial w_{k,l}(x)}{\partial \mu_{k,l}} + \frac{\partial \delta_{k,l}(x)}{\partial \mu_{k,l}}) \delta_{k,l}(x)}{w_k(x)} \right\|_2 + \left\| \frac{\frac{\partial w_k(x)}{\partial \mu_{k,l}}(\Sigma_{l=1}^{n_k} w_{k,l}(x) \delta_{k,l}(x))}{w_k^2(x)} \right\|_2 \right)$$

$$\le \frac{s_t^2}{\gamma_t^2}(R + s_t B_\mu)^2 + s_t + \frac{s_t}{\gamma_t^2}(R + s_t B_\mu) = O\left( \frac{s_t^2 (R + s_t B_\mu)^2}{\gamma_t^2} \right).$$

**The bound of** $\left\|\frac{\partial s_{k,\theta}(x,t)}{\partial U_{k,l}}\right\|_2$**.** Now we compute the part about $U_{k,l}$. Through some simple algebra, we know that

$$\frac{\partial s_{k,\theta}(x,t)}{\partial U_{k,l}} = -\frac{1}{\gamma_t^2}\frac{\Sigma_{l=1}^{n_k}(\frac{\partial w_{k,l}(x)}{\partial U_{k,l}}\delta_{k,l}(x) + \frac{\partial \delta_{k,l}(x)}{\partial U_{k,l}}w_{k,l}(x))w_k(x) - \frac{\partial w_k(x)}{\partial U_{k,l}}(\Sigma_{l=1}^{n_k}w_{k,l}(x)\delta_{k,l}(x))}{w_k^2(x)}\,.$$

Then, we have the following inequality

$$\frac{\partial s_{k,\theta}(x,t)}{\partial U_{k,l}} = -\frac{1}{\gamma_t^2}\frac{\Sigma_{l=1}^{n_k}(\frac{\partial w_{k,l}(x)}{\partial U_{k,l}}\delta_{k,l}(x) + \frac{\partial \delta_{k,l}(x)}{\partial U_{k,l}}w_{k,l}(x))w_k(x) - \frac{\partial w_k(x)}{\partial U_{k,l}}(\Sigma_{l=1}^{n_k}w_{k,l}(x)\delta_{k,l}(x))}{w_k^2(x)}$$

$$\left\|\frac{\partial s_{k,\theta}(x,t)}{\partial U_{k,l}}\right\|_2 \leq \frac{1}{\gamma_t^2}\left(\left\|\frac{\Sigma_{l=1}^{n_k}(\frac{\partial w_{k,l}(x)}{\partial U_{k,l}}\delta_{k,l}(x) + \frac{\partial \delta_{k,l}(x)}{\partial U_{k,l}}w_{k,l}(x))}{w_k(x)}\right\|_2 + \left\|\frac{\frac{\partial w_k(x)}{\partial U_{k,l}}(\Sigma_{l=1}^{n_k}w_{k,l}(x)\delta_{k,l}(x))}{w_k^2(x)}\right\|_2\right)\,.$$

Similar with $\left\|\frac{\partial s_{k,\theta}(x,t)}{\partial \mu_{k,l}}\right\|_2$, we need to provide:

(1) The upper bound of $\frac{\partial w_{k,l}}{\partial U_{k,l}}$ and $\frac{\partial \delta_{k,l}}{\partial U_{k,l}}$,

(2) The upper bound of $\left\|\frac{\Sigma_{l=1}^{n_k}(\frac{\partial w_{k,l}(x)}{\partial U_{k,l}}\delta_{k,l}(x) + \frac{\partial \delta_{k,l}(x)}{\partial U_{k,l}}w_{k,l}(x))}{w_k(x)}\right\|_2$ and $\left\|\frac{\frac{\partial w_k(x)}{\partial U_{k,l}}*(\Sigma_{l=1}^{n_k}w_{k,l}(x)\delta_{k,l}(x))}{w_k^2(x)}\right\|_2$.

(1) The upper bound of $\frac{\partial w_{k,l}}{\partial U_{k,l}}$ and $\frac{\partial \delta_{k,l}}{\partial U_{k,l}}$.

For the first term, we have the following form

$$\frac{\partial w_{k,l}}{\partial U_{k,l}} = \pi_{k,l}\frac{\partial \mathcal{N}(x; s_t\mu_{k,l}, \Sigma_{k,l})}{\partial U_k}$$

$$= 2\pi_{k,l}s_t^2[\mathcal{N}(x; s_t\mu_{k,l}, \Sigma_{k,l})(\Sigma_k^{l^{-1}}(x - s_t\mu_{k,l})(x - s_t\mu_{k,l})^\top\Sigma_{k,l}^{-1} - \Sigma_{k,l}^{-1})]U_{k,l}\,.$$

Then, we know that

$$\left\|\frac{\partial w_{k,l}}{\partial U_{k,l}}\right\|_2 \leq 2\pi_{k,l}\mathcal{N}(x; s_t\mu_{k,l}, \Sigma_{k,l})s_t^2(\frac{(R + s_t\|\mu_{k,l}\|_2)^2}{\gamma_t^4} + \frac{1}{\gamma_t^2})$$

$$\leq 2\pi_{k,l}\mathcal{N}(x; s_t\mu_{k,l}, \Sigma_{k,l})s_t^2(\frac{(R + s_tB_\mu)^2}{\gamma_t^4} + \frac{1}{\gamma_t^2})\,.$$

For the second term, we have that

$$\frac{\partial \delta_{k,l}(x)}{\partial U_{k,l}} = -2\frac{s_t^2}{s_t^2 + \gamma_t^2}(U_{k,l}^\top(x - s_t\mu_{k,l})I + U_{k,l}(x - s_t\mu_{k,l})^\top)\,,$$

which indicates

$$\left\|\frac{\partial \delta_{k,l}(x)}{\partial U_{k,l}}\right\|_2 \leq 2\frac{s_t^2}{s_t^2 + \gamma_t^2}(R + \|s_t\mu_{k,l}\|_2) \leq 2(R + \|s_t\mu_{k,l}\|_2)$$

$$\leq 2(R + s_tB_\mu)\,.$$

(2) The upper bound of $\left\|\frac{\Sigma_{l=1}^{n_k}(\frac{\partial w_{k,l}(x)}{\partial U_{k,l}}\delta_{k,l}(x) + \frac{\partial \delta_{k,l}(x)}{\partial U_{k,l}}w_{k,l}(x))}{w_k(x)}\right\|_2$ and $\left\|\frac{\frac{\partial w_k(x)}{\partial U_{k,l}}*(\Sigma_{l=1}^{n_k}w_{k,l}(x)\delta_{k,l}(x))}{w_j^2(x)}\right\|_2$.

$$\left\|\frac{\Sigma_{l=1}^{n_k}(\frac{\partial w_{k,l}(x)}{\partial U_{k,l}}\delta_{k,l}(x) + \frac{\partial \delta_{k,l}(x)}{\partial U_{k,l}}w_{k,l}(x))}{w_k(x)}\right\|_2 \leq s_t^2(\frac{(R + s_tB_\mu)^3}{\gamma_t^4} + \frac{1}{\gamma_t^2}) + 2(R + s_tB_\mu)$$

We also have

$$\left\|\frac{\frac{\partial w_k(x)}{\partial U_{k,l}}(\Sigma_{l=1}^{n_k}w_{k,l}(x)\delta_{k,l}(x))}{w_k^2(x)}\right\|_2 \leq s_t^2(\frac{(R + s_tB_\mu)^2}{\gamma_t^4} + \frac{1}{\gamma_t^2})(R + s_tB_\mu)$$

$$\left\|\frac{\partial s_{k,\theta}(x,t)}{\partial U_{k,l}}\right\|_2 \leq s_t^2\left(\frac{(R+s_tB_\mu)^2}{\gamma_t^4}+\frac{1}{\gamma_t^2}\right)+2(R+s_tB_\mu)+s_t^2\left(\frac{(R+s_tB_\mu)^2}{\gamma_t^4}+\frac{1}{\gamma_t^2}\right)(R+s_tB_\mu)$$

$$= O\left(\frac{(R+s_tB_\mu)^3 s_t^2}{\gamma_t^4}\right).$$

Therefore, $s_{\theta,k}$ is $L_k$-lipshiz, where

$$L_k \leq \sqrt{n_k(L_{\mu_{k,l}}^2 + L_{U_{k,l}}^2)} = O\left(n_k^{\frac{1}{2}}\frac{(R+s_tB_\mu)^3 s_t^2}{\gamma_t^4}\right).$$

Furthermore, we know that

$$\|s_\theta(x)-s_\theta(y)\|_2 = \left(\sum_{i=1}^K\left\|s_{\theta,i}(x^{(i)})-s_{\theta,i}(y^{(i)}))\right\|^2\right)^{\frac{1}{2}} \leq (\sum_{i=1}^K L_i\|(x^{(i)}-y^{(i)}\|_2^2)^{\frac{1}{2}} \leq \sqrt{\sum_{i=1}^k L_i^2}\|x-y\|_2.$$

Thus,

$$L = \sqrt{\sum_{i=1}^k L_i^2} = O\left(\sqrt{\sum_{i=1}^k n_i^{\frac{1}{2}}\frac{(R+s_tB_\mu)^3 s_t^2}{\gamma_t^4}}\right).$$

After obtaining the Lipschitz constant for $s_\theta$, we bound the gap between $s_\theta$ and $s^*$:

$$\nabla\log p_{t,k}(x) = -\frac{1}{\gamma_t^2}\frac{\Sigma_{l=1}^{n_k}\pi_{k,l}\mathcal{N}(x;s_t\mu_{k,l},s_t^2 U_{k,l}^\star U_{k,l}^{\star\top}+\gamma_t^2 I)\left(x-s_t\mu_{k,l}-\frac{s_t^2}{s_t^2+\gamma_t^2}U_{k,l}^\star U_{k,l}^{\star\top}(x-s_t\mu_{k,l})\right)}{\Sigma_{l=1}^{n_k}\pi_{k,l}\mathcal{N}(x;s_t\mu_{k,l},s_t^2 U_{k,l}^\star U_{k,l}^{\star\top}+\gamma_t^2 I)}.$$

With the following bound

$$\left\|x-s_t\mu_{k,l}-\frac{s_t^2}{s_t^2+\gamma_t^2}U_{k,l}^\star U_{k,l}^{\star\top}(x-s_t\mu_{k,l})\right\|_2 \leq R+s_tB_\mu,$$

we have that

$$\|\nabla\log p_{t,k}(x)\|_2 \leq \frac{1}{\gamma_t^2}(R+s_tB_\mu)\,,\text{ and }\|s_{k,\theta}(x)\|_2 \leq \frac{1}{\gamma_t^2}(R+s_tB_\mu)\,,$$

which indicates

$$\|s_{k,\theta}(x)-\nabla\log p_{t,k}(x)\|_2 \leq \frac{2}{\gamma_t^2}(R+s_tB_\mu)\,.$$

Hence, we obtain that

$$L_l \leq 2\|s_{k,\theta}(x)-\nabla\log p_{t,k}(x)\|_2 = O(R+s_tB_\mu)\,.$$

∎

**Lemma A.2.** *[Rademacher Complexity] Let $\mathcal{F} = \{\ell(\theta;\cdot,\cdot)\colon \theta\in\Theta\}$ and suppose $\Theta$ has diameter $R_\Theta$. Then the empirical Rademacher complexity satisfies*

$$\widehat{\mathcal{R}}_n(\mathcal{F}) = O\left(L'\sqrt{\frac{p}{n}}\right).$$

**Proof.** Let function class be $\mathcal{F} = \{s_\theta(x) : \theta = (\{\{\mu_{k,l}, U_{k,l}\}_{l=1}^{n_k}\}_{k=1}^K) \in \Theta\}$ with $\mu_{k,l} \in \mathbb{R}^d, U_{k,l} \in \mathbb{R}^d$. We know that the number of parameters

$$p = \Sigma_{k=1}^K n_k(d+d) = 2\Sigma_{k=1}^K n_k d_k.$$

And the covering number of the parameter space is

$$\mathcal{N}(\epsilon,\Theta,\|\cdot\|_2) \leq \left(\frac{C}{\epsilon}\right)^p$$

If $f$ is $L$-lipschitz, we know that

$$\forall \theta_1, \theta_2 \in \Theta, \|f_{\theta_1} - f_{\theta_2}\|_{L_2(p)} \leq L\|\theta_1 - \theta_2\|_2 \qquad and \qquad \forall \theta, \ \exists \theta_j, \ s.t. \|\theta - \theta_j\|_2 \leq \frac{\epsilon}{L}$$

$$\Rightarrow \|f_\theta - f_{\theta_j}\|_{L_2(p)} \leq L\|\theta - \theta_j\|_2 \leq \epsilon.$$

Thus, assume that $\|\theta_i - \theta_j\|_2 \leq C_1$ for any $\theta_i, \theta_j \in \Theta$

$$\mathcal{N}(\epsilon, \Theta, \|\cdot\|_2) \leq \left(\frac{C_1}{\epsilon}\right)^p$$

$$\Rightarrow \mathcal{N}\left(\frac{\epsilon}{L}, \Theta, \|\cdot\|_2\right) \leq \left(\frac{C_1 L}{\epsilon}\right)^p$$

$$\Rightarrow \mathcal{N}(\epsilon, \mathcal{F}, \|\cdot\|_{L_2(p)}) \leq \mathcal{N}\left(\frac{\epsilon}{L}, \Theta, \|\cdot\|_2\right) \leq \left(\frac{C_1 L}{\epsilon}\right)^p.$$

We also know that $\text{diam}(\mathcal{F}) \leq L \, \text{diam}(\Theta) = C_1 L$, with Dudley integral, we have

$$\mathcal{R}_n(\mathcal{F}) \leq \frac{12}{\sqrt{n}} \int_0^{\text{diam}(\mathcal{F})} \sqrt{\log N(\epsilon, \mathcal{F}, \|\cdot\|_{L_2(p)})} d\epsilon$$

$$\leq \frac{12}{\sqrt{n}} \int_0^{C_1 L} \sqrt{p \log(\frac{C_1 L}{\epsilon})} d\epsilon$$

$$\leq \frac{12}{\sqrt{n}} \int_0^\infty pCL\sqrt{t} \exp(-t) dt = \frac{6\sqrt{\pi p}}{\sqrt{n}} C_1 L = O(C_1 L \sqrt{\frac{p}{n}}).$$

We take the squared loss function.

$$\mathcal{R}_n(\mathcal{L}) \leq L_l \mathcal{R}_n(\mathcal{F}) = O\left(C_1 L_l L \sqrt{\frac{p}{n}}\right).$$

∎

**Theorem 5.3.** *Denote by $\widehat{\mathcal{L}}_n(\theta)$ the empirical loss on $n$ i.i.d. samples and by $\mathcal{L}(\theta)$ its population counterpart. Then there exist constants $C_1, C_2$ such that with probability at least $1 - \delta$, for all $\theta \in \Theta$,*

$$\left|\mathcal{L}(\theta) - \widehat{\mathcal{L}}_n(\theta)\right| \leq O\left(C_1 \frac{(R + s_t B_\mu)^4 s_t^2 \sqrt{\Sigma_{k=1}^K n_k}}{\gamma_t^6} \sqrt{\frac{\Sigma_{k=1}^K n_k d_k}{n}} + C_2 \sqrt{\frac{\log(1/\delta)}{n}}\right).$$

*where $C_1 = \max_{\theta \in \Theta} \|\theta_i - \theta_j\|_2$, $C_2 = \sigma \log 2$, $\sigma^2 = \sup_{\theta \in \Theta} Var[\ell(\theta; X, t)]$.*

**Proof.** Using Lemma A.2, since

$$L_l \mathcal{R}_n(\mathcal{F}) = O(C_1 L_l L \sqrt{\frac{p}{n}}).$$

We have

$$\Delta = \sup_{\theta \in \Theta} |\hat{L}(\theta) - L(\theta)| = O(C_1 L_l L \sqrt{\frac{p}{n}})$$

Thus, by taking the expectation on both sides, we have:

$$\mathbb{E}[\Delta] = O(C_1 L_l L \sqrt{\frac{p}{n}}).$$

By Bernstein inequality,let $\sigma^2 = \sup_{\theta \in \Theta} Var[l(X; \theta)]$,we know that

$$Pr(\sup_{\theta \in \Theta} |\hat{L}(\theta) - L(\theta)| \geq \mathbb{E}[\Delta] + \epsilon) \leq 2\exp(-\frac{n\epsilon^2}{2(\sigma^2 + L_l L C_1 \epsilon/3)}) \leq 2\exp(-\frac{n\epsilon^2}{3\sigma^2}).$$

Let $2\exp(-\frac{n\epsilon^2}{3\sigma^2}) < \delta$, we can obtain that

$$Pr(\sup_{\theta \in \Theta} |\hat{L}(\theta) - L(\theta)| \geq C_1 L L_l \sqrt{\frac{p}{n}} + C_2 \sqrt{\frac{\log(1/\delta)}{n}}) \leq \delta.$$

∎

### A.3 APPROXIMATION

Since our network can represent $\nabla \log p(x)$ strictly, we have

$$\text{Approximation Error} = 0$$

## B  2-MODE MoG OPTIMIZATION

In this section, we analyze the optimization process when the latent distribution at each subspace is 2-mode MoG. In the next part, we prove the extension to multi-mode MoG and show how to remove the highly separated Gassuian assumption.

$$\nabla \log p_{t,k}(x) = \frac{\nabla p_{t,k}(x)}{p_{t,k}(x)} = -\frac{1}{\gamma_t^2} \frac{\frac{1}{2}\mathcal{N}(x;s_t\mu_k,s_t^2 U_k^\star U_k^{\star\top}+\gamma_t^2 I)\left(x-s_t\mu_k-\frac{s_t^2}{s_t^2+\gamma_t^2}U_k^\star U_k^{\star\top}(x-s_t\mu_k)\right)}{\frac{1}{2}\mathcal{N}(x;s_t\mu_k,s_t^2 U_k^\star U_k^{\star\top}+\gamma_t^2 I)+\frac{1}{2}\mathcal{N}(x;-s_t\mu_k,s_t^2 U_k^\star U_k^{\star\top}+\gamma_t^2 I)},$$

which can be reduced to

$$\nabla \log p_{t,k}(x) = -\frac{1}{\gamma_t^2} \frac{\frac{1}{2}\mathcal{N}(x;s_t\mu_k,\Sigma_k)\,\delta_k'(x)+\frac{1}{2}\mathcal{N}(x;-s_t\mu_k,\Sigma_k)\,\epsilon_k(x)}{\frac{1}{2}\mathcal{N}(x;s_t\mu_k,\Sigma_k)+\frac{1}{2}(x;-s_t\mu_k,\Sigma_k)}, \tag{5}$$

where $\epsilon_k(x) = x-s_t\mu_k-\frac{s_t^2}{s_t^2+\gamma_t^2}U_k^* U_k^{*\top}(x-s_t\mu_k)$, and $\delta_k'(x) = x+s_t\mu_k-\frac{s_t^2}{s_t^2+\gamma_t^2}U_k^* U_k^{*\top}(x+s_t\mu_k)$.

### B.1 OPTIMIZATION

**Assumption B.1.** [Separation within a cluster] Within each cluster $k$, the two symmetric peaks are well separated in the sense that $\|s_t\mu_k^* - (-s_t\mu_k^*)\| \geq \Delta_{\text{intra}}$, for some $\Delta_{\text{intra}} \gg \gamma_t$. Consequently, if a sample $x$ is drawn from the "+" peak then its responsibility under the "−" peak satisfies

$$r_k^-(x) = \frac{\frac{1}{2}\mathcal{N}(x;-s_t\mu_k^*,\Sigma_k^*)}{\frac{1}{2}\mathcal{N}(x;s_t\mu_k^*,\Sigma_k^*)+\frac{1}{2}\mathcal{N}(x;-s_t\mu_k^*,\Sigma_k^*)} = O\!\left(e^{-\Delta_{\text{intra}}^2/(2\gamma_t^2)}\right) \ll 1,$$

and symmetrically $r_k^+(x) \ll 1$ when $x$ is drawn from the "−" peak.

In the following discussion, we assume that $x \in k$-th manifold, which means that $w_i(x) = 0$ if $i \neq k$.

**Lemma B.2.** [Jacobian Simplification] Under Assumption 6.1, in a neighborhood of $\theta^*$ the first derivatives simplify to their "self-cluster" terms: $J_k^\mu(x) = \partial_{\mu_k}s_\theta \approx s_t(I - \alpha P_k)/\gamma_t^2$, and

$$J_k^U(x) \approx \frac{2s_t^2}{\gamma_t^2(s_t^2+\gamma_t^2)}\left(r_k^-(x)(U_k^\top(x+s_t\mu_k)I+(x+s_t\mu_k)U_k^\top)+r_k^+(x)(U_k^\top(x-s_t\mu_k)I+U_k(x-s_t\mu_k)^\top)\right).$$

**Proof.**

$J_k^\mu$

$$= -\frac{1}{\gamma_t^2} \frac{\left(\frac{\partial w_k^-(x)}{\partial \mu_k}\delta_k'(x)+\frac{\partial w_k^+(x)}{\partial \mu_k}\epsilon_k(x)+\frac{\partial \delta_k'(x)}{\partial \mu_k}w_k^-(x)+\frac{\partial \epsilon_k(x)}{\partial \mu_k}w_k^+(x)\right)\Sigma_{k=1}^K w_k(x)-\Sigma_{k=1}^K \frac{\partial w_k(x)}{\partial \mu_k}\Sigma_{k=1}^K(w_k^-(x)\delta_k'(x)+w_k^+(x)\epsilon_k(x))}{w_k^2(x))}$$

$$= \underbrace{\frac{w_k^-(x)\frac{\partial \delta_k'(x)}{\partial \mu_k}+w_k^+(x)\frac{\partial \epsilon_k(x)}{\partial \mu_k}}{\gamma_t^2 w_k(x)} - \frac{\frac{\partial w_k^-(x)}{\partial \mu_k}\delta_k'(x)+\frac{\partial w_k^+(x)}{\partial \mu_k}\epsilon_k(x)}{\gamma_t^2 w_k(x)}}_{\text{Term } A}$$

$$+ \underbrace{\frac{\frac{\partial w_k(x)}{\partial \mu_k}(w_k^-(x)\delta_k'(x)+w_k^+\epsilon_k(x))}{\gamma_t^2 w_k^2(x)}}_{\text{Term } B}.$$

We will now prove that term B can be ignored compared to term A under our assumptions.

For term B, we have

$$\frac{\frac{\partial w_k(x)}{\partial \mu_k}(w_k^-(x)\delta_k'(x) + w_k^+\epsilon_k(x))}{\gamma_t^2 w_k^2(x)} - \frac{\frac{\partial w_k^-(x)}{\partial \mu_k}\delta_k'(x) + \frac{\partial w_k^+(x)}{\partial \mu_k}\epsilon_k(x)}{\gamma_t^2 w_k(x)}$$

$$= \frac{1}{\gamma_t^2 w_k^2(x)}\left(\frac{\partial w_k(x)}{\partial \mu_k}(w_k^-(x)\delta_k'(x) + w_k^+(x)\epsilon_k(x)) - w_k(x)\left(\frac{\partial w_k^-(x)}{\partial \mu_k}\delta_k'(x) + \frac{\partial w_k^+(x)}{\partial \mu_k}\epsilon_k(x)\right)\right)$$

$$= \frac{1}{\gamma_t^2 w_k^2(x)}\left(\frac{\partial w_k^+(x)}{\partial \mu_k}w_k^-(x)\delta_k'(x) + \frac{\partial w_k^-(x)}{\partial \mu_k}w_k^+(x)\epsilon_k(x) - w_k^+(x)\frac{\partial w_k^-(x)}{\partial \mu_k}\delta_k'(x) - w_k^-(x)\frac{\partial w_k^+(x)}{\partial \mu_k}\epsilon_k(x)\right)$$

$$= \frac{1}{\gamma_t^2 w_k^2(x)}\left(\frac{\partial w_k^+}{\partial \mu_k}w_k^- - \frac{\partial w_k^-}{\partial \mu_k}w_k^+\right)(\epsilon_k(x) - \delta_k'(x))$$

$$= -\frac{2}{\gamma_t^2 w_k^2(x)}\left(\frac{\partial w_k^+}{\partial \mu_k}w_k^- - \frac{\partial w_k^-}{\partial \mu_k}w_k^+\right)\left(I + \frac{s_t^2}{s_t^2 + \gamma_t^2}U_k U_k^\top\right)s_t\mu_k$$

$$= -\frac{4}{\gamma_t^2 w_k^2(x)}s_t^2 w_k^- w_k^+\Sigma_k^{-1}x\left(I + \frac{s_t^2}{s_t^2 + \gamma_t^2}U_k U_k^\top\right)\mu_k = O\left(\frac{r_k^+ r_k^-}{\gamma_t^4}s_t\|\mu_k\|_2\|x\|_2\right).$$

And for term A, we have

$$\frac{w_k^-(x)\frac{\partial \delta_k'(x)}{\partial \mu_k} + w_k^+(x)\frac{\partial \epsilon_k(x)}{\partial \mu_k}}{\gamma_t^2 w_k(x)} = O\left(\frac{s_t\|\mu_k\|_2}{\gamma_t^2}|w_k^+ - w_k^-|\right).$$

Thus,

$$\frac{O\left(\frac{r_k^+ r_k^-}{\gamma_t^4}s_t\|\mu_k\|_2\|x\|_2\right)}{O\left(\frac{s_t\|\mu_k\|_2}{\gamma_t^2}|w_k^+ - w_k^-|\right)} = O\left(\frac{r_k^+ r_k^- w_k\|x\|_2}{\gamma_t^2|r_k^+ - r_k^-|}\right) = O\left(\frac{r_k^+ r_k^- w_k\|x\|_2}{\gamma_t^2}\right) \to 0.$$

Thus, $J_k^\mu \approx -\frac{1}{\gamma_t^2}\left(r_k^+(x)\frac{\partial \delta_k'(x)}{\partial \mu_k} + r_k^-(x)\frac{\partial \epsilon_k(x)}{\partial \mu_k}\right) = -\frac{s_t}{\gamma_t^2}(r_k^+(x) - r_k^-(x))\left(I - \frac{s_t^2}{s_t^2 + \gamma_t^2}U_k U_k^\top\right).$

We will analyze $J_k^U$ now. Recall that

$$J_k^U = -\frac{1}{\gamma_t^2}\frac{(\frac{\partial w_k^-(x)}{\partial U_k}\delta_k'(x) + \frac{\partial \delta_k'(x)}{\partial U_k}w_k^-(x) + \frac{\partial \epsilon_k(x)}{\partial U_k}w_k^+(x) + \frac{\partial w_k^+(x)}{\partial U_k}\epsilon_k(x))w_k(x) - \frac{\partial w_k(x)}{\partial U_k}(w_k^-(x)\delta_k'(x) + w_k^+\epsilon_k(x))}{w_k^2(x)}$$

$$= -\frac{1}{\gamma_t^2}\left(\frac{\frac{\partial \delta_k'(x)}{\partial U_k}w_k^-(x) + \frac{\partial \epsilon_k(x)}{\partial U_k}w_k^+(x)}{w_k(x)}\right.$$

$$\left. + \frac{\frac{\partial w_k^-(x)}{\partial U_k}\delta_k'(x) + \frac{\partial w_k^+(x)}{\partial U_k}\epsilon_k(x)}{w_k(x)} - \frac{\frac{\partial w_k(x)}{\partial U_k}(w_k^-(x)\delta_k'(x) + w_k^+\epsilon_k(x))}{w_k^2(x)}\right).$$

By calculating, we have

$$\frac{\frac{\partial w_k^-(x)}{\partial U_k}\delta_k(x) + \frac{\partial w_k^+(x)}{\partial U_k}\epsilon_k(x)}{w_k(x)} - \frac{\frac{\partial w_k(x)}{\partial U_k}(w_k^-(x)\delta_k'(x) + w_k^+\epsilon_k(x))}{w_k^2(x)}$$

$$= \frac{1}{w_k^2(x)}\left((w_k(x)(\frac{\partial w_k^-(x)}{\partial U_k}\delta_k'(x) + \frac{\partial w_k^+(x)}{\partial U_k}\epsilon_k(x)) - \frac{\partial w_k(x)}{\partial U_k}(w_k^-(x)\delta_k'(x) + w_k^+\epsilon_k(x))\right)$$

$$= \frac{1}{w_k^2(x)}\left(w_k(x)(\frac{\partial w_k^-(x)}{\partial U_k}\delta_k'(x) + \frac{\partial w_k^+(x)}{\partial U_k}\epsilon_k(x)) - \frac{\partial w_k(x)}{\partial U_k}(w_k^-(x)\delta_k'(x) + w_k^+\epsilon_k(x))\right)$$

$$= \frac{1}{w_k^2(x)}\left(\frac{\partial w_k^+}{\partial U_k}w_k^- - \frac{\partial w_k^-}{\partial U_k}w_k^+\right)(\epsilon_k(x) - \delta_k'(x))$$

$$= -\frac{2\,s_t^3}{w_k^2(x)}\left[\mathcal{N}(x; s_t\mu_k, \Sigma)\,M^+(x) - \mathcal{N}(x; -s_t\mu_k, \Sigma)\,M^-(x)\right]U_k\left(I - \alpha\,U_k U_k^\top\right)\mu_k$$

$$= O\left(r_k^+ r_k^- \frac{s_t^3}{\gamma_t^2(s_t^2 + \gamma_t^2)}\right).$$

where $M^+(x) = \Sigma^{-1}(x - s_t\mu_k)(x - s_t\mu_k)^\top\Sigma^{-1} - \Sigma^{-1}, M^-(x) = \Sigma^{-1}(x + s_t\mu_k)(x + s_t\mu_k)^\top\Sigma^{-1} - \Sigma^{-1}, \alpha = \frac{s_t^2}{s_t^2+\gamma_t^2}$.

We also know that

$$\frac{\Sigma_{k=1}^K\left(\frac{\partial\delta_k(x)}{\partial U_k}w_k^-(x) + \frac{\partial\epsilon_k(x)}{\partial U_k}w_k^+(x)\right)}{\Sigma_{k=1}^K w_k(x)} = O\left(\frac{s_t^2\|x\|_2}{s_t^2+\gamma_t^2}\right) = O\left(\frac{s_t^3\|\mu_k\|_2}{s_t^2+\gamma_t^2}\right)$$

$$\frac{O\left(r_k^+ r_k^- \frac{s_t^3}{\gamma_t^2(s_t^2+\gamma_t^2)}\right)}{O\left(\frac{s_t^3\|\mu_k\|_2}{s_t^2+\gamma_t^2}\right)} \to 0.$$

Thus,

$$J_k^U \approx$$
$$\frac{2s_t^2}{\gamma_t^2(s_t^2+\gamma_t^2)}(r_k^-(x)(U_k^\top(x + s_t\mu_k)I + (x + s_t\mu_k)U_k^\top) + r_k^+(x)(U_k^\top(x - s_t\mu_k)I + U_k(x - s_t\mu_k)^\top)).$$

∎

Before we provide the simplification of Hessian, we first prove that for $a, b \in \mathbb{R}^n$ $M = a^\top b I_n + ba^\top$, $MM^\top$ is positive-definite if and only if $b^\top a \neq 0$. At the same time, we provide the minimum eigenvalue of $MM^\top$, which will be used later.

**Lemma B.3.** *Let $a, b \in \mathbb{R}^n$ and $M = a^\top b I_n + ba^\top$. $MM^\top$ is positive-definite if and only if $b^\top a \neq 0$.*

*Moreover,*

$$\lambda_{\min}(MM^\top) = \mu_2 = \frac{4(a^\top b)^2 + \|a\|_2^2\|b\|_2^2 - \|a\|_2\|b\|_2\sqrt{8(a^\top b)^2 + \|a\|_2^2\|b\|_2^2}}{2}.$$

**Proof.** Let $M = a^\top b I_n + ba^\top$, $c = a^\top b$. We know that $\forall x \in \mathbb{R}^n$,
$$x^\top MM^\top x = (M^\top x)^\top(M^\top x)$$
$$= \|M^\top x\|_2^2 \geq 0.$$

Thus, $MM^\top$ is semi-positive definite.

We can also have that

$$|M| = |a^\top b I_n + ba^\top| = c^n|I_n + \frac{1}{c}ba^\top| = 2c^n \geq 0,$$

where $c^n = 0$ if and only if $b^\top a = 0$.

The last equation holds because

$$|I_n + uv^\top| = 1 + v^\top u$$

Thus, $|MM^\top| > 0$, $MM^\top$ is positive definite.

We can further get the eigenvalues of $MM^\top$.

Expanding gives the convenient representation

$$MM^\top = (a^\top b)^2 I_n + a^\top b(ba^\top + ab^\top) + a^\top abb^\top. \tag{6}$$

$\forall x \in \mathbb{R}^n$, if $x^\top a = 0$ and $x^\top b = 0$, we have:

$$MM^\top x = (a^\top b)^2 x.$$

Thus, $(a^\top b)^2$ is an eigenvalue of $M$, and its eigenspace contains the orthogonal complement of $\mathrm{span}\{a, b\}$. If $a$ and $b$ are linearly independent then $\dim(\mathrm{span}\{a, b\}) = 2$, so the multiplicity of the eigenvalue $\alpha^2$ is at least $n - 2$.

To find the remaining eigenvalues we restrict $M$ to the subspace $\mathcal{S} := \mathrm{span}\{a, b\}$. Assume first that $a$ and $b$ are linearly independent so that $\mathcal{S}$ is two-dimensional.

Using equation 6, we can compute $tr(MM^\top)$, which is

$$\begin{aligned}
tr(MM^\top) &= tr((a^\top b)^2 I_n + a^\top b(ba^\top + ab^\top) + a^\top abb^\top) \\
&= n(a^\top b)^2 + 2(a^\top b)^2 + \|a\|_2^2 \|b\|_2^2 \\
&= (n+2)(a^\top b)^2 + \|a\|_2^2 \|b\|_2^2.
\end{aligned}$$

The second equation holds because of $tr(xy^\top) = tr(y^\top x) = y^\top x$.

We set the other two eigenvalues are $\mu_1$ and $\mu_2$. Thus

$$tr(MM^\top) = \Sigma_{i=1}^n \lambda_i = (n-2)(a^\top b)^2 + \mu_1 + \mu_2 = (n+2)(a^\top b)^2 + \|a\|_2^2 \|b\|_2^2,$$

and

$$|MM^\top| = \Pi_{i=1}^n \lambda_i = (a^\top b)^{2(n-2)} \mu_1 \mu_2 = 4(a^\top b)^{2n}.$$

So $\mu_1$ and $\mu_2$ are the two solutions of

$$x^2 - \left(4(a^\top b)^2 + \|a\|_2^2 \|b\|_2^2\right) x + 4(a^\top b)^4 = 0. \tag{7}$$

Solving equation 7, we have

$$\mu_1, \mu_2 = \frac{4(a^\top b)^2 + \|a\|_2^2 \|b\|_2^2 \pm \|a\|_2 \|b\|_2 \sqrt{8(a^\top b)^2 + \|a\|_2^2 \|b\|_2^2}}{2}.$$

Now we obtain all eigenvalues. Moreover, we can calculate the minimum of eigenvalues.

$$\lambda_{\min}(MM^\top) = \mu_2 = \frac{4(a^\top b)^2 + \|a\|_2^2 \|b\|_2^2 - \|a\|_2 \|b\|_2 \sqrt{8(a^\top b)^2 + \|a\|_2^2 \|b\|_2^2}}{2}.$$

∎

**Lemma B.4.** *[Eigenvalues of the Hessian blocks] Under the same conditions, $H$ is convex. If $\forall x \in \mathbb{R}^{d_k}, r_k^+(x) = 1$ or $r_k^-(x) = 1$ are strictly satisfied, the eigenvalues of the Hessian at $\theta^*$ are*

$$\lambda_{\min}(H_{\mu_k \mu_k}) = \frac{s_t^2}{(s_t^2 + \gamma_t^2)^2}, \text{ and}$$

$$\lambda_{\min}(H_{U_k U_k}) = \frac{4(U_k^\top \mu_k))^2 + \|U_k\|_2^2 \|\mu_k\|_2^2 - \|U_k\|_2 \|\mu_k\|_2 \sqrt{8(U_k^\top \mu_k))^2 + \|U_k\|_2^2 \|\mu_k\|_2^2}}{2}.$$

**Proof.** We first state the convexity of the loss function near the true value $\theta^\star$.

Let $\theta = \theta^\star + \Delta\theta$, we have

$$s_\theta(x, t) = s_{\theta^\star}(x, t) + (\nabla_\theta s_\theta(x, t)|_{\theta^\star})^\top [\Delta\theta] + O(\|\Delta\theta\|_2^2).$$

Thus,

$$\begin{aligned}
L(\theta) &= \mathbb{E}_{x \sim p_t(x)}[(s_\theta(x, t) - \nabla \log p_t(x))^\top (s_\theta(x, t) - \nabla \log p_t(x))] \\
&= \mathbb{E}_{x \sim p_t(x)}[(s_{\theta^\star}(x, t) + (\nabla_\theta s_\theta(x, t)|_{\theta^\star})^\top [\Delta\theta] + O(\|\Delta\theta\|_2^2) - \nabla \log p_t(x))^\top \\
&\quad (s_{\theta^\star}(x, t) + (\nabla_\theta s_\theta(x, t)|_{\theta^\star})^\top [\Delta\theta] + O(\|\Delta\theta\|_2^2) - \nabla \log p_t(x))] \\
&= \mathbb{E}_{x \sim p_t(x)}[((\nabla_\theta s_\theta(x, t)|_{\theta^\star})^\top [\Delta\theta])^\top (\nabla_\theta s_\theta(x, t)|_{\theta^\star} [\Delta\theta])] + O(\|\Delta\theta\|_2^3) \\
&= (\Delta\theta)^\top \mathbb{E}_{x \sim p_t(x)}[(\nabla_\theta s_\theta(x, t)|_{\theta^\star})(\nabla_\theta s_\theta(x, t)|_{\theta^\star})^\top] \Delta\theta \\
&\overset{\Delta}{=} (\Delta\theta)^\top H \Delta\theta.
\end{aligned}$$

$$\frac{\partial^2 L(\theta)}{\partial \theta^2} = 2H.$$

We then analyze the convexity of $\mathbb{E}_{x \sim p_t(x)}[(\nabla_\theta s_\theta(x,t)|_{\theta^\star})(\nabla_\theta s_\theta(x,t)|_{\theta^\star})^\top] \triangleq H$. We can divide H into 4 parts: $H_{\mu\mu}, H_{UU}, H_{\mu U}$ and $H_{U\mu}$, where $H_{U\mu} = (H_{\mu U})^\top$.

Let $J_k^\mu|_\theta = \frac{\partial s_\theta}{\partial \mu_k}|_\theta$.

$$H = \mathbb{E}_{x \sim p_t(x)}[(\nabla_\theta s_\theta(x,t)|_{\theta^\star})(\nabla_\theta s_\theta(x,t)|_{\theta^\star})^\top]$$
$$= \mathbb{E}_{x \sim p_t(x)}[J_{\theta^\star}(x,t)J_{\theta^\star}(x,t)^\top.$$

**Term $H_{\mu\mu}$**

We will show that $H_{\mu_k \mu_k}$ is $\alpha$-convex, where $\alpha > 0$.

$$H_{\mu_k \mu_k} = \mathbb{E}_{x \sim p_t(x)}[J_k^\mu J_k^{\mu\top}]$$

$$H_{\mu_k \mu_k} \approx \mathbb{E}_{x \sim p_t(x)}[J_k^\mu J_k^{\mu\top}] \approx \frac{s_t^2}{\gamma_t^4} \mathbb{E}_{x \sim p_t(x)}[(r_k^+(x) - r_k^-(x))^2](I - \frac{s_t^2}{s_t^2 + \gamma_t^2} U_k U_k^\top)^2.$$

Let $P_k = U_k U_k^\top$, $\alpha = \frac{s_t^2}{s_t^2 + \gamma_t^2}$,

$$(I - \alpha P_k)(I - \alpha P_k)^\top = (I - \alpha P_k)^2 = I - 2\alpha P_k + \alpha^2 P_k^2 = (I - \alpha P_k)^2.$$

We then prove that $\lambda_{\min}((I - \alpha P_k)^2) = (\frac{\gamma_t^2}{s_t^2 + \gamma_t^2})^2$.

First, we calculate the eigenvalue of $P$.

$$P^2 = P \Rightarrow \lambda_1 = 1, \ \lambda_2 = 0.$$

Then we take subspace $Col(P) = \{v \ ; v = Px, \ x \in \mathcal{R}^D\}$ corresponding to $\lambda_1$, and subspace $Ker(P) = \{v \ ; Pv = 0, \ x \in \mathcal{R}^D\}$ corresponding to $\lambda_2$.

If $w \in Col(P)$, $Pw = w$:

$$(I - \alpha P)w = (1 - \alpha)w,$$

and

$$(I - \alpha P)^2 w = (1 - \alpha)^2 w,$$

Thus,

$$\lambda_1' = (1 - \alpha)^2.$$

Similar to the previous derivation, if $w \in Ker(P)$, $Pw = 0$:

$$(I - \alpha P)w = w$$
$$(I - \alpha P)^2 w = w$$

Thus,

$$\lambda_2' = 1.$$

Therefore, $\lambda_{\min}((I - \alpha P_k)^2) = \left(\frac{\gamma_t^2}{s_t^2 + \gamma_t^2}\right)^2$. Hence, we have

$$\lambda_{\min}(H_{\mu_k \mu_k}) \geq \frac{s_t^2}{\gamma_t^4} \frac{c_k \gamma_t^4}{(s_t^2 + \gamma_t^2)^2} \approx \frac{s_t^2}{(s_t^2 + \gamma_t^2)^2},$$

where $c_k = \mathbb{E}_{x \sim p_t(x)}[(r_k^+(x) - r_k^-(x))^2] \approx 1$.

**Term $H_{U_k U_k}$**

$$
\begin{aligned}
H_{U_k U_k} &\approx \mathbb{E}_{x \sim p_t(x)}[J_U^k J_U^{k\top}] \\
&\approx \frac{4s_t^4}{\gamma_t^4(s_t^2 + \gamma_t^2)^2} \mathbb{E}_{x \sim p_t(x)}[(U_k^\top(x + s_t\mu_k)I + (x + s_t\mu_k)U_k^\top)(U_k^\top(x + s_t\mu_k)I + (x + s_t\mu_k)U_k^\top)^\top] \\
&= \frac{4s_t^4}{\gamma_t^4(s_t^2 + \gamma_t^2)^2}(s_t^2 U_k^\top \mu_k \mu_k^\top U_k I + s_t^2 \mu_k^\top U_k(\mu_k U_k^\top + U_k \mu_k^\top) + \mu_k U_k^\top U_k \mu_k^\top + M(x),)
\end{aligned}
$$

where $M(x)$ is semi-positive for $\mathbb{E}_{x \sim p_t(x)}[x] = 0$.

Using lemma B.3, we can take $a = U_k$ and $b = \mu_k$ and obtain that

$H_{U_k U_k}$ is positive definite and

$$
\lambda_{\min}(H_{U_k U_k}) = \frac{4(U_k^\top \mu_k))^2 + \|U_k\|_2^2 \|\mu_k\|_2^2 - \|U_k\|_2 \|\mu_k\|_2 \sqrt{8(U_k^\top \mu_k))^2 + \|U_k\|_2^2 \|\mu_k\|_2^2}}{2}.
$$

**Term $H_{\mu_k U_k}$ and Term $H_{U_k \mu_k}$**

Since $H_{U_k \mu_k} = H_{\mu_k U_k}^\top$, we just analyze $H_{\mu_k U_k}$. We want to analyze the Hessian block

$$
H_{\mu_k U_k} = \mathbb{E}_{x \sim p_t}\left[J_k^U(x)(J_k^\mu(x))^\top\right],
$$

and show that under symmetric assumptions, this cross-term is zero.

The first-order derivative with respect to $\mu_k$ is approximately:

$$
J_k^\mu(x) \approx -\frac{s_t}{\gamma_t^2}(r_k^+(x) - r_k^-(x))(I - \alpha U_k U_k^\top), \qquad \alpha = \frac{s_t^2}{s_t^2 + \gamma_t^2}.
$$

The first-order derivative with respect to $U_k$ is approximately:

$$
J_k^U(x) \approx -\frac{1}{\gamma_t^2}\left[r_k^-(x)\frac{\partial \delta_k(x)}{\partial U_k} + r_k^+(x)\frac{\partial \epsilon_k(x)}{\partial U_k}\right],
$$

with

$$
\frac{\partial \delta_k(x)}{\partial U_k} = -2\frac{s_t^2}{s_t^2 + \gamma_t^2}U_k(x + s_t\mu_k), \qquad \frac{\partial \epsilon_k(x)}{\partial U_k} = -2\frac{s_t^2}{s_t^2 + \gamma_t^2}U_k(x - s_t\mu_k).
$$

combining terms:

$$
J_k^U(x) = C \cdot U_k\left[r_k^-(x)(x + s_t\mu_k) + r_k^+(x)(x - s_t\mu_k)\right],
$$

where $C = \frac{2s_t^2}{\gamma_t^2(s_t^2 + \gamma_t^2)}$. Assume that the underlying component distribution $p_k(x)$ is symmetric:

$$
p_k(x) = p_k(-x),
$$

and the weights satisfy:

$$
r_k^+(-x) = r_k^-(x), \qquad r_k^-(-x) = r_k^+(x).
$$

Then we have:

**(a) $J_k^\mu(x)$ is an odd function:**

$$
\begin{aligned}
J_k^\mu(-x) &= -\frac{s_t}{\gamma_t^2}(r_k^+(-x) - r_k^-(-x))(I - \alpha U_k U_k^\top) \\
&= -\frac{s_t}{\gamma_t^2}(r_k^-(x) - r_k^+(x))(I - \alpha U_k U_k^\top) \\
&= -J_k^\mu(x).
\end{aligned}
$$

**(b) $J_k^U(x)$ is an odd function:**

$$
\begin{aligned}
J_k^U(-x) &= C\,U_k\left[r_k^-(-x)(-x+s_t\mu_k)+r_k^+(-x)(-x-s_t\mu_k)\right]\\
&= C\,U_k\left[r_k^+(x)(-x+s_t\mu_k)+r_k^-(x)(-x-s_t\mu_k)\right]\\
&= -C\,U_k\left[r_k^-(x)(x+s_t\mu_k)+r_k^+(x)(x-s_t\mu_k)\right]\\
&= -J_k^U(x).
\end{aligned}
$$

Now compute:

$$
H_{\mu_k U_k} = \int J_k^U(x)\,(J_k^\mu(x))^\top\,p_k(x)\,dx.
$$

Using symmetry:

$$
= \int J_k^U(-x)\,(J_k^\mu(-x))^\top\,p_k(-x)\,dx = \int (-J_k^U(x))\,(-J_k^\mu(x))^\top\,p_k(x)\,dx = H_{\mu_k U_k}.
$$

Thus,

$$
H_{\mu_k U_k} = \mathbb{E}_{x\sim p_{data}}[J_k^\mu(J_k^U)^\top] = \mathbb{E}_{x\sim p_{data}}\Big[\frac{2s_t^3}{\gamma_t^4(s_t^2+\gamma_t^2)}(r_k^+(x)-r_k^-(x))(1-\frac{s_t^2}{s_t^2+\gamma_t^2}U_kU_k^\top)
$$
$$
(r_k^-(x)(U_k^\top(x+s_t\mu_k)I+U_k(x+s_t\mu_k)^\top)+r_k^+(x)(U_k^\top(x-s_t\mu_k)I+U_k(x-s_t\mu_k)^\top))\Big].
$$

$$
\begin{aligned}
\lambda_{H_{\mu\mu}} &= \mathbb{E}_{x\sim p_{data}}[(u^\top J_\mu^k)^2]\\
\lambda_{H_{UU}} &= \mathbb{E}_{x\sim p_{data}}[(u^\top J_U^k)^2]\\
\lambda_{H_{\mu U}} &= \mathbb{E}_{x\sim p_{data}}[(u^\top J_\mu^k)(u^\top J_U^k)] \le \sqrt{\lambda_{H_{\mu\mu}}\lambda_{H_{\mu U}}}.
\end{aligned}
$$

∎

**Analyze H**

We have

$$
H = \begin{pmatrix} H_{\mu_k\mu_k} & H_{\mu_k U_k}\\ H_{\mu_k U_k} & H_{U_k U_k} \end{pmatrix}.
$$

If we can prove that $H_{\mu_k\mu_k}-H_{U_k\mu_k}H_{U_kU_k}^{-1}H_{U_k\mu_k}^\top$ is positive-definite, then $H$ is positive-definite for Schur's Theorem.

We know that

$$
\lambda_H \ge \lambda_S \ge \lambda_{H_{\mu_k\mu_k}} - \frac{r^2\lambda_{H_{\mu_k\mu_k}}\lambda_{H_{U_kU_k}}}{\lambda_{H_{U_kU_k}}} = (1-r^2)\lambda_{H_{\mu_k\mu_k}} \ge (1-r^2)\frac{s_t^2}{(s_t^2+\gamma_t^2)^2} > 0,
$$

$$
r = \max_{\|u\|=1,\|v\|=1}\frac{u^\top H_{\mu_k U_k}v}{\sqrt{u^\top H_{\mu_k\mu_k}u \cdot v^\top H_{U_kU_k}v}} \le 1,
$$

where $r=1$ if and only if $u^\top J_\mu^k = cv^\top J_U^k, c\ne 0$, which is almost impossible to happen.

More specially, if we assume that $\forall x\in\mathbb{R}^{d_k}, r_k^+=1$ or $r_k^-=1$, since

$$
H_{\mu_k U_k} = \mathbb{E}_{x\sim p_{data}}[J_k^\mu(J_k^U)^\top] = \mathbb{E}_{x\sim p_{data}}\Big[\frac{2s_t^3}{\gamma_t^4(s_t^2+\gamma_t^2)}(r_k^+(x)-r_k^-(x))(1-\frac{s_t^2}{s_t^2+\gamma_t^2}U_kU_k^\top)
$$
$$
(r_k^-(x)(U_k^\top(x+s_t\mu_k)I+U_k(x+s_t\mu_k)^\top)+r_k^+(x)(U_k^\top(x-s_t\mu_k)I+U_k(x-s_t\mu_k)^\top))\Big]
$$
$$
= \mathbb{E}_{x\sim\mathcal{N}(s_t\mu_k,\Sigma_k)}\Big[\frac{2s_t^3}{\gamma_t^4(s_t^2+\gamma_t^2)}(r_k^+(x)-r_k^-(x))\left(1-\frac{s_t^2}{s_t^2+\gamma_t^2}U_kU_k^\top\right)
$$
$$
(r_k^-(x)(U_k^\top(x+s_t\mu_k)I+U_k(x+s_t\mu_k)^\top)+r_k^+(x)(U_k^\top(x-s_t\mu_k)I+U_k(x-s_t\mu_k)^\top))\Big]
$$
$$
= \mathbb{E}_{x\sim\mathcal{N}(s_t\mu_k,\Sigma_k)}\Big[\frac{2s_t^3}{\gamma_t^4(s_t^2+\gamma_t^2)}\left(1-\frac{s_t^2}{s_t^2+\gamma_t^2}U_kU_k^\top\right)+(U_k^\top(x-s_t\mu_k)I+U_k(x-s_t\mu_k)^\top))\Big]
$$
$$
= 0,
$$

We have $r = 0$,

$$\alpha = \min\{\frac{s_t^2}{(s_t^2 + \gamma_t^2)^2}, \frac{4(U_k^\top \mu_k))^2 + \|U_k\|_2^2\|\mu_k\|_2^2 - \|U_k\|_2\|\mu_k\|_2\sqrt{8(U_k^\top \mu_k))^2 + \|U_k\|_2^2\|\mu_k\|_2^2}}{2}\}.$$

Utill now, We have shown that $H$ is $\alpha$-convex and L-lipschiz, where $\alpha = (1 - r^2)\lambda_{H_{\mu_k \mu_k}}$. And we can know that $L(\theta)$ is exponentially convergent.

**Theorem B.5.** *If we take $\eta_t = \eta = \frac{2}{\eta + L}$, and $\kappa = \frac{L}{\alpha}$, then*

$$\|\theta^t - \theta^\star\|_2 \leq \left(\frac{\kappa - 1}{\kappa + 1}\right)^t \|\theta^{(0)} - \theta^\star\|_2.$$

## C   MULTI-MODE MoG OPTIMIZATION

In this section, we analyze the convergence guarantee of multi-modal MoG latent with highly separated Gaussain assumption. For the $k$-th subspace, we have that

$$\nabla \log p_{t,k}(x) = \frac{\nabla p_{t,k}(x)}{p_{t,k}(x)}$$

$$= -\frac{1}{\gamma_t^2} \frac{\Sigma_{l=1}^{n_k}\pi_{k,l}\mathcal{N}(x; s_t\mu_{k,l}, s_t^2 U_{k,l}^\star U_{k,l}^{\star\top} + \gamma_t^2 I)\left(x - s_t\mu_{k,l} - \frac{s_t^2}{s_t^2 + \gamma_t^2}U_{k,l}^\star U_{k,l}^{\star\top}(x - s_t\mu_{k,l})\right)}{\Sigma_{l=1}^{n_k}\pi_{k,l}\mathcal{N}(x; s_t\mu_{k,l}, s_t^2 U_{k,l}^\star U_{k,l}^{\star\top} + \gamma_t^2 I)}.$$

### C.1   OPTIMIZATION

**Assumption C.1.** [Highly Separated Gaussian] Consider the Gaussian mixture

$$p_k(x) = \sum_{l=1}^{n_k} \pi_{k,l}\,\mathcal{N}(x; \mu_{k,l}, \Sigma_{k,l}), \qquad r_{k,l}(x) := \frac{\pi_{k,l}\,\mathcal{N}(x; \mu_{k,l}, \Sigma_{k,l})}{\sum_{i=1}^{n_k}\pi_{k,i}\,\mathcal{N}(x; \mu_{k,i}, \Sigma_{k,i})}.$$

There exist constants $\varepsilon \ll 1$ and $\delta \ll 1$ such that when $x \sim p_k$ we have

$$\Pr_{x \sim p_k}\left(\exists l \in \{1, \ldots, n_k\} \text{ with } r_{k,l}(x) \geq 1 - \varepsilon\right) \geq 1 - \delta.$$

We assume that the gap between the subspaces is large, and the gap within the subspace is relatively small, and the equivalent Gaussian is used to replace the whole subspace.

**Corollary C.2.** *Assume that $\|\mu_{k,i}^* - \mu_{k,j}^*\|_2 \leq \delta$, $\|U_{k,i}^* - U_{k,j}^*\|_2 \leq \epsilon$ and $\|x - \bar{\mu}_k^*\|_2 \leq \Delta$. We have*

$$\|\log p(x) - \log \bar{p}(x)\|_2 = O(\epsilon + \delta\Delta + \Delta^3)$$

**Proof.** For $k$-th subspace, $w_k(x) = \Sigma_{l=1}^{n_k}\pi_{k,l}\mathcal{N}(x; s_t\mu_{k,l}, \Sigma_{k,l})$, we take

$$\widetilde{w}_k(x) = \mathcal{N}(x; \bar{\mu}_k, \bar{\Sigma}_k).$$

where

$$\mathbb{E}_{\widetilde{w}_k}[x] = \bar{\mu}_k = \mathbb{E}_{w_k}[x] = \Sigma_{l=1}^{n_k}\pi_{k,l}s_t\mu_{k,l}$$

$$\text{Cov}_{\widetilde{w}_k}(x) = \text{Cov}_{w_k}(x) = \mathbb{E}[(x - \bar{\mu}_k)(x - \bar{\mu}_k)^\top] = \Sigma_{l=1}^{n_k}\pi_{k,l}(\Sigma_{k,l} + s_t^2\mu_{k,l}\mu_{k,l}^\top - s_t^2\bar{\mu}_{k,l}\bar{\mu}_{k,l}^\top)$$

$$\Rightarrow \bar{\Sigma}_k = \Sigma_{l=1}^{n_k}(\Sigma_{k,l} + s_t^2\mu_{k,l}\mu_{k,l}^\top - s_t^2\bar{\mu}_{k,l}\bar{\mu}_{k,l}^\top).$$

We next show the order of the estimation under the condition that $\|\mu_{k,i} - \mu_{k,j}\|_2 \leq \delta$, $\|U_{k,i} - U_{k,j}\|_2 \leq \epsilon$ and $\|x - \bar{\mu}_k\|_2 \leq \Delta$. Using Taylor's Theorem and take $x_0 = \bar{\mu}_k$, we can obtain that

$$\log p(x) = \log p(x_0) + (x - x_0)^\top \nabla \log p(x_0) + \frac{1}{2}(x - x_0)^\top \nabla^2 \log p(x_0)(x - x_0) + O(\|x - x_0\|^3)$$

$$\log \tilde{p}(x) = \log \tilde{p}(x_0) + (x - x_0)^\top \nabla \log \tilde{p}(x_0) + \frac{1}{2}(x - x_0)^\top \nabla^2 \log \tilde{p}(x_0)(x - x_0) + O(\|x - x_0\|^3).$$

We analyzed the results of their subtraction item by item.

First,

$$
\begin{aligned}
\log p(x_0) - \log \tilde{p}(x_0) &= \log \frac{\Sigma_{l=1}^{n_k} \pi_{k,l} \mathcal{N}(x_0; \mu_{k,l}, \Sigma_{k,l})}{\mathcal{N}(x_0; \bar{\mu}_k, \bar{\Sigma}_k)} \\
&= \log \left( \Sigma_{l=1}^{n_k} \pi_{k,l} \frac{1}{|\Sigma_{k,l}|^{\frac{1}{2}}} \exp(-\frac{1}{2}(\bar{\mu} - \mu_{k,l})^\top \Sigma_{k,l}^{-1}(\bar{\mu} - \mu_{k,l})) \right) + \frac{1}{2} \log |\bar{\Sigma}_k| \\
&= \log \left( \Sigma_{l=1}^{n_k} \pi_{k,l} \frac{1}{|\Sigma_{k,l}|^{\frac{1}{2}}} (1 + O(\delta^2)) \right) + \frac{1}{2} \log |\bar{\Sigma}_k| \\
&= \log \left( \Sigma_{l=1}^{n_k} \pi_{k,l} \frac{|\bar{\Sigma}_k|^{\frac{1}{2}}}{|\Sigma_{k,l}|^{\frac{1}{2}}} + O(\delta^2) \right) \\
&= O \left( \Sigma_{l=1}^{n_k} \pi_{k,l} (\frac{|\bar{\Sigma}_k|^{\frac{1}{2}}}{|\Sigma_{k,l}|^{\frac{1}{2}}} - 1) \right) + O(\delta^2),
\end{aligned}
$$

and

$$
\| \log p(x_0) - \log \tilde{p}(x_0) \|_2 = O(\epsilon + \delta^2).
$$

For the first derivative, we also have

$$
\begin{aligned}
\nabla \log p(x_0) - \nabla \log \tilde{p}(x_0) &= \nabla \log \Sigma_{l=1}^{n_k} \pi_{k,l} \mathcal{N}(x; \mu_{k,l}, \Sigma_{k,l})|_{x_0} \\
&= \frac{\Sigma_{l=1}^{n_k} \pi_{k,l} \mathcal{N}(x_0; \mu_{k,l}, \Sigma_{k,l})(-\Sigma_{k,l}^{-1}(\bar{\mu} - \mu_{k,l})))}{p(x_0)}.
\end{aligned}
$$

$$
\| \nabla \log p(x_0) - \nabla \log \tilde{p}(x_0) \|_2 = O(\delta).
$$

For the second derivative,

$$
\begin{aligned}
\nabla^2 \log p(x_0) - \nabla^2 \log \tilde{p}(x_0) &= \frac{\nabla^2 p(x_0)}{p(x_0)} - (\frac{\nabla p(x_0)}{p(x_0)})(\frac{\nabla p(x_0)}{p(x_0)})^\top - \frac{\nabla^2 \tilde{p}(x_0)}{\tilde{p}(x_0)} \\
&= (\frac{\nabla^2 p(x_0)}{p(x_0)} - \frac{\nabla^2 \tilde{p}(x_0)}{\tilde{p}(x_0)}) - (\frac{\nabla p(x_0)}{p(x_0)})(\frac{\nabla p(x_0)}{p(x_0)})^\top.
\end{aligned}
$$

$$
\| \nabla^2 \log p(x_0) - \nabla^2 \log \tilde{p}(x_0) \|_2 = O(\epsilon^2 + \delta^2).
$$

Thus, $\| \log p(x) - \log \tilde{p}(x) \|_2 = O(\epsilon + \delta \Delta + \Delta^3)$. ∎

**Lemma C.3.** *[Eigenvalues of the Hessian] Assume Theorem 6.4, the Hessian at the $k$-th subspace is convex on a neighborhood of $\theta^*$. If $\forall x \in \mathbb{R}^{d_k}$, $r_k^+(x) = 1$ or $-1$ are strictly satisfied, we have*

$$
\lambda_{\min}(H_{\mu_{k,l}\mu_{k,l}}) = \frac{\pi_{k,l} s_t^2}{(s_t^2 + \gamma_t^2)^2},
$$

*and $\lambda_{\min}(H_{U_{k,l}U_{k,l}})$ has the following form:*

$$
\left( \pi_{k,l} 4(U_{k,l}^\top \mu_{k,l}))^2 + \|U_{k,l}\|_2^2 \|\mu_{k,l}\|_2^2 - \|U_{k,l}\|_2 \|\mu_{k,l}\|_2 \sqrt{8(U_{k,l}^\top \mu_{k,l}))^2 + \|U_{k,l}\|_2^2 \|\mu_{k,l}\|_2^2} \right) / 2.
$$

**Proof.** According to the previous conclusion, we only need to calculate $J_\mu$ and $J_U$. With these assumptions and simplifications, similar to the symmetry case, we will prove that $J_{k,l}^\mu$ and $J_{k,l}^U$ have

dominant terms.

$$J_{k,l}^{\mu}(x)$$

$$= -\frac{1}{\gamma_t^2} \frac{\partial s_\theta(x,t)}{\partial \mu_{k,l}}$$

$$= -\frac{1}{\gamma_t^2} \frac{\Sigma_{l=1}^{n_k} \left( \frac{\partial w_{k,l}(x)}{\partial \mu_{k,l}} \delta_{k,l}(x) + \frac{\partial \delta_{k,l}(x)}{\partial \mu_{k,l}} w_{k,l}(x) \right) w_k(x) - \frac{\partial w_k(x)}{\partial \mu_{k,l}} \Sigma_{l=1}^{n_k} w_{k,l}(x)\delta_{k,l}(x)}{w_k^2(x)}$$

$$= -\frac{1}{\gamma_t^2} \left( \frac{\Sigma_{l=1}^{n_k} \frac{\partial w_{k,l}(x)}{\partial \mu_{k,l}} \delta_{k,l}(x)}{w_k(x)} + \frac{\Sigma_{l=1}^{n_k} \frac{\partial \delta_{k,l}(x)}{\partial \mu_{k,l}} w_{k,l}(x)}{w_k(x)} - \frac{\frac{\partial w_k(x)}{\partial \mu_{k,l}} \Sigma_{l=1}^{n_k} w_{k,l}(x)\delta_{k,l}(x)}{w_k^2(x)} \right).$$

Let's go ahead and do the calculation.

$$\frac{\Sigma_{l=1}^{n_k} \frac{\partial w_{k,l}(x)}{\partial \mu_{k,l}} \delta_{k,l}(x)}{w_k(x)} - \frac{(\frac{\partial w_k(x)}{\partial \mu_{k,l}}) \Sigma_{l=1}^{n_k} w_{k,l}(x)\delta_{k,l}(x)}{w_k^2(x)} = \frac{\frac{\partial w_{k,l}(x)}{\partial \mu_{k,l}}}{w_k(x)} (\delta_{k,l}(x) - \bar{\delta}_k(x))$$

$$\frac{\Sigma_{l=1}^{n_k} \frac{\partial \delta_{k,l}(x)}{\partial \mu_{k,l}} w_{k,l}(x)}{w_k(x)} \approx \frac{s_t}{\gamma_t^2} \Sigma_{l=1}^{n_k} r_{k,l}(x) \left( I - \frac{s_t^2}{s_t^2 + \gamma_t^2} U_{k,l} U_{k,l}^\top \right).$$

where $r_{k,l}(x) = \frac{\pi_{k,l} \mathcal{N}\left(x; \bar{\mu}_k, \bar{\Sigma}_k\right)}{\Sigma_{j=1}^K \mathcal{N}\left(x; \bar{\mu}_j, \bar{\Sigma}_j\right)}.$

Therefore, we can obtain that

$$\left\| \frac{\Sigma_{l=1}^{n_k} \frac{\partial w_{k,l}(x)}{\partial \mu_{k,l}} \delta_{k,l}(x)}{w_k(x)} - \frac{\frac{\partial w_k(x)}{\partial \mu_{k,l}} \Sigma_{l=1}^{n_k} (w_{k,l}(x)\,\delta_{k,l}(x))}{w_k^2(x)} \right\|_2 = O\left( \delta\,(R + s_t B_\mu)\,\frac{s_t^2}{\gamma_t^2} \right)$$

$$\left\| \frac{\Sigma_{l=1}^{n_k} \frac{\partial \delta_{k,l}(x)}{\partial \mu_{k,l}} w_{k,l}(x)}{w_k(x)} \right\|_2 = O\left( s_t \right).$$

where $\delta \le \| \mu_{k,i} - \mu_{k,j} \|_2 \ll 1$.

Thus, we have

$$J_{k,l}^{\mu}(x) = \frac{\partial s_\theta}{\partial \mu_{k,l}} \approx \frac{s_t}{\gamma_t^2} r_{k,l}(x) \left( I - \frac{s_t^2}{s_t^2 + \gamma_t^2} U_{k,l} U_{k,l}^\top \right).$$

We know that

$$H_{\mu_{k,l}\mu_{k,l}} = \mathbb{E}_{x \sim p_t} \left[ J_{k,l}^{\mu}(x)\, J_{k,l}^{\mu}(x)^\top \right]$$

$$= \frac{s_t^2}{\gamma_t^4} \mathbb{E}\left[ r_{k,l}(x)^2 \right] \left( I - \frac{s_t^2}{s_t^2 + \gamma_t^2} U_{k,l} U_{k,l}^\top \right) \left( I - \frac{s_t^2}{s_t^2 + \gamma_t^2} U_{k,l} U_{k,l}^\top \right)^\top.$$

For a given $x$, since we focus on the equivalent Gaussian distribution for each cluster, we have

$$H_{\mu_k \mu_k} \approx diag(\mathbb{E}[r_{k,1}^2] H_{\mu_{k,1}\mu_{k,1}}, \mathbb{E}[r_{k,2}^2] H_{\mu_{k,2}\mu_{k,2}}, \ldots, \mathbb{E}[r_{k,n_k}^2] H_{\mu_{k,n_k}\mu_{k,n_k}}).$$

We first show that $\mathbb{E}[r_{k,l}^2] H_{\mu_{k,l}\mu_{k,l}}$ is positive-definite, then we will further show that $H_{\mu_k \mu_k}$ is positive-definite.

For $H_{\mu_{k,l}\mu_{k,l}}$, we know that

$$\lambda_{\min}(H_{\mu_{k,l}\mu_{k,l}}) = c_{k,l} \lambda_{\min}(J_{k,l}^{\mu}(J_{k,l}^{\mu})^\top)$$

$$= c_{k,l} \lambda_{\min}((I - \alpha P_k)^2)$$

$$= \frac{c_{k,l} \gamma_t^4}{(s_t^2 + \gamma_t^2)^2},$$

where

$$c_{k,l} = \frac{s_t^2}{\gamma_t^4}\mathbb{E}[r_{k,l}^2] \approx \pi_{k,l}\frac{s_t^2}{\gamma_t^4}.$$

We know that for a block matrix $A = diag(A_1, A_2, \ldots, A_k)$,

$$\lambda(A) = \cup_{i=1}^k \lambda(A_i).$$

Therefore,

$$\lambda_{\min}(H_{\mu_k\mu_k}) = \min_{l=1\ldots,n_k} \frac{c_{k,l}\gamma_t^4}{(s_t^2+\gamma_t^2)^2}.$$

Thus, we take

$$\lambda_{H_{\mu_k\mu_k}} = \frac{c_{k,n_k}\gamma_t^4}{(s_t^2+\gamma_t^2)^2}.$$

Similar to previous situation, since

$$\frac{\left\|\frac{\Sigma_{l=1}^{n_k}(\frac{\partial \delta_{k,l}(x)}{\partial U_{k,l}}w_{k,l}(x))(w_k(x)) - (\frac{\partial w_k(x)}{\partial U_{k,l}})\Sigma_{l=1}^{n_k}w_{k,l}(x)\delta_{k,l}(x)}{w_k^2(x)}\right\|_2}{\left\|\frac{\Sigma_{l=1}^{n_k}\frac{\partial \delta_{k,l}(x)}{\partial U_{k,l}}w_{k,l}(x)}{w_k(x)}\right\|_2} \to 0.$$

we can obtain that

$$J_{k,l}^U(x) = -\frac{1}{\gamma_t^2}\frac{\Sigma_{l=1}^{n_k}(\frac{\partial w_{k,l}(x)}{\partial U_{k,l}}\delta_{k,l}(x) + w_{k,l}(x)\frac{\partial \delta_{k,l}(x)}{\partial U_{k,l}})w_k(x) - (\frac{\partial w_k(x)}{\partial U_{k,l}})\Sigma_{l=1}^{n_k}w_{k,l}(x)\delta_{k,l}(x)}{w_k^2(x)}$$

$$= -\frac{1}{\gamma_t^2}\frac{\Sigma_{l=1}^{n_k}w_{k,l}(x)\frac{\partial \delta_{k,l}(x)}{\partial U_{k,l}}}{w_k(x)}$$

$$\approx \frac{1}{\gamma_t^2}\frac{s_t^2}{s_t^2+\gamma_t^2}\ r_{k,l}(x)\left[U_{k,l}(x-\mu_{k,l})^\top\ +\ (x-\mu_{k,l})^\top U_{k,l}I\right].$$

And

$$H_{U_kU_k} \approx diag(\mathbb{E}[r_{k,1}^2]H_{U_{k,1}U_{k,1}}, \mathbb{E}[r_{k,2}^2]H_{U_{k,2}U_{k,2}}, \ldots, \mathbb{E}[r_{k,n_k}^2]H_{U_{k,n_k}U_{k,n_k}}),$$

where

$$H_{U_{k,l}U_{k,l}} = \mathbb{E}[J_{k,l}^U(x)(J_{k,l}^U(x))^\top]$$

$$= \mathbb{E}[(\frac{\alpha}{\gamma_t^2})^2\left(U_{k,l}(x-\mu_{k,l})^\top(x-\mu_{k,l})U_{k,l}^\top + U_{k,l}^\top(x-\mu_{k,l})U_{k,l}(x-\mu_{k,l})^\top\right)]$$

$$+ \mathbb{E}[(\frac{\alpha}{\gamma_t^2})^2\left(U_{k,l}^\top(x-\mu_{k,l})(x-\mu_{k,l})U_{k,l}^\top + (U_{k,l}^\top(x-\mu_{k,l}))^2\right)].$$

Similar to our calculation in B.4, we can use B.3 to calculate the minimum eigenvalue of $H_{U_{k,l}U_{k,l}}$.

$H_{U_{k,l}U_{k,l}}$ is positive definite and

$$\lambda_{\min}(H_{U_{k,l}U_{k,l}}) = \frac{4(U_{k,l}^\top\mu_{k,l})^2 + \|U_{k,l}\|_2^2\|\mu_{k,l}\|_2^2 - \|U_{k,l}\|_2\|\mu_{k,l}\|_2\sqrt{8(U_{k,l}^\top\mu_{k,l})^2 + \|U_{k,l}\|_2^2\|\mu_{k,l}\|_2^2}}{2}.$$

Recall that

$$H_{U_kU_k} \approx diag(\mathbb{E}[r_{k,1}^2]H_{U_{k,1}U_{k,1}}, \mathbb{E}[r_{k,2}^2]H_{U_{k,2}U_{k,2}}, \ldots, \mathbb{E}[r_{k,n_k}^2]H_{U_{k,n_k}U_{k,n_k}}).$$

and $\mathbb{E}[r_{k,l}^2] \approx \pi_{k,l}$, we can obtain the minimum eigenvalue of $H_{U_kU_k}$, which is

$$\min_{l=1,2,\ldots,n_k} \pi_{k,l}\frac{4(U_{k,l}^\top\mu_{k,l})^2 + \|U_{k,l}\|_2^2\|\mu_{k,l}\|_2^2 - \|U_{k,l}\|_2\|\mu_{k,l}\|_2\sqrt{8(U_{k,l}^\top\mu_{k,l})^2 + \|U_{k,l}\|_2^2\|\mu_{k,l}\|_2^2}}{2}.$$

∎

**Lemma C.4.** *[Local Strong Convexity] Assume Theorem 6.4, in a neighborhood of $\theta^*$, $\nabla^2 \mathcal{L}(\theta) \succeq \alpha' I, \alpha' > 0, \forall \theta \in \Theta$. If $\forall x \in \mathbb{R}^{d_k}, \exists l \in [n_k], r_{k,l}(x) = 1$ are strictly satisfied, $\alpha' = \min\{\lambda_1, \lambda_2\}$, where $\lambda_1 = \min_{l=1\dots,n_k} \frac{c_{k,l}\gamma_t^4}{(s_t^2+\gamma_t^2)^2}$, $\lambda_2 = \min_{l=1,2,\dots,n_k} = \lambda_{\min}(H_{U_{k,l}U_{k,l}})$.*

**Proof.**

$$H_{\mu_k U_k} = diag(H_{\mu_{k,1}U_{k,1}}, H_{\mu_{k,2}U_{k,2}}, \dots, H_{\mu_{k,1n_k}U_{k,n_k}}).$$

$$\|H_{\mu_k U_k}\| \leq \sqrt{\|H_{\mu_k \mu_k}\| \|H_{U_k U_k}\|} = O\Big(\frac{s_t^3}{\gamma_t^2 (s_t^2 + \gamma_t^2)^2}\Big).$$

$$H = \begin{pmatrix} \text{diag}\big(H_{\mu_{k,1}\mu_{k,1}}, \dots, H_{\mu_{k,n_k}\mu_{k,n_k}}\big) & \text{diag}\big(H_{\mu_{k,1}U_{k,1}}, \dots, H_{\mu_{k,n_k}U_{k,n_k}}\big) \\ \text{diag}\big(H_{\mu_{k,1}U_{k,1}}, \dots, H_{\mu_{k,n_k}U_{k,n_k}}\big) & \text{diag}\big(H_{U_{k,1}U_{k,1}}, \dots, H_{U_{k,n_k}U_{k,n_k}}\big) \end{pmatrix}.$$

Let

$$S = H_{\mu\mu} - H_{\mu U} H_{UU}^{-1} H_{U\mu}$$

we have

$$\lambda_H \geq \lambda_S \geq \lambda_{H_{\mu_k \mu_k}} - \frac{r^2 \lambda_{H_{\mu_k \mu_k}} \lambda_{H_{U_k U_k}}}{\lambda_{H_{U_k U_k}}} = (1-r^2)\lambda_{H_{\mu_k \mu_k}} \geq (1-r^2)\frac{s_t^2}{(s_t^2+\gamma_t^2)^2} > 0.$$

$$r = \max_{\|u\|=1,\|v\|=1} \frac{u^\top H_{\mu_k U_k} v}{\sqrt{u^\top H_{\mu_k \mu_k} u \cdot v^\top H_{U_k U_k} v}} \leq 1.$$

$r = 1$ if and only if $u^\top J_\mu^k = cv^\top J_U^k$, $c \neq 0$, which is almost impossible to happen.

More specifically, if we assume that $\forall x \in \mathbb{R}^{d_k}, \exists l \in [n_k], r_{k,l}(x) = 1$, we have

$$H_{\mu_{k,l}U_{k,l}} = \mathbb{E}_{x \sim p_k}\left[J_{k,l}^U(x)\left(J_{k,l}^\mu(x)\right)^\top\right]$$
$$= \frac{1}{\gamma_t^4}\frac{s_t^3}{s_t^2+\gamma_t^2} \mathbb{E}_{x \sim p_k}\left[r_{k,l}(x)^2((x-\mu_{k,l})U_{k,l}^\top + (x-\mu_{k,l})^\top U_{k,l}I)\right]\left(I - \frac{s_t^2}{s_t^2+\gamma_t^2}U_{k,l}U_{k,l}^\top\right)$$
$$= \frac{1}{\gamma_t^4}\frac{s_t^3}{s_t^2+\gamma_t^2} \mathbb{E}_{x \sim \pi_{k,l}\mathcal{N}_{k,l}}\left[r_{k,l}(x)^2((x-\mu_{k,l})U_{k,l}^\top + (x-\mu_{k,l})^\top U_{k,l}I)\right]\left(I - \frac{s_t^2}{s_t^2+\gamma_t^2}U_{k,l}U_{k,l}^\top\right)$$
$$\approx 0$$

The second equation holds because $\forall x$, if $x \notin \mathcal{N}_{k,l}(\mu_{k,l}, \Sigma_{k,l})$, $r_{k,l}(x) = 0$. And the third equation holds because if $x \sim \mathcal{N}_{k,l}, (\mu_{k,l}, \Sigma_{k,l}), \forall \text{ Const } C$,

$$\mathbb{E}_{x \sim \pi_{k,l}\mathcal{N}_{k,l}}[C(x - \mu_{k,l})] = 0.$$

.

Thus, let $\alpha'$ be the minimum eigenvalue of $H$,

$$\alpha' = \min\{\lambda_1, \lambda_2\}, \tag{8}$$

where

$$\lambda_1 = \min_{l=1\dots,n_k} \frac{c_{k,l}\gamma_t^4}{(s_t^2+\gamma_t^2)^2},$$

and

$$\lambda_2 = \min_{l=1,2,\dots,n_k} \pi_{k,l} \frac{4(U_{k,l}^\top \mu_{k,l}))^2 + \|U_{k,l}\|_2^2\|\mu_{k,l}\|_2^2 - \|U_{k,l}\|_2\|\mu_{k,l}\|_2\sqrt{8(U_{k,l}^\top \mu_{k,l}))^2 + \|U_{k,l}\|_2^2\|\mu_{k,l}\|_2^2}}{2}.$$

$\blacksquare$

# D  EXTENSION TO MOG LATENT WITHOUT SEPARATION ASSUMPTION

## D.1  2-MODE ANALYSIS

In this section, we relax the high separation assumption (where $r_k^+(x)r_k^-(x) \approx 0$). Instead, we treat the overlap between manifold components as a bounded perturbation to the ideal system. We aim to prove that the Hessian remains positive definite provided the overlap factor is sufficiently small.

### D.1.1  DEFINITION OF OVERLAP FACTOR

We define the pointwise overlap factor $\xi_k(x)$ as the product of the assignment probabilities for the positive and negative components of the $k$-th manifold:

$$\xi_k(x) \triangleq r_k^+(x)r_k^-(x). \tag{9}$$

Since $r_k^+(x), r_k^-(x) \in [0,1]$ and $r_k^+(x) + r_k^-(x) = 1$, the overlap factor is naturally bounded: $0 \le \xi_k(x) \le 0.25$.

We denote the maximum expected overlap magnitude as $\epsilon_{\text{overlap}}$:

$$\epsilon_{\text{overlap}} = \sup_{x \in \text{supp}(p_t)} \xi_k(x). \tag{10}$$

### D.1.2  JACOBIAN ANALYSIS

We revisit the derivation of the Jacobian $J_k^\mu$. In the original derivation, $J_k^\mu$ was decomposed into Term A (dominant term) and Term B (previously ignored):

$$J_k^\mu(x) = \underbrace{J_{\text{ideal}}^\mu(x)}_{\text{Term A}} + \underbrace{E^\mu(x)}_{\text{Term B}}.$$

When $\xi_k(x) \to 0$, we can recover the ideal Jacobian derived previously:

$$J_{\text{ideal}}^\mu(x) = -\frac{s_t}{\gamma_t^2}(r_k^+(x) - r_k^-(x))\left(I - \frac{s_t^2}{s_t^2 + \gamma_t^2}U_k U_k^\top\right).$$

Term B contains the cross-product of weights, which is exactly our overlap factor $\xi_k(x)$. Specifically:

$$E^\mu(x) = -\frac{4s_t^2}{\gamma_t^2 w_k^2(x)} \cdot \xi_k(x) \cdot \Sigma_k^{-1} x \left(I + \frac{s_t^2}{s_t^2 + \gamma_t^2}U_k U_k^\top\right)\mu_k.$$

We can bound the norm of this error term. Since terms like $\frac{x}{w_k(x)}$ and projection matrices are bounded within the support, there exists a constant $C_1$ such that:

$$\|E^\mu(x)\|_2 \le C_1 \cdot \xi_k(x). \tag{11}$$

Similarly, for the Jacobian with respect to $U_k$, we can decompose it into an ideal part and an error part proportional to the overlap:

$$J_k^U(x) = J_{\text{ideal}}^U(x) + E^U(x), \quad \text{where } \|E^U(x)\|_F \le C_2 \cdot \xi_k(x).$$

### D.1.3  HESSIAN ANALYSIS

The Hessian matrix $H$ is defined as the expected outer product of the Jacobians:

$$H = \mathbb{E}_{x \sim p_t(x)}[J(x)J(x)^\top].$$

Let $J(x) = J_{\text{ideal}}(x) + E(x)$. Substituting this into the Hessian definition:

$$H = \mathbb{E}\left[(J_{\text{ideal}} + E)(J_{\text{ideal}} + E)^\top\right]$$
$$= \underbrace{\mathbb{E}[J_{\text{ideal}} J_{\text{ideal}}^\top]}_{H_{\text{ideal}}} + \underbrace{\mathbb{E}[J_{\text{ideal}} E^\top + E J_{\text{ideal}}^\top + E E^\top]}_{\Delta H}.$$

Here, $H_{\text{ideal}}$ is the Hessian matrix under the high separation assumption and $\Delta H$ is the perturbation matrix induced by the overlap.

From the previous proof , we established that $H_{\text{ideal}}$ is block-diagonal (or has negligible off-diagonals due to symmetry) and positive definite. Let $\alpha > 0$ be its minimum eigenvalue:

$$\lambda_{\min}(H_{\text{ideal}}) \approx \mathbb{E}[(r_k^+(x) - r_k^-(x))^2] \min\left(\lambda_{\min}(H_{\mu_k \mu_k}), \lambda_{\min}(H_{U_k U_k})\right)$$
$$= \mathbb{E}[(1 - 4\xi_k(x))] \min\left(\lambda_{\min}(H_{\mu_k \mu_k}), \lambda_{\min}(H_{U_k U_k})\right)$$
$$\geq (1 - 4\epsilon_{\text{overlap}}) \min\left(\lambda_{\min}(H_{\mu_k \mu_k}), \lambda_{\min}(H_{U_k U_k})\right) \triangleq \alpha.$$

We apply the Triangle Inequality and Cauchy-Schwarz inequality to bound the spectral norm of $\Delta H$:

$$\|\Delta H\|_2 \leq 2\|\mathbb{E}[J_{\text{ideal}} E^\top]\|_2 + \|\mathbb{E}[E E^\top]\|_2$$
$$\leq 2\sqrt{\mathbb{E}[\|J_{\text{ideal}}\|^2] \mathbb{E}[\|E\|^2]} + \mathbb{E}[\|E\|^2].$$

Since $\|E^\mu(x)\| \leq C_1 \cdot \xi_k(x)$ and $\|E^U(x)\| \leq C_2 \cdot \xi_k(x)$, the perturbation norm is dominated by the overlap factor:

$$J_k^U(x) = J_{\text{ideal}}^U(x) + E^U(x), \quad \text{where } \|E^U(x)\|_F \leq C_2 \cdot \xi_k(x).$$

The Hessian perturbation matrix is given by $\Delta H \approx \mathbb{E}[J_{\text{ideal}} E^\top + E J_{\text{ideal}}^\top]$. To bound its spectral norm $\|\Delta H\|_2$, we define the signal bounds

$$S_\mu \triangleq \sup_x \|J_{\text{ideal}}^\mu(x)\|_2 \approx \frac{s_t}{\gamma_t^2}$$

and

$$S_U \triangleq \sup_x \|J_{\text{ideal}}^U(x)\|_2 \approx \frac{s_t R^2}{\gamma_t^2}.$$

We can define the composite perturbation constant $C'$ as:

$$C' = 2(S_\mu + S_U)(C_1 + C_2).$$

And thus,

$$\|\Delta H\|_2 \leq C' \cdot \epsilon_{\text{overlap}}.$$

### D.1.4 POSITIVE DEFINITENESS VIA WEYL'S INEQUALITY

We now use Matrix Perturbation Theory to prove the convexity of the actual loss landscape. With Weyl's Inequality for Hermitian Matrices, we have: Let $H = H_{\text{ideal}} + \Delta H$. The eigenvalues of $H$ are bounded by:

$$\lambda_{\min}(H) \geq \lambda_{\min}(H_{\text{ideal}}) - \|\Delta H\|_2. \tag{12}$$

Substituting our bounds:

$$\lambda_{\min}(H) \geq \alpha - C' \cdot \epsilon_{\text{overlap}}. \tag{13}$$

**Condition for Convexity:** For the Hessian $H$ to remain positive definite (ensuring strong convexity), we require:

$$\alpha - C' \cdot \epsilon_{\text{overlap}} > 0 \implies \epsilon_{\text{overlap}} < \frac{\alpha}{C'}. \tag{14}$$

This physically implies that as long as the manifolds are not excessively overlapping , the loss function remains locally strongly convex.

### D.1.5  CONVERGENCE ANALYSIS

Based on the perturbation analysis, we state the revised convergence theorem.

**Theorem D.1** (Linear Convergence under Bounded Overlap). *Let $L(\theta)$ be the loss function. Assume the overlap factor satisfies $\epsilon_{overlap} < \frac{\alpha}{C'}$. Then, the Hessian $H$ at $\theta^\star$ is positive definite with minimum eigenvalue:*

$$\lambda_{\min}(H) \geq \alpha_{eff} = \alpha - C'\epsilon_{overlap} > 0.$$

*Consequently, gradient descent with step size $\eta$ converges linearly:*

$$\|\theta^t - \theta^\star\|_2 \leq \left(\frac{\kappa_{eff} - 1}{\kappa_{eff} + 1}\right)^t \|\theta^{(0)} - \theta^\star\|_2,$$

*where the effective condition number is degraded by the overlap:*

$$\kappa_{eff} = \frac{L}{\alpha - C'\epsilon_{overlap}}.$$

**Proof.** The proof follows directly from the strong convexity of $L(\theta)$ established by Weyl's inequality. As $\epsilon_{\text{overlap}} \to 0$, we recover the ideal convergence rate. ∎

### D.2  MULTI-MODE ANALYSIS

In this section, we analyze the convergence properties for the mutli-Mode Mixture of Gaussians model. We explicitly model the **overlap** between Gaussian components as a perturbation.

### D.2.1  THE OVERLAP FACTOR

We formally define the **Pairwise Overlap Factor** $\xi_{i,j}(x)$ between two components $i$ and $j$:

$$\xi_{i,j}(x) \triangleq r_{k,i}(x)r_{k,j}(x). \tag{15}$$

And we define the **Maximum Expected Overlap** $\epsilon_{\text{overlap}}$ for the manifold as:

$$\epsilon_{\text{overlap}} = \max_i \sum_{j \neq i} \mathbb{E}_{x \sim p_t}[\xi_{i,j}(x)]. \tag{16}$$

This scalar $\epsilon_{\text{overlap}}$ quantifies the deviation from the ideal high separation regime. If components are perfectly separated, $\xi_{i,j} \to 0$ and $\epsilon_{\text{overlap}} \to 0$.

### D.2.2  JACOBIAN DERIVATION

We need to compute the Jacobian of the score matching error vector $s_\theta(x,t) - \nabla \log p_t(x)$ with respect to the parameter $\mu_{k,l}$. Let $J_l^\mu(x) = \frac{\partial}{\partial \mu_{k,l}} \nabla \log p_{t,k}(x)$.

Similarly, we decompose the Jacobian for the $l$-th component into a **Signal Term** (Self) and a **Noise Term** (Interference).

$$J_\mu^l(x) = \underbrace{J_{\mu,\text{ideal}}^l(x)}_{\text{Signal}} + \underbrace{E_{\mu,\text{cross}}^l(x)}_{\text{Noise}}.$$

This term arises when we ignore the change in weights of other clusters ($j \neq l$). It dominates when $r_{k,l} \approx 1$:

$$J_{\mu,\text{ideal}}^l(x) \approx -\frac{s_t}{\gamma_t^2} r_{k,l}(x) \left( I - \frac{s_t^2}{s_t^2 + \gamma_t^2} U_{k,l} U_{k,l}^\top \right).$$

This term captures the gradient leaking into other clusters due to overlap:

$$E_{\mu,\text{cross}}^l(x) = \sum_{j=1}^{n_k} C_1'(x) \cdot \underbrace{r_{k,j}(x) r_{k,l}(x)}_{\xi_{j,l}(x)}, \tag{17}$$

where $C_1'(x)$ collects bounded vector terms. The norm of the error term is strictly bounded by the overlap:

$$\|E_{\mu,\text{cross}}^l(x)\|_2 \leq C_1' \sum_{j \neq l} \xi_{j,l}(x).$$

For the Jacobian with respect to $U_k$, we have Similar derivation.

$$\|E_{U,\text{cross}}^l(x)\|_2 \leq C_2' \sum_{j \neq l} \xi_{j,l}(x).$$

### D.2.3   HESSIAN BLOCK STRUCTURE

The Hessian $H$ for the parameters $\boldsymbol{\mu} = [\mu_{k,1}, \ldots, \mu_{k,n_k}]$ is a block matrix composed of $n_k \times n_k$ blocks, where each block is $D \times D$.

$$H_{\boldsymbol{\mu\mu}} = \begin{pmatrix} H_{1,1} & H_{1,2} & \cdots & H_{1,n_k} \\ H_{2,1} & H_{2,2} & \cdots & H_{2,n_k} \\ \vdots & \vdots & \ddots & \vdots \\ H_{n_k,1} & H_{n_k,2} & \cdots & H_{n_k,n_k} \end{pmatrix}.$$

The $(i,j)$-th block is defined as:

$$H_{i,j} = \mathbb{E}_x[J_i^\mu(x)(J_j^\mu(x))^\top].$$

For diagonal blocks ($i = j = l$), the curvature is strictly determined by the expectation of the squared weights $\mathbb{E}[r_{k,l}(x)^2]$. Crucially, overlap causes **signal attenuation**, as the weight $r_{k,l}(x)$ drops below 1 in transition regions.

Using the identity $r_{k,l}(x)^2 = r_{k,l}(x)(1 - \sum_{j \neq l} r_{k,j}(x))$, we derive the exact expectation:

$$\mathbb{E}[r_{k,l}(x)^2] = \mathbb{E}[r_{k,l}(x)] - \sum_{j \neq l} \mathbb{E}[r_{k,l}(x) r_{k,j}(x)]$$

$$= \pi_{k,l} - \sum_{j \neq l} \mathbb{E}[\xi_{j,l}(x)]$$

$$= \pi_{k,l} - \epsilon_{k,l}^{\text{total}}.$$

Thus, we lower-bound the diagonal curvature by accounting for the total overlap mass $\epsilon_{k,l}^{\text{total}}$ leaking from cluster $l$:

$$H_{l,l} \approx \mathbb{E}[(J_l^{\text{ideal}})(J_l^{\text{ideal}})^\top] \succeq \lambda_{\text{diag,l}} \cdot I,$$

where the effective base curvature is:

$$\lambda_{\text{diag,l}} = (\pi_{k,l} - \epsilon_{k,l}^{\text{total}}) \min\left(\lambda_{\min}(H_{\mu_{k,l}\mu_{k,l}}), \lambda_{\min}(H_{U_{k,l}U_{k,l}})\right)$$

Here, the term $(\pi_{k,l} - \epsilon_{k,l}^{\text{total}})$ represents the effective probability mass contributing to convexity. This formulation explicitly shows that smaller clusters (small $\pi_{k,l}$) are significantly more vulnerable to instability, as the effective mass can vanish if the overlap $\epsilon_{k,l}^{\text{total}}$ becomes comparable to the cluster size $\pi_{k,l}$.

For $i \neq j$, the block $H_{i,j}$ represents the interference.

$$H_{i,j} \approx \mathbb{E}_x[J_i^{\text{ideal}}(J_j^{\text{ideal}})^\top] \propto \mathbb{E}[r_{k,i}(x)r_{k,j}(x)].$$

### D.2.4 PERTURBATION ANALYSIS

We write the full Hessian as a sum of a block-diagonal matrix and a perturbation matrix:

$$H_{\boldsymbol{\mu\mu}} = H_{\text{diag}} + \Delta H_{\text{overlap}}.$$

**For the minimum eigenvalue of $H_{\text{diag}}$,**

$$\lambda_{\min}(H_{\text{diag}}) = \min_l \lambda_{\min}(H_{l,l}) = \min_l \lambda_{\text{diag,l}} \triangleq \lambda_{\text{base}}.$$

For **Spectral Norm of $\Delta H_{\text{overlap}}$**, by Weyl's Inequality, the minimum eigenvalue of the full Hessian is:

$$\lambda_{\min}(H) \geq \lambda_{\min}(H_{\text{diag}}) - \|\Delta H_{\text{overlap}}\|_2.$$

and

$$\Delta H_{\text{overlap}} \leq \tilde{C} \cdot \mathbb{E}[\xi_{i,j}(x)], \tag{18}$$

where

$$\tilde{C} = 2\left(S_\mu C_1' + S_U C_2'\right)$$

Substituting the bounds:

$$\lambda_{\min}(H) \geq \lambda_{\text{base}} - \tilde{C} \cdot \epsilon_{\text{overlap}}.$$

Therefore, $H$ is positive definite **if and only if**:

$$\epsilon_{\text{overlap}} < \frac{\lambda_{\text{base}}}{\tilde{C}}.$$

**Interpretation:** The optimization landscape is locally strictly convex provided the overlap between clusters is smaller than the intrinsic curvature of the individual Gaussians.

### D.2.5 FULL CONVERGENCE THEOREM

Combining the analysis of $\mu$ and the similar decoupling argument for $U$ (using Schur complements to handle $H_{\mu U}$ terms which are also $O(\epsilon)$), we arrive at the final result.

**Theorem D.2.** *Let $\mathcal{L}(\theta)$ be the score matching loss. Assume the maximum expected overlap $\epsilon_{overlap}$ satisfies the condition $\epsilon_{overlap} < \tau$ for some threshold $\tau \propto \lambda_{base}$. Then the Hessian $H(\theta^\star)$ is strictly positive definite.*

*Linear Convergence: Gradient descent with step size $\eta$ converges as:*

$$\|\theta^{(t)} - \theta^{\star}\|_2 \leq \rho^t \|\theta^{(0)} - \theta^{\star}\|_2,$$

*where the convergence rate $\rho < 1$ is determined by the effective condition number:*

$$\kappa_{eff} = \frac{L}{\lambda_{base} - \tilde{C}\epsilon_{overlap}}.$$

This proves that the High Separation Assumption is not a binary requirement, but rather a continuum. The algorithm is robust to finite overlap, with the convergence rate degrading gracefully as the overlap increases.

*Remark* D.3. It is important to note that physically, $\epsilon_{\text{overlap}}$ will not be arbitrarily large.

## E    THE DETAIL OF THE REAL-WORLD EXPERIMENTS

In the part, we provide the detail of the experiments, including dataset and training pipeline. We use MNIST and CIFAR-10 as the datasets, and we adopt the mixture Gaussian distribution as the prior distribution in both cases.

For MNIST, our model consists of MLP-based encoder and decoder networks, each with a single hidden layer of 256 dimensions. The model is trained with the AdamW optimizer at a learning rate of 0.0005. We train 10 VAEs with the numbers 1 to 10 as the ten clusters.

On CIFAR-10, we implement a 3-layer RNN encoder and decoder for CIFAR-10. The encoder hidden dimensions are [64, 128, 256], and the decoder's are [256, 128, 64].And we train 10 VAEs for each of the ten clusters based on the classification by category. Each layer in both networks stacks 3 recurrent blocks.The model is trained with the AdamW optimizer at a learning rate of 0.0001.

Our experiment was conducted on RTX4090.

