# OpenReview forum: "Multi-Subspace Multi-Modal Modeling for Diffusion Models: Estimation, Convergence and Mixture of Experts"
_ICLR.cc/2026/Conference — ICLR 2026 Poster_

### Official Review · Reviewer_fdxV · 2025-10-31

**Soundness:** 1
**Presentation:** 2
**Contribution:** 2
**Rating:** 4
**Confidence:** 4

**Summary:**

This paper extends low-dimensional data modeling (MoLR-MoG) to analyze the training of latent diffusion models. The assumption is that different classes lie near low-dimensional subspaces and can be modeled as mixtures of Gaussians on those subspaces. The network is parameterized to approximate the optimal score, with learnable means and covariances.

Under this setup, the authors establish uniform concentration and Lipschitz continuity of the empirical DSM loss. Using a separability condition, they derive Hessian bounds and prove local strong convexity and linear convergence. The analysis is developed first for a two-Gaussian case and then extended to multiple clusters.

**Strengths:**

1. The paper formalizes an intrinsic low-dimensional structure: after encoding (e.g., with a VAE), the latent distribution becomes MoLR-MoG. This is a natural refinement because VAEs encourage approximately Gaussian latents.
2. The theoretical treatment under the MoLR-MoG assumption is comprehensive, providing concentration, smoothness, and local convergence guarantees.

**Weaknesses:**

1. The implications feel modest: much of the contribution is incremental over existing MoLRG-style assumptions. Moreover there are some possible overstatements:
   1. Claims such as “escaping the curse of dimensionality” follow directly from the assumption rather than the learning procedure.
   2. The “Mixture of Experts” terminology seems overstated; the parameterization behaves closer to a one-layer gated linear unit than an MoE in the LLM sense.
2. **Theory setup and presentation issues.**
   1. The framework appears to require learning $K$ VAEs (one per class/subspace) and then training a MoG score within each subspace. The linear maps $A_k$ are introduced but not further investigated (e.g., not showing up in the Hessian)
   2. This means $A_k$ is used mainly to reduce the problem to learning within each subspace: first, this is not standard and is like training an independent diffusion model for each class; second, does this require the $A_k$ to be orthogonal and have similar dimensions? There seems to be a lot missing here.
   3. To obtain strong convexity and convergence bounds, the analysis effectively turns softmax into hardmax ($r_k \approx 1$), which reduces the problem to (piecewise) linear models. Moreover, the arguments are all local.
3. There is no numerical verification of the convergence behavior or deeper experiments illustrating the intuition and implications of MoLR-MoG.
4. **Lack of Clarity.** Several lemmas are straightforward calculations that could be moved to the appendix; the core setup only becomes clear around pp. 4–5. There are also typos (e.g., “$r_k^+(x)=1\ \text{or}\ 1$”) that should be fixed.

**Questions:**

1. Fig. 1 suggests linear encoders/decoders $A_k$ to project into low-dimensional subspaces, whereas Fig. 2 seems to imply a nonlinear VAE before training. Do you ultimately replace $A_k$ with a nonlinear VAE in the experiments? If so, this differs from the linear $A_k$ setup. Please clarify which setting is analyzed theoretically versus used empirically.
2. Can you provide rates or Hessian eigenvalue bounds without turning soft assignments into hard assignments (i.e., without $r_k \approx 1$)?

---

> ### Author Response · Authors · 2025-11-24
> **Rebuttal Part 1: Larger Real World Experiments with Loss Curve**
>
> Thank you for your valuable comments and suggestions. As a start, we first provide the larger experiments and discussion on them, which is helpful in responding to other questions. Then, we provide our response to each question below. For the revised paper, we highlight the new part in green.
>
> **Weakness 3: Larger Real-world experiments and Quantitative Results (ImageNet256).**
>
> We extend our experiments to the larger ImageNet256 dataset and show that the MoLR-MoG NN, using $ 10\times$ smaller parameters compared with MoLR-Unet (a Comparison of latent diffusion models within the manifold, as it uses the same VAE), can achieve comparable performance. On the contrary, MoLR-Gaussian still produces blurry images due to its limited ability, which stems from the linear latent score of the Gaussian (as shown in Fig. 2 of our revised paper).
>
> (a) The relationship between theory, applications, and our real-world experiments.
>
> Before discussing the results, we first discuss the role of our experiments, the theoretical modeling, and how to scale up the application. In our current experiments, since MNIST and CIFAR-10 have label information, we employ hard clustering instead of soft clustering (soft weights) used in theoretical modeling, which is also adopted by [2]. More specifically, we train a VAE for each class, which can be viewed as a simplification of soft activation, to demonstrate that with a suitable VAE to embed the input images in the correct manifold, a small 2-layer MoG expert is sufficient to generate images.
>
> For large-scale datasets without labels, we can utilize a clustering algorithm to partition the data into distinct clusters. Then, we can train a VAE encoder, decoder, and latent MoG score for each cluster. For the VAE training, we do not require training the VAE from a sketch. We can LoRA fine-tune a VAE pretrained on large-scale datasets (for example, DC-AE [1] for our ImageNet experiments) for each expert, which shares a pretrained VAE backbone and has a smaller model size. When generating images, we activate different VAE LoRA according to the clustering weight (can choose Top1 or Top2 experts), which matches the spirit of MoE.
>
> (b) Experiments Discussion, Converged Loss Comparison and Quantitative Results
>
> As a start, we focus on the parachute class of ImageNet256 (to achieve a good performance with the whole ImageNet256 dataset, we can follow the above process to do clustering, VAE LoRA, and MoG experts training). More specifically, we use $1k$ parachute images to fine-tune the good pretrained DC-AE to help the VAE encode the images to the parachute manifold. Then, we train the Unet, 2-layer SoftMax GMM NN, and linear Gaussian NN with these parachute images. As shown in Fig.2, the generated results of MoLR-MoG are comparable to the generated results of MoLR-Unet.
> From a quantitative perspective, we calculate the CLIP score for the parachute class in ImageNet using the text prompt "a photo of a parachute". The Clip score for MoLR with Unet, MoG, and Gaussian NN is $0.304$, $0.293$, and $0.254$, respectively, indicating that MoLR-MoG achieves almost comparable text-to-image alignment with MoE-Unet. We also provide the loss curves of latent Unet, MoG, and Gaussian, which show that the loss of MoE-MoG NN is significantly smaller than that of MoE-Gaussian and is close to that of MoE-Unet, indicating that MoE-MoG NN efficiently approximates the ground-truth score and supports our theoretical results.
>
> (c) Expert-specific VAE and the comparison with single VAE
>
> In this part, we also highlight the role of expert-specific VAE. As shown in the score of MoLR-MoG, different from latent diffusion models with a single VAE, there are $K$ VAEs to encode the input to the corresponding manifold. We note that this operation is important for MoLR-MoG with small MoG experts. Otherwise, with a unified VAE, the latent space is complex and requires a large latent space, such as Unet. As shown in Fig.3, with a unified VAE, the unified latent is complex, and a MoG expert can not learn a meaningful image with the target class. Hence, with a unified VAE, latent diffusion models require a large latent Unet. However, with an expert-specific VAE (for example, we fine-tune the pretrained VAE with the parachute class dataset), the latent manifold becomes simple, and latent MoG experts are enough to generate clear models, which also supports our theoretical modeling.
> We have now added a detailed results and discussion in the revision of our paper (Sec 4, highlighted in green).

---

> > ### Author Response · Authors · 2025-11-24
> > **Rebuttal Part 2: The Discussion on the Curse of Dimensionality and MoE**
> >
> > **Weakness 1: The Curse of Dimensionality and MoE**
> >
> > (a) Escape the Curse of Dimensionality with our MoLR-MoG
> >
> > As MoLR-MoG modeling (a reasonable modeling) is one of the main contributions in this work, we first show that with this modeling, we can achieve good performance with MoLR-Unet with fewer parameters (fewer parameters always correspond to a better estimation error).
> >
> > Based on the empirical observation, we have the intuition that MoLR-MOG can achieve a good balance in the estimation error and good performance (on the contrary, MoLR-Gaussian, though it has a good estimation error, has a poor performance). As a result, we achieve an estimation error that escapes the curse of dimensionality by taking advantage of the MoLR-MoG setting.
> >
> > (b) The MoE structure for MoLR-MoG
> >
> > In weakness 1 (a), we have provided how to run our MoE structure in the training and inference phase. In this part, we show why the score of MoLR-MoG satisfied the definition of LLM.
> >
> > The standard MoE form $\sum_k g_k(x)E_k(x)$ contains: $g_k(x)$ plays the role of an input-dependent gating network, and each expert $E_k(x)$ specializes in a different latent manifold and its associated MoG structure. Our score function indeed corresponds to a Mixture-of-Experts (MoE) architecture with $K$ experts, each expert being a latent Mixture-of-Gaussians (MoG) score. More specifically, we first group the mixture over the expert index $k$ by defining the marginal density of expert $k$,
> >
> > $$
> > q_k(x):= \frac{1}{K} \sum\_{l=1}^{n_k} \pi\_{k,l} \mathcal{N}(x; s\_t \mu\_{k,l}^{\*} A\_k^{\*}, A\_k\^{\*} \Sigma_{k,l,t,A} A\_k^{\*T}),
> > $$
> >
> > and the corresponding gating weight
> >
> > $$
> > g_k(x)
> > := \frac{q_k(x)}{\sum_{k'=1}^K q_{k'}(x)}.
> > $$
> > Within each expert $k$, we normalize over the MoG components and define the intra-expert responsibilities
> > $$
> > \alpha\_{k,l}(x)
> > := \frac{\pi_{k,l}\,\mathcal{N}\big(x;\,s_t\mu_{k,l}^{\*}A_k^{\*},\,A_k^{\*}\Sigma_{k,l,t,A}A_k^{\*\top}\big)}
> > {\sum_{l'=1}^{n_k}\pi_{k,l'}\,\mathcal{N}\big(x;\,s_t\mu_{k,l'}^{\*}A_k^{\*},\,A_k^{\*}\Sigma_{k,l',t,A}A_k^{\*\top}\big)},
> > $$
> >
> > and the expert output
> >
> > $$
> > E_k(x)
> > := \sum_{l=1}^{n_k}\alpha_{k,l}(x)\,\delta_{k,l,t,A}(x),
> > $$
> >
> > which is precisely the latent nonlinear MoG score on the $k$-th manifold (the $k$-th manifold). With these definitions, the score can be rewritten as
> >
> > $$
> > \nabla\log p_t(x)
> > = -\frac{1}{\gamma_t^2}\sum_{k=1}^K g_k(x)\,E_k(x),
> > $$
> >
> > which matches the standard MoE form in LLM.
> >
> >
> >
> > **Weakness 2.1 & 2.2: The Discussion on $K$ VAE.**
> >
> > Thanks for the helpful comments on VAEs. The $A_k$ is the orthonormal basis for the $k$-th manifold, and we do not require that $d_k$ is exactly the same for each manifold. As shown in [3], the different manifold has different $d_k$. We have updated these discussions in our revised paper.
> >
> > *The Analysis for VAEs.*
> >
> > Many empirical works directly use frozen pretrained VAE (such as SD-VAE) and optimize the diffusion part in the latent space. In this work, we also adopt this operation in empirical and theoretical perspectives. From the empirical, we first fine-tune the pretrained VAE to locate the corresponding manifold (as shown in Weakness 1). Then, we freeze the VAE and train the diffusion model part in the latent manifold. From the theoretical perspective, we focus on the optimization dynamics in the latent space. We also note that even considering only the latent space, our analysis is more complex compared with previous works since it involves $\mu$ and $U$.

---

> > > ### Author Response · Authors · 2025-11-24
> > > **Rebuttal Part 3: Extention to Results without highly separated Gaussians**
> > >
> > > **Weakness 2.3 & Q2: Extended  Theoretical Results without highly separated Gaussians**
> > >
> > > Thanks for the helpful and constructive comments to extend the analysis scope of our work. We now extend our results to the latent MoG with overlap from almost independent setting. More specifically, we define the pairwise overlap factor $\xi_{i,j}(x)$ between two components $i$ and $j$
> > >
> > > $$
> > > \xi_{i,j}(x) \triangleq r_{k,i}(x) r_{k,j}(x)\,.
> > > $$
> > >
> > > and the Maximum Expected Overlap $\epsilon_{\text{overlap}}$ for the manifold as:
> > >
> > > $$
> > > \epsilon\_{\text{overlap}} = \max\_{i} \sum\_{j \neq i} \mathbb{E}\_{x \sim p_t} [\xi_{i,j}(x)]\,.
> > > $$
> > >
> > > We start from $2$-mode MoG latent as a start to show our intuition (the formal proof for $2$-mode and $K$-mode has been provided in the end of Sec. 6 and Appendix E, highlighted in green). Intuitively speaking, we can bound the additional negative influence in the Hessian introduced by overlap. Then, we can use  Hessian Perturbation Analysis to obtain the smallest eigenvalue under this setting, which leads to strongly convex.
> > >
> > > Similar to the process of our main content, we first revisit the derivation of the Jacobian $J_k^\mu$ ($J_k^U$ is similar). In the original derivation, $J_k^\mu$ was decomposed into Term A (dominant term) and Term B (previously ignored term introduced by the overlap):
> > >
> > > $$
> > > J_k^\mu(x) = \underbrace{\text{Term A}(x)}\_{J\_{\text{ideal}}^\mu(x)} + \underbrace{\text{Term B}(x)}\_{E^\mu(x)}.
> > > $$
> > >
> > > When $\xi_k(x) \to 0$, we recover the ideal Jacobian derived previously:
> > >
> > > $$
> > > J_{\text{ideal}}^\mu(x) = -\frac{s_t}{\gamma_t^2}(r_k^{+}(x)-r_k^{-}(x))\left(I-\frac{s_t^2}{s_t^2+\gamma_t^2}U_kU_k^\top\right).
> > > $$
> > > The term previously ignored (Term B) contains the cross-product of weights, which is exactly our overlap factor $\xi_k(x)$. Specifically:
> > > $$
> > > E^\mu(x) = -\frac{4 s_t^2}{\gamma_t^2 w_k^2(x)} \cdot \xi_k(x) \cdot \Sigma_k^{-1}x \left(I+\frac{s_t^2}{s_t^2+\gamma_t^2}U_kU_k^\top\right)\mu_k,
> > > $$
> > >
> > > where $w_k(x)=\Sigma_{l=1}^{n_k}\pi_{k,l}\mathcal{N}(x;s_t\mu_{k,l},\Sigma_{k,l})$. Since terms like $\frac{x}{w_k(x)}$ and projection matrices are bounded within the support, there exists a constant $C_1$ such that:
> > >
> > > $$
> > > \|E^\mu(x)\|\_2 \leq C_1 \cdot \xi_k(x).
> > > $$
> > >
> > > Similarly, for the Jacobian with respect to $U_k$, we can decompose it into an ideal part and an error part proportional to the overlap:
> > >
> > > $$
> > > J_k^U(x) = J\_{\text{ideal}}^U(x) + E^U(x), \quad \text{where } \|E^U(x)\|\_F \leq C_2 \cdot \xi\_k(x).
> > > $$
> > >
> > > The Hessian matrix $H$ is defined as the expected outer product of the Jacobians:
> > >
> > > $$
> > > H = \mathbb{E}_{x \sim p_t(x)} [J(x)J(x)^\top].
> > > $$
> > >
> > > Let $J(x) = J_{\text{ideal}}(x) + E(x)$. Substituting this into the Hessian definition:
> > >
> > > $$
> > > \begin{aligned}
> > > H = \mathbb{E} \left[ (J\_{\text{ideal}} + E)(J\_{\text{ideal}} + E)^\top \right] = \underbrace{\mathbb{E}[J\_{\text{ideal}}J\_{\text{ideal}}^\top]}_{H\_{\text{ideal}}} + \underbrace{\mathbb{E}[J\_{\text{ideal}}E^\top + E J\_{\text{ideal}}^\top + E E^\top]}\_{\Delta H}\,,
> > > \end{aligned}
> > > $$
> > >
> > > Where $H_{\text{ideal}}$ is the Hessian matrix under the high separation assumption and $\Delta H$ is the perturbation matrix induced by the overlap.
> > >
> > > From the previous proof (assuming separation), we established that $H_{\text{ideal}}$ is block-diagonal (or has negligible off-diagonals due to symmetry) and positive definite. Let $\alpha > 0$ be its minimum eigenvalue:
> > >
> > > $$
> > > \lambda_{\min}(H_{\text{ideal}}) \geq {(1-4\epsilon_{\text{overlap}})}\min{(\lambda_{\min}(H_{\mu_{k}\mu_{k}}),\lambda_{\min}(H_{U_{k}U_{k}}))} \triangleq \alpha,
> > > $$
> > > where $\lambda_{\min}(H_{\mu_{k}\mu_{k}})$ and $\lambda_{\min}(H_{U_{k}U_{k}})$ has been provided  in Lemma 6.7.
> > >
> > > For $\Delta H$, we know that
> > > $$
> > > \begin{aligned}
> > > \\|\Delta H\\|_2 \leq 2\\|\mathbb{E}[J\_{\text{ideal}}E^\top]\\|_2 + \\|\mathbb{E}[EE^\top]\\|_2 \leq 2 \sqrt{\mathbb{E}[\|J\_{\text{ideal}}\|^2] \mathbb{E}[\|E\|^2]} + \mathbb{E}[\|E\|^2].
> > > \end{aligned}
> > > $$
> > >
> > > Following the above calculation, the perturbation norm is dominated by the overlap factor:
> > >
> > > $$
> > > \\|\Delta H\\|_2 \leq C_3 \cdot \epsilon\_{\text{overlap}}
> > > $$
> > >
> > > where $C_3$ is a constant depending on the bounds of the Jacobians.
> > >
> > > *Positive Definiteness via Weyl's Inequality*
> > >
> > > With Weyl's Inequality for Hermitian Matrices, we have:
> > > Let $H = H_{\text{ideal}} + \Delta H$. The eigenvalues of $H$ are bounded by  ($C_3$ has been provided in Lem. 6.11 and 6.12):
> > >
> > > $$
> > > \lambda_{\min}(H) \geq \lambda_{\min}(H_{\text{ideal}}) - \|\Delta H\|\_2 \geq \alpha - C_3 \cdot \epsilon_{\text{overlap}}.
> > > $$
> > >
> > >
> > > where require:
> > >
> > > $$\alpha - C_3 \cdot \epsilon_{\text{overlap}} > 0 \implies \epsilon_{\text{overlap}} < \frac{\alpha}{K}.$$
> > >
> > > We note that though the above analysis goes beyond the separation lemma, we still require $\epsilon*{\text{overlap}}$ to be small enough. From the empirical perspective, we test the $\xi*{i,j}$ under the ImageNet256 (parachute) and show that $\xi_{i,j}$ is smaller than $0.001$, which is small.

---

> > > > ### Author Response · Authors · 2025-11-24
> > > > **Rebuttal Part 4: The presentation and Question**
> > > >
> > > > **Weakness 4: The Presentation and Clarity.**
> > > >
> > > > Thanks for the constructive comments on our presentation. In our revision paper, we have (a) removed the Lemma about the calculation of Jacobian and Hessian to the appendix and provided more experimental results, discussion, and proof sketch; (b) by compressing the introduction and related work, we have presented our MoLR-MoG earlier in the main content; (c) we have fixed typos.
> > > >
> > > > **Q1: The Linear and Nonlinear VAE.**
> > > >
> > > > In the theoretical part, similar to a series of theoretical works for diffusion models, we also adopt the linear encoder and decoder in our analysis to make a step toward explaining the good performance of diffusion models. In the real-world experiments, we adopt nonlinear VAEs to achieve a good performance. We have added the above discussion in our experiments part.
> > > >
> > > > [1] Chen, Junyu, et al. "Deep Compression Autoencoder for Efficient High-Resolution Diffusion Models." *The Thirteenth International Conference on Learning Representations*.
> > > >
> > > > [2]Wang, Peng, et al. "Diffusion models learn low-dimensional distributions via subspace clustering." *The Second Conference on Parsimony and Learning (Recent Spotlight Track)*.
> > > >
> > > > [3] Brown, Bradley CA, et al. "Verifying the Union of Manifolds Hypothesis for Image Data." *The Eleventh International Conference on Learning Representations*.

---

> ### Comment · Reviewer_fdxV · 2025-11-26
>
> I thank the authors for their reply and for their effort in taking mode overlapping into consideration. However, I still retain a few concerns:
> 1. To me, this modeling does not imply much further insights beyond the convergence loss, given that the model assumption is not standard or practical. Also, is it correct that one must assume all $A_k$ are orthogonal to each other, therefore consider the training on the MoG in each subspace?
> 2. Regarding the "MoE" wording in the title: although the assumed model acts like an MoE layer in an LLM, this appears to be a coincidence regarding the optimal score form. Since MoE is not standard in image diffusion, and you are not studying LLMs here, I feel the term should not be advertised as a primary contribution.
> 3. I am not entirely convinced by the ImageNet results in Figure 2.
>
> I have decided to maintain my current rating, but I am willing to lower my confidence score to reflect the open questions regarding the theoretical assumptions.

---

> > ### Author Response · Authors · 2025-11-26
> > **Data assumption, MoE Form and Further Insights**
> >
> > Dear reviewer fdxV:
> >
> > Thanks for the positive feedback on our extension with overlapping settings. We are more than happy to conduct communication based on further comments and concerns. We provide our response to each question below.
> >
> > **Q1: The Property of $A_k$.**
> >
> > For the property of $A_k$, we said that *"a orthonormal basic matrix  $A_k\in \mathbb{R}^{D \times d_k}$ with orthonormal columns for the $k$-th manifold"*, this do not means $A_k$ is orthonormal to each other $A_i,  i\in [K]$ \ $\\{k\\}$. We means that $A_k$ has orthonormal columns and $A_k^{\top}A_k=I_{d_k}$ (span the $k$-th subspace). The description is the same as [1], and the orthonormal basis $A$ with orthonormal columns has also been adopted by a series of works [2]. We also note that though we do not require $A_k$ to be orthonormal to each other $A_i, i\in [K]$ \ $\\{k\\}$, we still need to train a MoG at each manifold due to the MoE structure. However, since MoG NN is small, it will not require many computational resources and time.
> >
> > We hope our clarification on $A_k$ will help resolve the reviewer's concern.
> >
> > **Q2: The data assumption, MoE Form and Further Insights.**
> >
> > Thanks again for the reviewer's constructive concerns to help us show our insights. We respectfully disagree with the view that *the model assumption is not standard or practical* and *the MoE structure is a coincidence*. As the solid theoretical foundation of the diffusion model is one of the key reasons for its charm, we aim to bridge the gap between theory and practice from a theoretical perspective and make them truly useful in practice.
> >
> > **Our insight** is that with suitable MoLR-MoG modeling (which is closer to *practical* data compared to the current *standard* assumption), we can naturally induce NN with a special structure and enjoy a much smaller number of parameters (to achieve the comparable performance with NN), which provides a new perspective in designing lighter diffusion models. Our estimation error and optimization play the role of the theoretical basis instead of the role of insight. We provide the discussion on data and why MoE is natural instead of coincidence in the following part.
> >
> > (i) The data assumption
> >
> > Our work follows a natural line that (a) first observe the shortcomings of existing theoretical *standard* modeling (the union of low dimensional modeling with Gaussian latent and MoG modeling in the full space, which can not reflect the low-dimensional and multi-modal properties) and (b) provide MoLR-MoG modeling closer to the real-world *practical* data (we conduct real-world experiments to supports our intuition).
> >
> > (ii) The MoE form
> >
> > We acknowledge that the reviewer said that the score form looks like an MoE layer in an LLM. As the MoE structure was first widely used in LLM areas, the spirit of MoE is quickly shared and adopted by other areas. We note that the MoE structure score was not artificially constructed by us, but rather naturally derived from MoLR-MoG modeling and the theoretical basis of diffusion models. This natural form (VAE level MoE score) shares the spirit of MoE and provides a new perspective in designing the NN structure of diffusion models, and we conduct a series of preliminary real-world experiments to support our discussion.
> >
> > **Q3: The Real-world Experiments.**
> >
> > For the real-world experiments, we have provided qualitative and quantitative experimental results on a large ImageNet256 dataset in a limited rebuttal time, which supports our modeling and is consistent with MNIST, CIFAR-10, and our theory. Admittedly, the CLIP score for MoLR-MoG NN is slightly worse than  (but comparable to)  MoLR-Unet, which is as expected (since we require a much smaller NN).
> >
> > We really hope the above discussion can address the reviewer's concerns. We are more than happy and looking forward to further constructive communication.
> >
> >
> >
> >
> >
> > [1]Wang, Peng, et al. "Diffusion models learn low-dimensional distributions via subspace clustering." *The Second Conference on Parsimony and Learning (Recent Spotlight Track)*.
> >
> > [2]Chen, Minshuo, et al. "Score approximation, estimation and distribution recovery of diffusion models on low-dimensional data." *International Conference on Machine Learning*. PMLR, 2023.

---

### Official Review · Reviewer_LLnu · 2025-10-31

**Soundness:** 3
**Presentation:** 3
**Contribution:** 3
**Rating:** 6
**Confidence:** 5

**Summary:**

The paper introduces Mixture of Low-Rank Mixtures of Gaussians (MoLR-MoG), a novel latent modeling framework for diffusion models. In this approach, high-dimensional data, such as images, are represented as a union of low-dimensional subspaces, with each subspace modeled by a mixture-of-Gaussian latent distribution. This modeling naturally induces a mixture-of-experts (MoE) nonlinear score function, enabling the network to capture both multi-modal and nonlinear latent structures far more effectively than prior Gaussian-based latent models. By leveraging this structured latent representation, diffusion models can generate high-quality samples efficiently, even from relatively small datasets. Furthermore, the framework comes with theoretical guarantees on estimation error and convergence, providing a principled explanation for the observed sample efficiency and fast optimization in practical diffusion model training.

**Strengths:**

(1) **Novel latent modeling framework**: The paper introduces MoLR-MoG, which combines low-dimensional subspace modeling with a mixture-of-Gaussian latent structure, enabling diffusion models to capture multi-modal and nonlinear latent structures more effectively than prior Gaussian-based approaches.

(2) **Theoretical contributions**: It provides provable estimation error bounds that show how the model can escape the curse of dimensionality, and also establishes convergence guarantees for gradient descent on the nonlinear MoE-latent score, offering strong theoretical support for the framework.

(3) **Empirical Efficiency**: Experiments demonstrate that the MoE-latent MoG network can generate high-quality samples comparable to a MoE-latent UNet while using 10× fewer parameters, highlighting both practical efficiency and effectiveness of the proposed approach.

**Weaknesses:**

Here are some concerns about this paper:

(1) **Inconsistency between experiments and theoretical results**: In Section 3.1, the authors derive a score network architecture based on the MoLR-MoG model. However, in Section 4, they train 10 VAEs to serve as encoders and decoders before applying the network architecture. This experimental setup appears to deviate from the theoretical framework, and it is unclear how it aligns with the assumptions and analysis presented in Section 3.

(2) **Comparison with the existing literature**: In Wang et al. 2024 (https://arxiv.org/pdf/2409.02426), the experiments on MNIST and CIFAR look much better than those implemented in this paper. Please carefully check it and make a fair comparison.

 (3) **Convergence analysis and empirical validation**: In Lemma 6.10, the authors establish linear convergence for their gradient descent procedure under the MoLR-MoG model. However, it appears that no empirical experiments are provided to support this theoretical claim. Including such validation, for example, by plotting convergence curves or reporting iteration counts for training on real datasets, would help demonstrate that the theoretical guarantees translate to practical performance.

**Questions:**

**Q1.** Is it a good assumption that the latent vector is Gaussian?  The paper assumes that the latent vectors follow a Gaussian distribution, which is a common and convenient choice for theoretical analysis, particularly for deriving sampling complexity and convergence guarantees. While this assumption is understandable for establishing tractable results, it is unclear how realistic it is for modeling real-world data, which may exhibit strongly non-Gaussian or multi-modal latent structures.

**Q2.** Is it possible to improve the sampling efficiency in the reverse process under this MoLR-MoG model?

---

> ### Author Response · Authors · 2025-11-24
> **Rebuttal Part 1:  Discussion on Experiments and Sampling**
>
> Thank you for your valuable comments and suggestions. We provide our response to each question below. For the revised paper, we highlight the new part in green.
>
> **Weakness 1: The relationship between theory, applications, and our real-world experiments.**
>
> In this part, we discuss the role of our experiments, the theoretical modeling, and how to scale up the application. In our current experiments, since MNIST and CIFAR-10 have label information, we employ hard clustering instead of soft clustering (soft weights) used in theoretical modeling, which is also adopted by [2]. More specifically, we train a VAE for each class, which can be viewed as a simplification of soft activation, to demonstrate that with a suitable VAE to embed the input images in the correct manifold, a small 2-layer MoG expert is sufficient to generate images.
>
> For large-scale datasets without labels, we can utilize a clustering algorithm to partition the data into distinct clusters. Then, we can train a VAE encoder, decoder, and latent MoG score for each cluster. For the VAE training, we do not require training the VAE from a sketch. We can LoRA fine-tune a VAE pretrained on large-scale datasets (for example, DC-AE [1] for our ImageNet experiments) for each expert, which shares a pretrained VAE backbone and has a smaller model size. When generating images, we activate different VAE LoRA according to the clustering weight (can choose Top1 or Top2 experts), which matches the spirit of MoE.
>
> **Weakness 2: Comparison with the experiments of Wang et al. 2024.**
>
> In this part, we are more than happy to show the difference between our experiments. As shown in Appendix C of Wang et al. 2024, they *do not learn* the MoLRG and directly use the ground truth score function of MoLRG under a large $d_k$ (which does not match the detected manifold dimension) by calculating SVD (to obtain $U_k$). More specifically, they choose **$d_k=200$** for CIFAR-10. However, [3] shows the $d_k$ for CIFAR-10 is around $d_k=25$, which indicates a mismatch ([3] is an important support of Wang et al.). On the contrary, to achieve a fair comparison, following [3], we first train a VAE for each class. Then, we use exactly the same VAE and train latent Unet, latent MoG NN, and latent Gaussian NN till convergence (please see the loss curve in Fig.3 of our revised paper) and provide the final results, which can more directly show the ability of MoLRG modeling.
>
> **Weakness 3: The Loss Curve on real-world datasets.**
>
> In our revised paper (Fig.3 ), we provide the loss curves of latent Unet, MoG, and Gaussian, which show that the loss of MoE-MoG NN is significantly smaller than that of MoE-Gaussian, is close to that of MoE-Unet, and converges faster, indicating that MoE-MoG NN efficiently approximates the ground-truth score and supports our theoretical results.
>
> We also show the performance of our MoLR-MoG in the larger Imagenet 256 dataset. We focus on the parachute class and first fine-tune the pre-trained VAE [1] with the parachute class to guarantee the VAE can locate the corresponding manifold (please see the influence of expert-specific VAE in our revised paper). Then, we show that the MoLR-MoG NN, still using $ 10\times$ smaller parameters compared with MoLR-Unet (a comparison of latent diffusion models within the manifold, as it uses the same VAE), can achieve comparable performance. On the contrary, MoLR-Gaussian still produces blurry images due to its limited ability, which stems from the linear latent score of the Gaussian (as shown in Fig. 2 of our revised paper). From a quantitative perspective, we calculate the CLIP score using the text prompt "a photo of a parachute". The Clip score for MoLR with Unet, MoG, and Gaussian NN is $0.304$, $0.293$, and $0.254$, respectively, indicating that MoLR-MoG achieves almost comparable text-to-image alignment with MoE-Unet.
>
> We have now added a detailed setting, results, and discussion on experiments in the revision of our paper (Sec 4, highlighted in green).
>
> **Q2: Improvement of Sampling Efficiency with MoLR-MoG.**
>
> When considering the inference complexity and sampling efficiency of the MoE-latent MoG score, as shown in Weakness 1, in the inference time, similar to LLM, we can activate the Top1 or Top2 experts to generate samples. Since our diffusion part has many fewer parameters compared to the deep NN diffusion part (and achieves a good performance), the forward time for MoLR-MoG is smaller. As a result, the total sample time for MoLR-MoG is smaller compared to the deep Unet.

---

> > ### Author Response · Authors · 2025-11-24
> > **Rebuttal Part 2: Extention to Results without highly separated Gaussians**
> >
> > **Q1: Extended  Theoretical Results for Multi-Modal Distribution.**
> >
> > Thanks for the helpful and constructive comments to extend the analysis scope of our work. We note that in this work, the latent distribution for each manifold (each expert) is a Mixture of Gaussians (with $n_k$ modes) instead of a Gaussian, which is multi-modal.  In the estimation part, we make full use of the closed-form of the MoG score to escape the curse of dimensionality.
> >
> > For the optimization, since we assume the highly separated conditions for latent MoG, the optimization is relatively close to the optimization of a Gaussian. We now extend our results to the latent MoG with overlap from almost independent setting. More specifically, we define the pairwise overlap factor $\xi_{i,j}(x)$ between two components $i$ and $j$
> >
> > $$
> > \xi_{i,j}(x) \triangleq r_{k,i}(x) r_{k,j}(x)\,.
> > $$
> >
> > and the Maximum Expected Overlap $\epsilon_{\text{overlap}}$ for the manifold as:
> >
> > $$
> > \epsilon\_{\text{overlap}} = \max\_{i} \sum\_{j \neq i} \mathbb{E}\_{x \sim p_t} [\xi_{i,j}(x)]\,.
> > $$
> >
> > We start from $2$-mode MoG latent as a start to show our intuition (the formal proof for $2$-mode and $K$-mode has been provided in the end of Sec. 6 and Appendix E, highlighted in green). Intuitively speaking, we can bound the additional negative influence in the Hessian introduced by overlap. Then, we can use  Hessian Perturbation Analysis to obtain the smallest eigenvalue under this setting, which leads to strongly convex.
> >
> > We first revisit the derivation of the Jacobian $J_k^\mu$ ($J_k^U$ is similar). In the original derivation, $J_k^\mu$ was decomposed into Term A (dominant term) and Term B (previously ignored term introduced by the overlap):
> >
> > $$
> > J_k^\mu(x) = \underbrace{\text{Term A}(x)}\_{J\_{\text{ideal}}^\mu(x)} + \underbrace{\text{Term B}(x)}\_{E^\mu(x)}.
> > $$
> >
> > When $\xi_k(x) \to 0$, we recover the ideal Jacobian derived previously:
> >
> > $$
> > J_{\text{ideal}}^\mu(x) = -\frac{s_t}{\gamma_t^2}(r_k^{+}(x)-r_k^{-}(x))\left(I-\frac{s_t^2}{s_t^2+\gamma_t^2}U_kU_k^\top\right).
> > $$
> > The term previously ignored (Term B) contains the cross-product of weights, which is exactly our overlap factor $\xi_k(x)$. Specifically:
> > $$
> > E^\mu(x) = -\frac{4 s_t^2}{\gamma_t^2 w_k^2(x)} \cdot \xi_k(x) \cdot \Sigma_k^{-1}x \left(I+\frac{s_t^2}{s_t^2+\gamma_t^2}U_kU_k^\top\right)\mu_k,
> > $$
> >
> > where $w_k(x)=\Sigma_{l=1}^{n_k}\pi_{k,l}\mathcal{N}(x;s_t\mu_{k,l},\Sigma_{k,l})$. Since terms like $\frac{x}{w_k(x)}$ and projection matrices are bounded within the support, there exists a constant $C_1$ such that:
> >
> > $$
> > \\|E^\mu(x)\\|\_2 \leq C_1 \cdot \xi_k(x).
> > $$
> >
> > Similarly, we can decompose $J_k^U(x)$ it into an ideal part and an error part:
> >
> > $$
> > J_k^U(x) = J\_{\text{ideal}}^U(x) + E^U(x), \quad \text{where } \\|E^U(x)\\|\_F \leq C_2 \cdot \xi\_k(x).
> > $$
> >
> > The Hessian matrix $H$ is defined as the expected outer product of the Jacobians:
> >
> > $$
> > H = \mathbb{E}_{x \sim p_t(x)} [J(x)J(x)^\top].
> > $$
> >
> > Let $J(x) = J_{\text{ideal}}(x) + E(x)$. Substituting this into the Hessian definition:
> >
> > $$
> > H = \underbrace{\mathbb{E}[J\_{\text{ideal}}J\_{\text{ideal}}^\top]}_{H\_{\text{ideal}}} + \underbrace{\mathbb{E}[J\_{\text{ideal}}E^\top + E J\_{\text{ideal}}^\top + E E^\top]}\_{\Delta H}\,,
> > $$
> >
> > Where $H_{\text{ideal}}$ is the Hessian matrix under the high separation assumption and $\Delta H$ is the perturbation matrix induced by the overlap.
> >
> > From the previous proof (assuming separation), we established that $H_{\text{ideal}}$ is block-diagonal (or has negligible off-diagonals due to symmetry) and positive definite. Let $\alpha > 0$ be its minimum eigenvalue:
> >
> > $$
> > \lambda_{\min}(H_{\text{ideal}}) \geq {(1-4\epsilon_{\text{overlap}})}\min{(\lambda_{\min}(H_{\mu_{k}\mu_{k}}),\lambda_{\min}(H_{U_{k}U_{k}}))} \triangleq \alpha,
> > $$
> > where $\lambda_{\min}(H_{\mu_{k}\mu_{k}})$ and $\lambda_{\min}(H_{U_{k}U_{k}})$ has been provided  in Lem. 6.7.
> >
> > For $\Delta H$, we know that
> > $$
> > \\|\Delta H\\|_2 \leq 2\\|\mathbb{E}[J\_{\text{ideal}}E^\top]\\|_2 + \\|\mathbb{E}[EE^\top]\\|_2 \leq 2 \sqrt{\mathbb{E}[\|J\_{\text{ideal}}\|^2] \mathbb{E}[\|E\|^2]} + \mathbb{E}[\|E\|^2].
> > $$
> >
> > Following the above calculation, the perturbation norm is dominated by the overlap factor:
> >
> > $$
> > \\|\Delta H\\|_2 \leq C_3 \cdot \epsilon\_{\text{overlap}}
> > $$
> >
> > where $C_3$ is a constant depending on the bounds of the Jacobians.
> >
> > *Positive Definiteness via Weyl's Inequality*
> >
> > With Weyl's Inequality for Hermitian Matrices, we have:
> > Let $H = H_{\text{ideal}} + \Delta H$. The eigenvalues of $H$ are bounded by  ($C_3$ has been provided in Lem. 6.11 and 6.12):
> > $$
> > \lambda_{\min}(H) \geq \lambda_{\min}(H_{\text{ideal}}) - \\|\Delta H\\|\_2 \geq \alpha - C_3 \cdot \epsilon_{\text{overlap}}.
> > $$
> >
> > We note that though the above analysis goes beyond the separation lemma, we still require $\epsilon_{\text{overlap}}$ to be small enough. From the empirical perspective, we test the $\xi_{i,j}$ under the ImageNet256 (parachute) and show that $\xi_{i,j}$ is smaller than $0.001$, which is small.

---

> > > ### Author Response · Authors · 2025-11-24
> > > **Reference**
> > >
> > > [1] Chen, Junyu, et al. "Deep Compression Autoencoder for Efficient High-Resolution Diffusion Models." *The Thirteenth International Conference on Learning Representations*.
> > >
> > > [2]Wang, Peng, et al. "Diffusion models learn low-dimensional distributions via subspace clustering." *The Second Conference on Parsimony and Learning (Recent Spotlight Track)*.
> > >
> > > [3] Brown, Bradley CA, et al. "Verifying the Union of Manifolds Hypothesis for Image Data." *The Eleventh International Conference on Learning Representations*.

---

> > ### Comment · Reviewer_LLnu · 2025-11-27
> >
> > Thank the authors for addressing my concerns and questions! I am generally satisfied with the responses.
> >
> > - Based on the authors' response to **Weakness 1**, I have the following question:
> >
> > The authors train 10 VAEs to represent K-low-dimensional manifolds and also use them to generate images. This implies that real-world image datasets have good representations in the latent space instead of in the pixel space. Is it more reasonable to assume that latent representations have a MoLR-MoG structure?
> >
> > - Based on the authors' response to **Weakness 2**, do the authors also apply VAEs for MoLR-UNet and MoLR-Gaussian?
> >
> > I will be willing to raise my score if the authors are able to further address my comments.

---

> > > ### Author Response · Authors · 2025-11-27
> > > **The Discussion on MoLR-MoG and Experiment Setting**
> > >
> > > Dear Reviewer LLnu,
> > >
> > > Thanks for the positive feedback on our rebuttal and for further constructive comments. We provide our response to each question below.
> > >
> > > **Q1: The Discussion of MoLR-MoG.**
> > >
> > > As the kind and professional reviewer points out, *"real-world image datasets have good representations in the latent space instead of in the pixel space"*, this work aims to make use of the union of the low-dimensional subspace property (corresponding to the MoLR part) and the multi-modal property of real-world images to improve the training process.
> > >
> > > As shown in [1], they verify the MoLR property in the pixel space instead of the latent space (the image in the pixel space can be decomposed into a union of low-dimensional manifolds). As a result, we and [2] all assume the image in the pixel space satisfies the MoLR property to match the empirical observation. For the latent representation, we further assume that latent representations have a MoG structure (which further reflects the multi-modal property). These two parts consist of our MoLR-MoG. The role of MoLR is to map the pixel space into the $K$ low-dimensional manifolds (through $A_k$), and the MoG plays the role to model the multi-modal property of latent representations. Hence, it is more reasonable to model MoLR-MoG in the pixel space.
> > >
> > > We hope our clarification on MoLR-MoG will help resolve the reviewer's concern.
> > >
> > > **Q2: The Experimental setting of MoLR-Unet and MoLR-Gaussian.**
> > >
> > > For strictly fair comparison, we apply exactly the same VAEs for MoLR-MoG, MoLR-Unet, and MoLR-Gaussian to map the pixel space images into corresponding low-dimensional manifolds, which guarantees these three settings face the same latents.
> > >
> > > We really hope the above discussion can address the reviewer's concerns. We are more than happy and looking forward to further constructive communication.
> > >
> > > [1] Brown, Bradley CA, et al. "Verifying the Union of Manifolds Hypothesis for Image Data." *The Eleventh International Conference on Learning Representations*.
> > >
> > > [2] Wang, Peng, et al. "Diffusion models learn low-dimensional distributions via subspace clustering." *The Second Conference on Parsimony and Learning (Recent Spotlight Track)*.

---

> > > > ### Comment · Reviewer_LLnu · 2025-11-28
> > > >
> > > > Thanks for the response. I am satisfied that the authors have addressed my concerns. I am willing to raise my score to 8, but it appears that the review system is now closed. I will contact the AE regarding this.

---

### Official Review · Reviewer_DPpX · 2025-11-01

**Soundness:** 3
**Presentation:** 3
**Contribution:** 3
**Rating:** 6
**Confidence:** 3

**Summary:**

The paper introduces the Mixture of Low-Rank Mixture of Gaussians (MoLR-MoG) distribution, which is employed to model both the real data distribution and the neural network architecture. Empirically, the authors demonstrate that the MoLR-MoG distribution provides a reasonable approximation to the data distribution learned by a U-Net–based architecture. Theoretically, they prove that, under the MoLR-MoG assumption, diffusion model training can escape the curse of dimensionality. Furthermore, by assuming subspace separability, the paper establishes local strong convexity properties for the training dynamics.

**Strengths:**

1. The paper introduces a novel MoLR-MoG modeling framework that is applied to both the training data distribution and the network architecture design.

2. Empirically, the authors validate the effectiveness of this modeling by demonstrating performance comparable to that of the standard U-Net architecture.

3. Theoretically, they show that the MoLR-MoG framework mitigates the curse of dimensionality and establishes local strong convexity in the loss landscape. These theoretical results contribute valuable insights to the broader understanding of diffusion models from a theoretical perspective.

**Weaknesses:**

1. As a theoretically tractable model, MoLR-MoG inevitably exhibits a gap from real image distributions. As shown in Figure 2, there remains a noticeable discrepancy between CIFAR-10 images generated by MoLR-MoG and those from the true CIFAR-10 dataset.

2. The assumption of highly separated Gaussian components (Assumption 6.6) is unrealistic for real-world image data. In practice, different image classes often share overlapping or correlated semantic subspaces. Consequently, the condition required for local strong convexity in the loss landscape is unlikely to hold for real image distributions.

**Questions:**

1. What exactly is the MoLR-UNet used in Section 4? Why does the MoLR-UNet produce worse results than the standard U-Net architecture on MNIST and CIFAR-10, as shown in Figure 2?

---

> ### Author Response · Authors · 2025-11-24
> **Rebuttal Part 1:  The Experiments Setting, Performance, and Larger Real-world experiments**
>
> Thank you for your valuable comments and suggestions. We provide our response to each question below. For the revised paper, we highlight the new part in green.
>
> **Weakness 1 & Q1: The Experiments Setting, Performance, and Larger Real-world experiments and Quantitative Results (ImageNet256).**
>
> (a) The Performance of Unet and MoLR-MoG modeling.
>
> As shown in Appendix F, we adopt a shallow Unet ($2$-layer Unet for MNIST and $6$-layer Unet for CIFAR-10) for a preliminary experiment. To achieve a better performance, we can adopt a deeper Unet, which also results in significantly larger parameters. For MoLR-MoG modeling, we can also increase the number of modal $n_k$ for the latent MoG to improve performance.
>
> (b) Larger Real-world Experiments
>
> To further verify the above discussion, we extend our experiments to the larger ImageNet256 dataset and use an $11$-layer Unet, setting $n_k=40$ for MoLR-MoG, which achieves good performance. We focus on the parachute class and first fine-tune the pre-trained VAE [1] with the parachute class to guarantee the VAE can locate the corresponding manifold (please see the influence of expert-specific VAE in our revised paper). Then, we show that the MoLR-MoG NN, still using $ 10\times$ smaller parameters compared with MoLR-Unet (a comparison of latent diffusion models within the manifold, as it uses the same VAE), can achieve comparable performance. On the contrary, MoLR-Gaussian still produces blurry images due to its limited ability, which stems from the linear latent score of the Gaussian (as shown in Fig. 2 of our revised paper). From a quantitative perspective, we calculate the CLIP score using the text prompt "a photo of a parachute". The Clip score for MoLR with Unet, MoG, and Gaussian NN is $0.304$, $0.293$, and $0.254$, respectively, indicating that MoLR-MoG achieves almost comparable text-to-image alignment with MoE-Unet.
>
> We have now added a detailed setting, results, and discussion on experiments in the revision of our paper (Sec 4, highlighted in green).

---

> ### Author Response · Authors · 2025-11-24
> **Rebuttal Part 2: Extention to Results without highly separated Gaussians**
>
> **Weakness 2: Extended  Theoretical Results without highly separated Gaussians**
>
> Thanks for the helpful and constructive comments to extend the analysis scope of our work. We now extend our results to the latent MoG with overlap from the almost independent setting. More specifically, we define the pairwise overlap factor $\xi_{i,j}(x)$ between two components $i$ and $j$
>
> $$
> \xi_{i,j}(x) \triangleq r_{k,i}(x) r_{k,j}(x).
> $$
>
> and the Maximum Expected Overlap $\epsilon_{\text{overlap}}$ for the manifold as:
>
> $$
> \epsilon\_{\text{overlap}} = \max\_{i} \sum\_{j \neq i} \mathbb{E}\_{x \sim p_t} [\xi_{i,j}(x)].
> $$
>
> We start from $2$-mode MoG latent as a start to show our intuition (the formal proof for $2$-mode and $K$-mode has been provided in the end of Sec. 6 and Appendix E, highlighted in green). Intuitively speaking, we can bound the additional negative influence in the Hessian introduced by overlap. Then, we can use  Hessian Perturbation Analysis to obtain the smallest eigenvalue under this setting, which leads to strongly convex.
>
> Similar to the process of our main content, we first revisit the derivation of the Jacobian $J_k^\mu$ ($J_k^U$ is similar). In the original derivation, $J_k^\mu$ was decomposed into Term A (dominant term) and Term B (previously ignored term introduced by the overlap):
>
> $$
> J_k^\mu(x) = \underbrace{\text{Term A}(x)}\_{J\_{\text{ideal}}^\mu(x)} + \underbrace{\text{Term B}(x)}\_{E^\mu(x)}.
> $$
>
> When $\xi_k(x) \to 0$, we recover the ideal Jacobian derived previously:
>
> $$
> J_{\text{ideal}}^\mu(x) = -\frac{s_t}{\gamma_t^2}(r_k^{+}(x)-r_k^{-}(x))\left(I-\frac{s_t^2}{s_t^2+\gamma_t^2}U_kU_k^\top\right).
> $$
> The term previously ignored (Term B) contains the cross-product of weights, which is exactly our overlap factor $\xi_k(x)$. Specifically:
> $$
> E^\mu(x) = -\frac{4 s_t^2}{\gamma_t^2 w_k^2(x)} \cdot \xi_k(x) \cdot \Sigma_k^{-1}x \left(I+\frac{s_t^2}{s_t^2+\gamma_t^2}U_kU_k^\top\right)\mu_k,
> $$
>
> where $w_k(x)=\Sigma_{l=1}^{n_k}\pi_{k,l}\mathcal{N}(x;s_t\mu_{k,l},\Sigma_{k,l})$. Since terms like $\frac{x}{w_k(x)}$ and projection matrices are bounded within the support, there exists a constant $C_1$ such that:
>
> $$
> \\|E^\mu(x)\\|\_2 \leq C_1 \cdot \xi_k(x).
> $$
>
> Similarly, for the Jacobian with respect to $U_k$, we can decompose it into an ideal part and an error part proportional to the overlap:
>
> $$
> J_k^U(x) = J\_{\text{ideal}}^U(x) + E^U(x), \quad \text{where } \\|E^U(x)\\|\_F \leq C_2 \cdot \xi\_k(x).
> $$
>
> The Hessian matrix $H$ is defined as the expected outer product of the Jacobians:
>
> $$
> H = \mathbb{E}_{x \sim p_t(x)} [J(x)J(x)^\top].
> $$
>
> Let $J(x) = J_{\text{ideal}}(x) + E(x)$. Substituting this into the Hessian definition:
>
> $$
> \begin{aligned}
> H = \mathbb{E} \left[ (J\_{\text{ideal}} + E)(J\_{\text{ideal}} + E)^\top \right] = \underbrace{\mathbb{E}[J\_{\text{ideal}}J\_{\text{ideal}}^\top]}_{H\_{\text{ideal}}} + \underbrace{\mathbb{E}[J\_{\text{ideal}}E^\top + E J\_{\text{ideal}}^\top + E E^\top]}\_{\Delta H},
> \end{aligned}
> $$
>
> Where $H_{\text{ideal}}$ is the Hessian matrix under the high separation assumption and $\Delta H$ is the perturbation matrix induced by the overlap.
>
> From the previous proof (assuming separation), we established that $H_{\text{ideal}}$ is block-diagonal (or has negligible off-diagonals due to symmetry) and positive definite. Let $\alpha > 0$ be its minimum eigenvalue:
>
> $$
> \lambda_{\min}(H_{\text{ideal}}) \geq {(1-4\epsilon_{\text{overlap}})}\min{(\lambda_{\min}(H_{\mu_{k}\mu_{k}}),\lambda_{\min}(H_{U_{k}U_{k}}))} \triangleq \alpha,
> $$
> where $\lambda_{\min}(H_{\mu_{k}\mu_{k}})$ and $\lambda_{\min}(H_{U_{k}U_{k}})$ has been provided  in Lemma 6.7.
>
> For $\Delta H$, we know that
> $$
> \begin{aligned}
> \\|\Delta H\\|_2 \leq 2\\|\mathbb{E}[J\_{\text{ideal}}E^\top]\\|_2 + \\|\mathbb{E}[EE^\top]\\|_2 \leq 2 \sqrt{\mathbb{E}[\|J\_{\text{ideal}}\|^2] \mathbb{E}[\|E\|^2]} + \mathbb{E}[\|E\|^2].
> \end{aligned}
> $$
>
> Following the above calculation, the perturbation norm is dominated by the overlap factor:
>
> $$
> \\|\Delta H\\|_2 \leq C_3 \cdot \epsilon\_{\text{overlap}}
> $$
>
> where $C_3$ is a constant depending on the bounds of the Jacobians.
>
> *Positive Definiteness via Weyl's Inequality*
>
> With Weyl's Inequality for Hermitian Matrices, we have:
> Let $H = H_{\text{ideal}} + \Delta H$. The eigenvalues of $H$ are bounded by ($C_3$ has been provided in Lem. 6.11 and 6.12):
> $$
> \lambda_{\min}(H) \geq \lambda_{\min}(H_{\text{ideal}}) - \\|\Delta H\\|\_2 \geq \alpha - C_3 \cdot \epsilon_{\text{overlap}}.
> $$
>
> We note that though the above analysis go beyond the separation lemma, we still require $\epsilon_{\text{overlap}}$ is small enough. From the empirical perspective, we test the $\xi_{i,j}$ under the ImageNet256 (parachute) and show that $\xi_{i,j}$ is smaller than $0.001$, which is small.
>
>
>
> [1] Chen, Junyu, et al. "Deep Compression Autoencoder for Efficient High-Resolution Diffusion Models." *The Thirteenth International Conference on Learning Representations*.

---

### Official Review · Reviewer_qmDg · 2025-11-02

**Soundness:** 3
**Presentation:** 2
**Contribution:** 2
**Rating:** 4
**Confidence:** 3

**Summary:**

This paper introduces MoLR-MoG (Mixture of Low-Rank Mixture of Gaussians) modeling for diffusion models, aiming to better reflect the multi-subspace and multi-modal structure of real-world data. The authors derive a closed-form score function under this model, which naturally exhibits a Mixture-of-Experts (MoE) structure. They provide finite-sample estimation error bounds that avoid the curse of dimensionality by depending on latent dimensions and mixture counts rather than the ambient dimension. Additionally, they prove local strong convexity of the score-matching objective under MoLR-MoG, yielding gradient descent convergence guarantees.

**Strengths:**

1. MoLR-MoG is a novel generative prior that generalizes prior single-subspace Gaussian latent assumptions. The nonlinear MoG score derivation and its MoE interpretation are new theoretical contributions.
2. The paper delivers rigorous generalization bounds and optimization theory  that are tight up to log factors and explicit in all problem constants.

**Weaknesses:**

1. Experiments are small-scale (MNIST/CIFAR-10). No evaluation on high-resolution images (e.g., ImageNet) or complex datasets (e.g., text-to-image, multi-resolution). FID, LPIPS, or human evaluations are absent; only visual samples are shown. Additionally, the time comparison of the proposed new diffusion model with the original diffusion is lacking.
2. Theoretical guarantees rely on Δ ≫ γ_t (Assumption 6.1) and highly separated Gaussians (Assumption 6.6). No discussion of relaxation regimes or failure modes when clusters overlap significantly, which is common in real data.
3. MoLR-MoG needs K VAEs (one per subspace) and clustering at inference. Complexity scales linearly with K; no ablation on K vs. performance vs. compute, nor comparison with single-VAE baselines.

**Questions:**

Please refer to the weaknesses.

**Details Of Ethics Concerns:**

None.

---

> ### Author Response · Authors · 2025-11-24
> **Rebuttal Part 1: Larger Real World Experiments**
>
> Thank you for your valuable comments and suggestions.  We provide our response to each question below. For the revised paper, we highlight the new part in green.
>
> **Weakness 1: Larger Real-world experiments and Quantitative Results (ImageNet256).**
>
> We extend our experiments to the larger ImageNet256 dataset and show that the MoLR-MoG NN, using $ 10\times$ smaller parameters compared with MoLR-Unet (a Comparison of latent diffusion models within the manifold, as it uses the same VAE), can achieve comparable performance. On the contrary, MoLR-Gaussian still produces blurry images due to its limited ability, which stems from the linear latent score of the Gaussian (as shown in Fig. 2 of our revised paper).
>
> (a) The relationship between theory, applications, and our real-world experiments.
>
> Before discussing the results, we first discuss the role of our experiments, the theoretical modeling, and how to scale up the application. In our current experiments, since MNIST and CIFAR-10 have label information, we employ hard clustering instead of soft clustering (soft weights) used in theoretical modeling, which is also adopted by [2]. More specifically, we train a VAE for each class, which can be viewed as a simplification of soft activation, to demonstrate that with a suitable VAE to embed the input images in the correct manifold, a small 2-layer MoG expert is sufficient to generate images.
>
> For large-scale datasets without labels, we can utilize a clustering algorithm to partition the data into distinct clusters. Then, we can train a VAE encoder, decoder, and latent MoG score for each cluster. For the VAE training, we do not require training the VAE from a sketch. We can LoRA fine-tune a VAE pretrained on large-scale datasets (for example, DC-AE [1] for our ImageNet experiments) for each expert, which shares a pretrained VAE backbone and has a smaller model size. When generating images, we activate different VAE LoRA according to the clustering weight (can choose Top1 or Top2 experts), which matches the spirit of MoE.
>
> (b) Experiments Discussion and Quantitative Results
> As a start, we focus on the parachute class of ImageNet256 (to achieve a good performance with the whole ImageNet256 dataset, we can follow the above process to do clustering, VAE LoRA, and MoG experts training). More specifically, we use $1k$ parachute images to fine-tune the good pretrained DC-AE to help the VAE encode the images to the parachute manifold. Then, we train the Unet, 2-layer SoftMax GMM NN, and linear Gaussian NN with these parachute images. As shown in Fig.2, the generated results of MoLR-MoG are comparable to the generated results of MoLR-Unet.
> From a quantitative perspective, we calculate the CLIP score for the parachute class in ImageNet using the text prompt "a photo of a parachute". The Clip score for MoLR with Unet, MoG, and Gaussian NN is $0.304$, $0.293$, and $0.254$, respectively, indicating that MoLR-MoG achieves almost comparable text-to-image alignment with MoE-Unet. We also provide the loss curves of latent Unet, MoG, and Gaussian, which show that the loss of MoE-MoG NN is significantly smaller than that of MoE-Gaussian and is close to that of MoE-Unet, indicating that MoE-MoG NN efficiently approximates the ground-truth score and supports our theoretical results.
>
> We have now added a detailed results and discussion in the revision of our paper (Sec 4, highlighted in green).
>
> **Weakness 3: The Performance, Training and Inference Complexity of $K$ VAE.**
>
> (a) Expert-specific VAE and the comparison with single VAE
>
> In this part, we also highlight the role of expert-specific VAE. As shown in the score of MoLR-MoG, different from latent diffusion models with a single VAE, there are $K$ VAEs to encode the input to the corresponding manifold. We note that this operation is important for MoLR-MoG with small MoG experts. Otherwise, with a unified VAE, the latent space is complex and requires a large latent space, such as Unet. As shown in Fig.3 of our revised paper, with a unified VAE, the unified latent is complex, and a MoG expert can not learn a meaningful image with the target class. Hence, with a unified VAE, latent diffusion models require a large latent Unet. However, with an expert-specific VAE (for example, we fine-tune the pretrained VAE with the parachute class dataset), the latent manifold becomes simple, and latent MoG experts are enough to generate clear models, which also supports our theoretical modeling.
> We have now added a detailed results and discussion in the revision of our paper (The last paragraph of Sec 4, highlighted in green).
>
> (b) The inference complexity
>
> When considering the inference complexity of MoE-latent MoG score, as shown in Weakness 1 (a), in the inference time, similar to LLM, we can activate the Top1 or Top2 experts to generate samples. Since our diffusion part has many fewer parameters compared to the deep NN diffusion part, the inference time will not be slower (even faster).

---

> ### Author Response · Authors · 2025-11-24
> **Rebuttal Part 2: Extention to Results without highly separated Gaussians**
>
> **Weakness 2: Extended Theoretical Results without highly separated Gaussians**
>
> Thanks for the helpful and constructive comments to extend the analysis scope of our work. We now extend our results to the latent MoG with overlap from almost independent setting. More specifically, we define the pairwise overlap factor $\xi_{i,j}(x)$ between two components $i$ and $j$
>
> $$
> \xi_{i,j}(x) \triangleq r_{k,i}(x) r_{k,j}(x)\,.
> $$
>
> and the Maximum Expected Overlap $\epsilon_{\text{overlap}}$ for the manifold as:
>
> $$
> \epsilon\_{\text{overlap}} = \max\_{i} \sum\_{j \neq i} \mathbb{E}\_{x \sim p_t} [\xi_{i,j}(x)]\,.
> $$
>
> We start from $2$-mode MoG latent as a start to show our intuition (the formal proof for $2$-mode and $K$-mode has been provided in the end of Sec. 6 and Appendix E, highlighted in green). Intuitively speaking, we can bound the additional negative influence in the Hessian introduced by overlap. Then, we can use  Hessian Perturbation Analysis to obtain the smallest eigenvalue under this setting, which leads to strongly convex.
>
> Similar to the process of our main content, we first revisit the derivation of the Jacobian $J_k^\mu$ ($J_k^U$ is similar). In the original derivation, $J_k^\mu$ was decomposed into Term A (dominant term) and Term B (previously ignored term introduced by the overlap):
>
> $$
> J_k^\mu(x) = \underbrace{\text{Term A}(x)}\_{J\_{\text{ideal}}^\mu(x)} + \underbrace{\text{Term B}(x)}\_{E^\mu(x)}.
> $$
>
> When $\xi_k(x) \to 0$, we recover the ideal Jacobian derived previously:
>
> $$
> J_{\text{ideal}}^\mu(x) = -\frac{s_t}{\gamma_t^2}(r_k^{+}(x)-r_k^{-}(x))\left(I-\frac{s_t^2}{s_t^2+\gamma_t^2}U_kU_k^\top\right).
> $$
> The term previously ignored (Term B) contains the cross-product of weights, which is exactly our overlap factor $\xi_k(x)$. Specifically:
> $$
> E^\mu(x) = -\frac{4 s_t^2}{\gamma_t^2 w_k^2(x)} \cdot \xi_k(x) \cdot \Sigma_k^{-1}x \left(I+\frac{s_t^2}{s_t^2+\gamma_t^2}U_kU_k^\top\right)\mu_k,
> $$
>
> where $w_k(x)=\Sigma_{l=1}^{n_k}\pi_{k,l}\mathcal{N}(x;s_t\mu_{k,l},\Sigma_{k,l})$. Since terms like $\frac{x}{w_k(x)}$ and projection matrices are bounded within the support, there exists a constant $C_1$ such that:
>
> $$
> \\|E^\mu(x)\\|\_2 \leq C_1 \cdot \xi_k(x).
> $$
>
> Similarly, for the Jacobian with respect to $U_k$, we can decompose it into an ideal part and an error part proportional to the overlap:
>
> $$
> J_k^U(x) = J\_{\text{ideal}}^U(x) + E^U(x), \quad \text{where } \\|E^U(x)\\|\_F \leq C_2 \cdot \xi\_k(x).
> $$
>
> The Hessian matrix $H$ is defined as the expected outer product of the Jacobians:
>
> $$
> H = \mathbb{E}_{x \sim p_t(x)} [J(x)J(x)^\top].
> $$
>
> Let $J(x) = J_{\text{ideal}}(x) + E(x)$. Substituting this into the Hessian definition:
>
> $$
> \begin{aligned}
> H = \mathbb{E} \left[ (J\_{\text{ideal}} + E)(J\_{\text{ideal}} + E)^\top \right] = \underbrace{\mathbb{E}[J\_{\text{ideal}}J\_{\text{ideal}}^\top]}_{H\_{\text{ideal}}} + \underbrace{\mathbb{E}[J\_{\text{ideal}}E^\top + E J\_{\text{ideal}}^\top + E E^\top]}\_{\Delta H}\,,
> \end{aligned}
> $$
>
> Where $H_{\text{ideal}}$ is the Hessian matrix under the high separation assumption and $\Delta H$ is the perturbation matrix induced by the overlap.
>
> From the previous proof (assuming separation), we established that $H_{\text{ideal}}$ is block-diagonal (or has negligible off-diagonals due to symmetry) and positive definite. Let $\alpha > 0$ be its minimum eigenvalue:
>
> $$
> \lambda_{\min}(H_{\text{ideal}}) \geq {(1-4\epsilon_{\text{overlap}})}\min{(\lambda_{\min}(H_{\mu_{k}\mu_{k}}),\lambda_{\min}(H_{U_{k}U_{k}}))} \triangleq \alpha,
> $$
> where $\lambda_{\min}(H_{\mu_{k}\mu_{k}})$ and $\lambda_{\min}(H_{U_{k}U_{k}})$ has been provided  in Lemma 6.7.
>
> For $\Delta H$, we know that
> $$
> \begin{aligned}
> \\|\Delta H\\|_2 \leq 2\\|\mathbb{E}[J\_{\text{ideal}}E^\top]\\|_2 + \\|\mathbb{E}[EE^\top]\\|_2 \leq 2 \sqrt{\mathbb{E}[\|J\_{\text{ideal}}\|^2] \mathbb{E}[\|E\|^2]} + \mathbb{E}[\|E\|^2].
> \end{aligned}
> $$
>
> Following the above calculation, the perturbation norm is dominated by the overlap factor:
>
> $$
> \\|\Delta H\\|_2 \leq C_3 \cdot \epsilon\_{\text{overlap}}
> $$
>
> where $C_3$ is a constant depending on the bounds of the Jacobians.
>
> *Positive Definiteness via Weyl's Inequality*
>
> With Weyl's Inequality for Hermitian Matrices, we have:
> Let $H = H_{\text{ideal}} + \Delta H$. The eigenvalues of $H$ are bounded by  ($C_3$ has been provided in Lem. 6.11 and 6.12):
>
> $$
> \lambda_{\min}(H) \geq \lambda_{\min}(H_{\text{ideal}}) - \\|\Delta H\\|\_2 \geq \alpha - C_3 \cdot \epsilon_{\text{overlap}}.
> $$
>
>
> where require:
>
> $$\alpha - C_3 \cdot \epsilon_{\text{overlap}} > 0 \implies \epsilon_{\text{overlap}} < \frac{\alpha}{C_3}.$$
>
> We note that though the above analysis goes beyond the separation lemma, we still require $\epsilon_{\text{overlap}}$ to be small enough. From the empirical perspective, we test the $\xi_{i,j}$ under the ImageNet256 (parachute) and show that $\xi_{i,j}$ is smaller than $0.001$, which is small.

---

> > ### Author Response · Authors · 2025-11-24
> > **Reference**
> >
> > [1] Chen, Junyu, et al. "Deep Compression Autoencoder for Efficient High-Resolution Diffusion Models." *The Thirteenth International Conference on Learning Representations*.
> >
> > [2]Wang, Peng, et al. "Diffusion models learn low-dimensional distributions via subspace clustering." *The Second Conference on Parsimony and Learning (Recent Spotlight Track)*.

---

### Author Response · Authors · 2025-12-02
**Summary of Rebuttal**

Dear Area Chair,

Thanks for the valuable time in dealing with our submission under the special circumstances. To facilitate a quick understanding of  our work and rebuttal, we provide a concise summary below.

First of all, we were fortunate to have a chance to discuss our works with kind reviewers based on their constructive comments. We thank reviewer LLnu for the detailed multi-round discussion, positive feedback (on our additional experiments and theory), and clear support (with an increasing score from 6 to 8 before rollback); and the positive feedback on our additional theory and open attitude to the theoretical works of reviewer fdxV. We are also happy to share our insight and intuition in addressing the additional concerns (about insight) of the reviewer fdxV that arose during the discussion phase.

The shared constructive comments of reviewers are that (a) conducting larger-scale experiments (for example, ImageNet 256) to support our theoretical results; (b) extending our optimization analysis to multi-modal distribution with overlap instead of almost independent. We have provided additional experiments (sec. 4) and theory (sec. 6.3) in our revised paper (highlighted in green) and in the rebuttal comments. We also provide responses for each question and the weaknesses of reviewers.

In the following part, we briefly summarize the feedback for the common comments and additional concerns (provided by reviewer fdxV) in the rebuttal phase.

**(a) Larger Scale Real-world experiments and Quantitative Results**

We extend our experiments (MNIST and CIFAR10) to the larger ImageNet256 dataset and show that the MoLR-MoG NN, using $ 10\times$ smaller parameters (a 2-layer Softmax score for latent space) compared with MoLR-Unet (a comparison of latent diffusion models within the manifold, as these models use the same VAE), can still achieve comparable performance. On the contrary, MoLR-Gaussian still produces blurry images due to its limited ability, which stems from the linear latent score of the Gaussian (as shown in Fig. 2 of our revised paper).

From the quantitative, we provide CLIP score and training loss curve. The Clip score for MoLR with Unet, MoG, and Gaussian NN is $0.304$, $0.293$, and $0.254$, respectively, indicating that MoLR-MoG achieves almost comparable text-to-image alignment with MoE-Unet. For the loss curve, we show that the loss of MoE-MoG NN is significantly smaller than that of MoE-Gaussian and is close to that of MoE-Unet, indicating that MoE-MoG NN efficiently approximates the ground-truth score and supports our theoretical results.

**(b) Extended Theoretical Results without highly separated Gaussians.**

In the rebuttal phase, we extend our optimizaiton analysis to the latent MoG with overlap from almost independent setting by defining the pairwise overlap factor $\xi_{i,j}(x)$ between two components $i$ and $j$ of MoG and clearly showing the influence of $\xi\_{i,j}$ in the strongly convex parameters. Feedback from reviewer LLnu and fdxV are positive for this extension.

**(c) The Insight**

As the solid theoretical foundation of the diffusion model is one of the key reasons for its charm, we aim to bridge the gap between theory and practice from a theoretical perspective and make them truly useful in practice by starting from a natural intuition, conduct real-world experiments and providing theoretical guarantees.

Our work follows a natural line that (a) first observe the shortcomings of existing theoretical *standard* modeling (the union of low dimensional modeling with Gaussian latent and MoG modeling in the full space, which can not reflect the low-dimensional and multi-modal properties) and (b) provide MoLR-MoG modeling closer to the real-world *practical* data.

Our insight is that with suitable MoLR-MoG modeling (which is closer to *practical* data compared to the current *standard* assumption), we can naturally induce NN with a special structure (a natural MoE structure with multi-modal property of real-world data) and enjoy a much smaller number of parameters (to achieve the comparable performance with NN), which provides a new perspective in designing lighter diffusion models.

This natural MoE score (VAE level MoE score) shares the spirit of MoE and provides a new perspective in designing the NN structure of diffusion models. As a start, we conduct a series of preliminary real-world experiments to support our discussion. Then, our estimation error and optimization analysis provide a theoretical basis for our modeling.

---

### Meta-Review · Area_Chair_rC5B · 2026-01-08

**Summary:**

This paper proposes the mixture subspace of low-rank mixture of Gaussian (MoLR-MoG) modeling. The scores are 4, 6, 8 and 4 *(Please see the Reviewer Scores section)*. The strength is on its rigorous theory guarantee, novel latent modeling framework and good performance. Considering the comments of all reviewers, I tend to accept.

**Reviewer Concerns:**

The author responded in detail to the reviewers' concerns. some of the reviewers participated in the discussions and raised the scores.

**Reviewer Scores:**

The scores are 4, 6, 6 and 4. But Reviewer LLnu explicitly said to raise the score from 6 to 8. So the final scores are 4, 6, 8 and 4. https://openreview.net/forum?id=MPWIM6rxxU&noteId=8RPevwjSyQ

---

### Decision · Program_Chairs · 2026-01-26

Accept (Poster)